# Geometric Rate–Distortion Invariance for Domain Generalization

**Tong Liu** [1 2]   **Sen Liang** [1 2]   **Shuo Bai** [1 2]

## Abstract

Domain generalization (DG) aims to learn representations that remain predictive under distribution shifts. A key challenge is that the target domain is unobserved during training, which complicates the search for invariant representations: alignment objectives that ignore discriminative structure can become ill-conditioned under finite samples. This calls for shaping the geometry of class-conditional representations across domains, not merely matching their distributions. We propose Geometric **R**ate–**D**istortion **I**nvariance (**RDI**), a DG framework that realizes this principle by generalizing classical rate–distortion theory to Grassmann manifolds. **RDI** models class-conditional representations as low-dimensional subspaces and formulates DG as a joint optimization of (i) cross-domain subspace alignment (geometric distortion) and (ii) spectral–volumetric complexity (a capacity-regularized rate term), promoting stable alignment while preventing the collapse of discriminative geometry. We provide finite-sample stability guarantees under bounded shifts and show on DomainBed that **RDI** is competitive with strong DG baselines, with ablations confirming that both alignment and complexity control are necessary for reliable generalization.

## 1. Introduction

Deep neural networks achieve impressive performance when training and test distributions match, yet generalize poorly under distribution shifts (Bengio et al., 2013; Krizhevsky et al., 2012). Domain generalization (DG) addresses this by learning models that generalize to unseen target domains using only source domain data (Zhou et al., 2023). The central

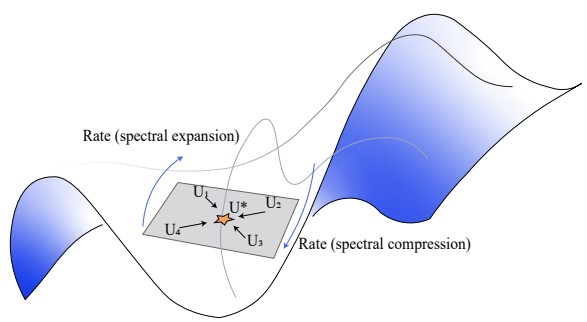

*Figure 1.* **Rate–distortion view of domain generalization.** Domain shift misaligns class-conditional subspaces (curved trajectories); RDI aligns them toward an invariant semantic center $U^*$ on the Grassmannian while regulating spectral complexity via rate control. Curved arrows indicate spectral expansion/compression.

challenge lies in *shaping the geometry* of class-conditional representations across domains, not merely matching their distributions.

Existing DG methods enforce invariance through diverse mechanisms: distribution alignment via moment matching or adversarial training (Sun & Saenko, 2016; Ganin et al., 2016), risk-based constraints (Arjovsky et al., 2019; Krueger et al., 2021), and information-theoretic regularization (Li et al., 2022; Cha et al., 2022). While effective in controlling statistical dependencies, these approaches do not explicitly model the *geometric organization* of class-conditional representations. This gap matters because deep features are widely observed to concentrate on low-dimensional structures whose intrinsic dimension is orders of magnitude smaller than the ambient feature dimension (Ansuini et al., 2019); in such regimes, semantic information is encoded in the relative geometry and spectral composition of class-conditional structures, not merely in marginal distributions. Purely distributional invariance can therefore destabilize discriminative geometry, leading to ill-conditioned alignment or representational collapse.

We address this gap with Geometric **R**ate–**D**istortion **I**nvariance (**RDI**), which operates as follows: for each class, **RDI** extracts a low-dimensional feature subspace within every source domain, aligns these class-conditional subspaces

[1]State Key Laboratory of Biopharmaceutical Preparation and Delivery, Institute of Process Engineering, Chinese Academy of Sciences, Beijing, China [2]University of Chinese Academy of Sciences, Beijing 100049, China. Correspondence to: Sen Liang <liangsen@ipe.ac.cn>, Shuo Bai <baishuo@ipe.ac.cn>.

*Proceedings of the 43rd International Conference on Machine Learning*, Seoul, South Korea. PMLR 306, 2026. Copyright 2026 by the author(s).

across domains on the Grassmann manifold, and simultaneously regularizes their spectral spread. The combined objective is formulated as a rate–distortion trade-off embodying a *dual principle*: *geometric distortion* measures cross-domain subspace misalignment, and the *rate term* controls within-subspace spectral complexity. We identify a fundamental instability addressed by this design: geometric alignment without spectral control becomes ill-conditioned under finite samples, while rate-only objectives collapse discriminative structure. As illustrated in Figure 1, **RDI** formulates DG as optimization on the Grassmann manifold, recovering domain-stable semantic centers without external anchors.

Our theoretical analysis establishes finite-sample guarantees under bounded orthogonal rotations—a canonical probe to isolate geometric misalignment (Section 3.1). Empirically, **RDI** proves effective beyond this idealized setting across diverse benchmarks (Section 4).

Our contributions are summarized as follows:

- **Geometric rate–distortion formulation for DG.** We cast domain generalization as a geometric rate–distortion problem on Grassmann manifolds, realizing a class-wise information bottleneck that compresses nuisance variation while preserving discriminative structure.

- **Spectral rate modeling with Fisher–Gaussian capacity.** We derive a Fisher-diagonal Gaussian approximation for the rate term, yielding a stable capacity-regularized spectral objective with provable upper-bound guarantees.

- **Risk-aware Rényi aggregation.** We introduce a Rényi-entropy–induced escort aggregation for class-wise rate regularization with geometric interpretation.

- **Empirical validation.** RDI achieves competitive performance on DomainBed, with consistent improvements on datasets with pronounced geometric shifts (e.g., PACS).

## 2. Related Work

**Statistical and Geometric Invariance.** DG methods enforce invariance via statistical matching (Sun & Saenko, 2016; Long et al., 2015), domain-adversarial training (Ganin et al., 2016; Li et al., 2018), gradient-based objectives (Arjovsky et al., 2019; Chen et al., 2023b; Rame et al., 2022; Krueger et al., 2021), flat minima (Cha et al., 2021; Li et al., 2025), causality (Lv et al., 2022), sharpness-aware optimization (Wang et al., 2023; Zhang et al., 2023), or training-recipe improvements (Teterwak et al., 2025). Geometric approaches align subspaces on manifolds (Gong et al., 2012; Jin et al., 2022) but typically lack explicit complexity control. ISR (Wang et al., 2022) provably recovers subspaces with invariant characteristics in a causal model but does not regularize spectral complexity.

**Information-Theoretic Methods.** The Information Bottleneck (Tishby et al., 2000; Alemi et al., 2017; Hassanpour et al., 2025; Ahuja et al., 2021; Achille & Soatto, 2018) compresses irrelevant information while preserving task signals. Related perspectives include generalization bounds of information theory (Hellström et al., 2025; Alquier, 2024), information decomposition (Lang et al., 2025), and transductive bounds (Tang & Liu, 2024). MIRO (Cha et al., 2022) achieves strong performance by regularizing mutual information between learned and frozen pre-trained representations, but does not explicitly model geometric structure.

**Representation Geometry and Other Directions.** Riemannian methods model structured representations (Huang & Van Gool, 2017; Ansuini et al., 2019), and rate–distortion principles have been applied to representation learning (Yu et al., 2020; Ding et al., 2022). Adjacent directions outside our scope—vision-language models (Cho et al., 2023; Singha et al., 2024; Bose et al., 2024), test-time adaptation (Chen et al., 2023a), meta-learning (Khoee et al., 2024), and adversarial DG extensions (Sicilia et al., 2023; Pham et al., 2024)—typically require additional modalities, target-domain access, or modified training paradigms.

## 3. Method

The remainder of this section instantiates this dual principle of geometric alignment and spectral control as a rate–distortion objective on the Grassmann manifold.

### 3.1. Preliminaries

**Problem Setting.** We consider standard domain generalization (DG). Let $\mathcal{E}_{\mathrm{tr}}$ be the set of $M$ training domains, each providing labeled samples $\mathcal{D}_e = \{(x_i^e, y_i^e)\}$ from a domain-specific distribution $P_e(X, Y)$. The goal is to learn a predictor that generalizes to unseen domains. Let $f_\theta : \mathcal{X} \to \mathbb{R}^d$ be a feature extractor. For each domain $e$ and class $c$, define the class-conditional feature set

$$\mathcal{F}_{e,c} = \{f_\theta(x) \mid (x,y) \in \mathcal{D}_e, \, y = c\}. \tag{1}$$

Domain shift manifests as variations of $\mathcal{F}_{e,c}$ across domains, even for fixed $c$.

Let $C$ denote the number of classes, $n_c$ the number of samples in class $c$ pooled across all training domains, and $N = \sum_{c=1}^{C} n_c$ the total sample size. We write $\hat{p}_c = n_c/N$ for the empirical class frequency. For theoretical analysis, we write $n$ for the number of samples per domain, assuming balanced domains so that $N = Mn$.

**Generative Model of Domain Shift.** We model class-conditional features as $\mathbf{f}_{e,c} = A_e U_c^* \mathbf{z} + \boldsymbol{\epsilon}_e$, where $U_c^*$ is an invariant ground-truth semantic subspace (the recovery target), $A_e$ is a domain-specific linear transformation, $\mathbf{z} \sim \mathcal{N}(\mathbf{0}, \Lambda_z)$ has covariance shared across domains, and $\boldsymbol{\epsilon}_e \sim \mathcal{N}(\mathbf{0}, \sigma^2 I)$ is isotropic noise. We adopt a zero-mean noise model to isolate rotational geometry as the canonical source of cross-domain misalignment; nonzero domain means $\boldsymbol{\mu}_e$ are a natural extension. Our theoretical analysis focuses on the case where $A_e$ is approximately orthogonal (Assumption 3.5), deferring general affine perturbations to future work.

**Grassmannian Subspace Representation.** Semantic information in deep representations is often concentrated in low-dimensional class-conditional subspaces, as evidenced by intrinsic dimensionality estimates of deep features (Ansuini et al., 2019) and explicit subspace-based representation learning (Gong et al., 2012; Wang et al., 2022). We represent $\mathcal{F}_{e,c}$ by its top-$k$ principal subspace

$$U_{e,c} \in \mathrm{Gr}(k,d), \tag{2}$$

where $\mathrm{Gr}(k,d)$ denotes the Grassmann manifold of $k$-dimensional subspaces in $\mathbb{R}^d$, and $U_{e,c}$ is the top-$k$ eigenspace of the true class-conditional covariance $\mathrm{Cov}(\mathcal{F}_{e,c})$. Its empirical counterpart, estimated from $n$ finite samples, is denoted $\widehat{U}_{e,c}$; the two coincide as $n \to \infty$, with the finite-sample gap analyzed in Section 3.3. Distances on $\mathrm{Gr}(k,d)$ naturally quantify geometric discrepancies across domains.

**Rényi Entropy for Spectral Control.** To control representation complexity, we employ Rényi entropy of order $\alpha > 0$ (Rényi, 1961) (formally defined in Eq. 11): higher $\alpha$ emphasizes top eigenvalues, while $\alpha \to 1$ recovers Shannon entropy. These preliminaries motivate decomposing domain variation into directional misalignment and spectral complexity.

## 3.2. Geometric Rate–Distortion Invariance

### 3.2.1. RDI CENTER

Classical rate–distortion theory seeks a representation $Z$ that balances distortion $d(X, Z)$ and coding rate $I(X; Z)$ (Shannon, 1959):

$$\min_{p(z|x)} \mathbb{E}[d(X, Z)] + \beta I(X; Z). \tag{3}$$

We adapt this principle to domain-invariant representation learning. The *RDI center* of class $c$, denoted by $\mathcal{C}_c^\star$, generalizes the rate–distortion optimum to the multi-domain setting:

$$\mathcal{C}_c^\star \in \arg\min_{\mathcal{F}_c} \sum_e D\big(\mathcal{F}_{e,c}, \mathcal{F}_c\big) + \beta \, \Phi(\mathcal{F}_c), \tag{4}$$

where $D$ is an abstract distortion functional measuring cross-domain discrepancy (instantiated on the Grassmannian as $d_{\mathrm{Gr}}$ in Section 3.2.2), and $\Phi$ is an abstract rate functional quantifying representational complexity (instantiated as the capacity-regularized rate $R_\tau$ in Section 3.2.3). We operationally define *geometric invariance* as the state where cross-domain Grassmannian subspace distances are minimized: $d_{\mathrm{Gr}}(U_{e,c}, U_{e',c}) \to 0$ for all domain pairs $(e, e')$.

The RDI center is a conceptual target, not computed directly. Instead, we design tractable surrogates for $D$ (Section 3.2.2) and $\Phi$ (Section 3.2.3).

**Geometric interpretation.** The RDI formulation decomposes domain variation into two *geometrically orthogonal* components: the distortion term $D$ controls *directional variability* via subspace alignment on the Grassmannian, while the rate term $\Phi$ regulates the *spectral volume* within each subspace—analogous to polar decomposition separating rotation from scaling. Crucially, neither subsumes the other: perfect alignment ($D = 0$) leaves the spectral volume unconstrained, and a bounded $\Phi$ does not ensure alignment.

### 3.2.2. DISTORTION MODELING ON THE GRASSMANN MANIFOLD

Domain shifts are commonly modeled as geometric misalignments between class-conditional subspaces (Gong et al., 2012; Fernando et al., 2013). For each domain $e$ and class $c$, we represent $\mathcal{F}_{e,c}$ by its $k$-dimensional principal subspace $U_{e,c} \in \mathrm{Gr}(k,d)$. We decompose the distortion into two components.

**Geometric primitives.** For two $k$-dimensional subspaces $U, V \in \mathrm{Gr}(k,d)$ with orthonormal bases, the *principal angles* $\theta_1, \ldots, \theta_k \in [0, \pi/2]$ are defined via

$$\cos\theta_i = \sigma_i(U^\top V), \qquad i = 1, \ldots, k, \tag{5}$$

where $\sigma_i$ denotes the $i$-th singular value (Edelman et al., 1998). The *geodesic distance* on $\mathrm{Gr}(k,d)$ is then

$$d_{\mathrm{Gr}}(U, V) = \sqrt{\tfrac{1}{k} \sum_{i=1}^k \theta_i^2}. \tag{6}$$

The *Fréchet mean* of a set of subspaces $\{U_e\}_{e \in \mathcal{E}}$ is defined as

$$\bar{U} \in \arg\min_{U \in \mathrm{Gr}(k,d)} \sum_{e \in \mathcal{E}} d_{\mathrm{Gr}}^2(U, U_e), \tag{7}$$

which exists and is unique under the small-diameter condition of Assumption 3.5 (Afsari, 2011).

**Geometric distortion (cross-domain alignment).**

$$\Delta_{\mathrm{geo}}^{(c)} = \sum_{e \neq e'} d_{\mathrm{Gr}}(U_{e,c}, U_{e',c}), \tag{8}$$

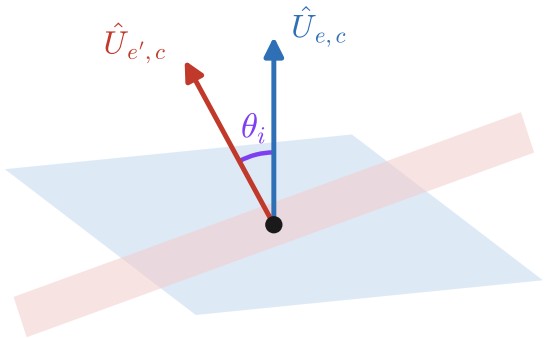

$$\theta_i = \arccos\big(\sigma_i(\hat{U}_{e,c}^\top \hat{U}_{e',c})\big)$$
$$= \textit{rotation to align subspaces}$$

*Figure 2.* Illustration of the Grassmannian distance in Eq. 8. The $i$-th principal angle $\theta_i$ between empirical class-conditional subspaces $\hat{U}_{e,c}$ and $\hat{U}_{e',c}$ captures the rotation required to align them; the Grassmann distance $d_{\mathrm{Gr}}$ aggregates over all $k$ such angles.

where $d_{\mathrm{Gr}}(U, V) = \sqrt{\frac{1}{k}\sum_{i=1}^{k}\theta_i^2}$ is the geodesic distance on the Grassmann manifold, with $\theta_i = \arccos(\sigma_i(U^\top V))$ being the principal angles; Figure 2 illustrates this geometrically.

**Projection distortion (non-degeneracy).** To prevent degenerate solutions where estimated subspaces decouple from the data distribution, we introduce a projection distortion term—also referred to as the reconstruction error in subspace learning literature (Tipping & Bishop, 1999; Yu et al., 2020):

$$\Delta_{\mathrm{rec}}^{(c)} := \mathbb{E}_{f\sim\mathcal{F}_c}\left\|f - \Pi_{\bar{U}_c}f\right\|_2^2, \qquad (9)$$

where $\bar{U}_c$ approximates the RDI center $\mathcal{C}_c^\star$ via the Fréchet mean of domain-specific subspaces. The total distortion aggregates geometric and projection terms:

$$\Delta(\mathcal{F}_c) = \lambda_{\mathrm{geo}}\,\Delta_{\mathrm{geo}}^{(c)} + \lambda_{\mathrm{rec}}\,\Delta_{\mathrm{rec}}^{(c)}. \qquad (10)$$

*Remark* 3.1 (Role of projection distortion). The projection term $\Delta_{\mathrm{rec}}^{(c)}$ serves as a regularizer to ground subspaces in actual feature statistics, not as a quantity to be minimized per se. The optimization may trade projection fidelity for tighter cross-domain alignment when beneficial—particularly in high-class-diversity settings, where *tolerating a small amount of within-class projection error* facilitates geometric consistency without sacrificing discriminability.

*Remark* 3.2 (On terminology). We use *projection distortion* to refer to the RDI loss term $\Delta_{\mathrm{rec}}^{(c)}$ (Eq. 9) and *reconstruction error* for the classical PCA-derived quantity $\mathbb{E}_f\|f - \Pi_U f\|_2^2$ used in subsequent analysis; Proposition 3.3 relates the two via Grassmannian geometry.

**Proposition 3.3** (Grassmannian distance as semantic distortion surrogate). *In the generative model with orthogonal domain transformations ($A_e \in \mathcal{O}(d)$) and shared $\Lambda_z$: (i) $U_{e,c}$ minimizes the expected reconstruction error over $\mathrm{Gr}(k,d)$; (ii) the excess reconstruction error from projecting features in $\mathcal{F}_{e,c}$ onto the misaligned subspace $U_{e',c}$, $\Delta\mathcal{E} := \mathbb{E}_{f\sim\mathcal{F}_{e,c}}\big[\|f - \Pi_{U_{e',c}}f\|_2^2 - \|f - \Pi_{U_{e,c}}f\|_2^2\big]$, satisfies $\lambda_k \cdot d_{\mathrm{Gr}}^2(U_{e,c}, U_{e',c}) \le \Delta\mathcal{E} \le \lambda_1 \cdot d_{\mathrm{Gr}}^2(U_{e,c}, U_{e',c})$, establishing that the Grassmann distance limits projection distortion up to signal eigenvalue scaling. Proof in Appendix A.6.*

*Remark* 3.4 (On the noise term). In the generative model with isotropic noise $\epsilon_e$ satisfying $\mathrm{Cov}(\epsilon_e) = \sigma^2 I$, the noise contribution to the reconstruction error is $\sigma^2(d-k)$, which is *invariant* to the choice of a subspace of dimensions $k$. Thus the noise term cancels in the cross-domain error difference $\Delta\mathcal{E}$, and the bound depends only on signal eigenvalues.

**Choice of subspace dimension $k$.** The parameter $k$ defines the dimensionality of the Grassmann manifold $\mathrm{Gr}(k,d)$ on which RDI performs cross-domain alignment, and the size of the class-conditional principal subspace representing each $\mathcal{F}_{e,c}$. It plays a different role from the rate term: $k$ sets the geometric stage for alignment (via $\Delta_{\mathrm{geo}}$, Eq. 8), while the rate term (Section 3.2.3) regulates within-class spectral complexity. Hence $k$ should accommodate the principal directions of class-discriminative variation, while compression is handled independently by the rate term.

Since $k$ also affects per-step SVD and buffer cost, and spectral measurements (Appendix D.5.2) confirm that moderate $k$ already controls effective subspace dimensions across all DomainBed benchmarks, we fix $k \in \{32, 64\}$ as a dataset-aware default rather than performing fine-grained tuning. CKA analysis (Appendix D.5.1) confirms this design: the rate term, not $k$, adapts to each benchmark's structure, preserving feature diversity on few-class benchmarks (e.g., PACS) and producing more concentrated representations on many-class benchmarks (e.g., OfficeHome). Full sensitivity to $k$ is in Appendix C.3.

### 3.2.3. RATE MODELING VIA RÉNYI ENTROPY

The rate term $R(\mathcal{F}_c)$ characterizes the residual *spectral volume* within each aligned subspace. We adopt Rényi entropy of order $\alpha > 0$ (Rényi, 1961):

$$H_\alpha(\mathbf{z}_c) = \frac{1}{1-\alpha}\log\int p(\mathbf{z}_c)^\alpha\, d\mathbf{z}_c, \qquad (11)$$

which reduces to Shannon entropy as $\alpha \to 1$.

**Capacity-regularized form.** Under Gaussian approximation $\mathbf{z}_c \sim \mathcal{N}(\mu_c, \Sigma_c)$, minimizing Rényi entropy is equivalent to minimizing $\log\det\Sigma_c$ (Appendix A.7). For numeri-

cal stability, we employ:

$$R_\tau(\mathcal{F}_c) = \frac{1}{d} \sum_{i=1}^{d} \log(1 + \tau\,\sigma_{c,i}^2 + \varepsilon), \qquad (12)$$

where $\sigma_{c,i}^2$ are marginal variances and $\tau > 0$ controls the rate–distortion operating point. This provides a tractable upper bound on $\log\det\Sigma_c$ via Hadamard's inequality, following standard practice in rate–distortion literature (Alemi et al., 2017; Yu et al., 2020). The form $\frac{1}{2}\log(1 + \tau\sigma^2)$ is the capacity of a scalar Gaussian channel with input variance $\sigma^2$ and noise level $1/\tau$; $R_\tau$ is thus an average per-coordinate channel capacity, and $\tau$ plays the role of the signal-to-noise operating point. We fix $\tau = 1$ throughout this paper, which corresponds to unit SNR—the natural scale of batch-normalized features. The parameter $\tau$ is retained in (12) to expose the rate–distortion structure; sensitivity analysis (Appendix C.3) confirms $\tau = 1$ is a robust choice.

**Task-adaptive behavior.** The gradient $\partial R_\tau/\partial\sigma_{c,i}^2 \propto (1 + \tau\sigma_{c,i}^2)^{-1}$ is bounded and saturates at large variances, so the rate cost of *slightly* expanding a near-collapsed direction is small in absolute terms; meanwhile, restoring a degenerate direction yields a disproportionate gain in geometric alignment, lowering the total objective. Conversely, dispersed directions incur a steady rate penalty and are gently compressed. We formalize this asymmetry and verify its predictions in Appendix D.5.2.

**Rényi risk aggregation.** The coefficient $\alpha$ reappears as the order of a Rényi risk functional for cross-class aggregation. Let $R_c = R_\tau(\mathcal{F}_c)$ and $\tilde{p}_c^{(\alpha)} = \hat{p}_c^\alpha / \sum_j \hat{p}_j^\alpha$ (escort distribution (Tsallis, 1988)):

$$\Psi_\alpha = \frac{1}{\alpha - 1} \log \sum_{c=1}^{C} \tilde{p}_c^{(\alpha)} \exp\big((\alpha - 1)R_c\big), \qquad (13)$$

interpolating between expected rate ($\alpha \to 1$) and worst-class rate ($\alpha \to \infty$).

**Practical implementation.** For optimization stability, we use:

$$\bar{R}(\mathcal{F}) = \frac{\sum_{c=1}^{C} n_c^{1-\alpha} R_c}{\sum_{c=1}^{C} n_c^{1-\alpha}}, \qquad (14)$$

recovering arithmetic mean at $\alpha = 1$ and inverse-frequency weighting at $\alpha = 2$. Here, $\alpha$ controls the escort geometry of class aggregation, while $\tau$ governs intra-class spectral capacity—together forming a two-level Rényi–Fisher geometric regularization.

### 3.2.4. FINAL RDI OBJECTIVE

Combining distortion and rate:

$$\mathcal{L}_{\mathrm{RDI}}(\theta) = \widehat{\mathcal{R}}_{\mathrm{src}}(\theta) + \gamma \left[ \sum_c \Delta(\mathcal{F}_c) + \beta\,\bar{R}(\mathcal{F}) \right], \quad (15)$$

where $\widehat{\mathcal{R}}_{\mathrm{src}}$ is empirical classification risk. Hyperparameters: $\gamma$ (RDI strength), $\lambda_{\mathrm{geo}}$, $\lambda_{\mathrm{rec}}$ (alignment vs. projection weights), $\beta$ (compression), $\alpha$ (risk aggregation). The capacity parameter $\tau$ in (12) is fixed to 1 by design (see Section 3.2.3).

### 3.3. Theoretical Analysis

**Scope of theoretical guarantees.** We analyze RDI under the bounded orthogonal rotation model (Assumption 3.5) as a canonical tractable regime—rotations fully parametrize subspace misalignment via principal angles, preserve eigenvalue structure for Davis–Kahan analysis, and admit clean Fréchet mean guarantees on the Grassmannian. This is an analytical device rather than a method prerequisite: the underlying rate–distortion mechanism extends beyond rotations (Appendix A.4), and our empirical results on benchmarks with severe non-rotational shifts (TerraIncognita, DomainNet) validate this generality.

**Theoretical strategy.** Building on this bounded rotation model, we derive three theoretical results: recovery of domain-invariant semantic subspaces, a generalization bound, and the necessity of rate control. The key insight—balancing geometric alignment and spectral complexity—extends beyond rotations (Appendix D.7). Since $R_\tau$ (Eq. 12) is monotonic in spectral measures, all guarantees hold up to rescaling.

**Assumption 3.5** (Geometric Perturbation Model: Rotation Instantiation). Let $\mathcal{E} = \mathcal{E}_{\mathrm{tr}}$ be the set of $M$ training domains. The domain-specific orthogonal transformations $\{Q_e\}_{e\in\mathcal{E}}$ satisfy $\max_{e,e'} \|Q_e - Q_{e'}\|_F \leq \delta_{\mathrm{rot}}$ with $\delta_{\mathrm{rot}} < 1/2$, ensuring a unique Fréchet mean on $\mathrm{Gr}(k, d)$ (Afsari, 2011). Additionally, the smallest signal eigenvalue satisfies $\lambda_k(\Lambda_z) \geq c_0 > 0$ (eigengap condition), ensuring signal subspace identifiability.

**Assumption 3.6** (Feature Concentration on Subspaces). For each domain $e$ and class $c$, there exists $\delta_{\mathrm{res}} > 0$ such that $\mathbb{E}_{f\sim\mathcal{F}_{e,c}}[\|f - \Pi_{U_{e,c}}f\|_2^2] \leq \delta_{\mathrm{res}}^2$. This assumption is standard in subspace-based representation learning and has been empirically observed in deep networks with pretrained initialization.

**Assumption 3.7** (Optimization Regularity). The optimizer reaches a stationary point with geometric distortion at most $O(\epsilon_{\mathrm{opt}})$ above its minimum, where $\epsilon_{\mathrm{opt}} = O(1/\sqrt{n})$.

**Assumption 3.8** (Spectral Regularity and Bounded Features). The class-conditional covariances satisfy $\kappa(\Sigma_c) \leq$

$K$, and features are bounded: $\|f_\theta(x)\|_2 \le B$.

*Remark* 3.9 (On the assumptions). We comment on the three assumptions above. (i) **Frechet mean uniqueness:** the bound $\delta_{\text{rot}} < 1/2$ guarantees that the population subspaces $\{Q_e U_c^*\}_{e \in \mathcal{E}}$ lie within a geodesic ball of radius $< \pi/4$ on $\mathrm{Gr}(k, d)$, which is sufficient for their Fréchet mean $\bar{U}_c$ to exist and be unique (Afsari, 2011). (ii) **Subspace estimation rate:** the eigengap $c_0$ and per-domain sample size $n$ interact via the Davis–Kahan theorem (Stewart & Sun, 1990): with high probability, the empirical subspace $\widehat{U}_{e,c}$ lies within $O(\sqrt{d/n}/c_0)$ of its population counterpart $Q_e U_c^*$ on $\mathrm{Gr}(k, d)$. The dependence on $d$ is tightened in practice because the rate term $\bar{R}$ reduces the operative dimensionality from $d$ to an effective quantity $d_{\text{eff}} \ll d$ (Corollary 3.13), making the $O(1/\sqrt{n})$ scaling achievable despite the high feature dimension (e.g., $d = 2048$ for the ResNet-50 backbone used in our experiments). (iii) **Recovery interpretation:** when domain rotations are approximately centered, $\bar{U}_c \approx U_c^*$ in Theorem 3.10; otherwise $\bar{U}_c$ may differ from $U_c^*$ by a global rotation, which is immaterial for classification as it can be absorbed by the linear classifier. See Appendix A.3 for full technical details.

**From assumptions to recovery guarantees.** With these assumptions in place, we establish that minimizing the RDI objective ensures both inter-domain subspace alignment and recovery of an invariant semantic anchor.

**Theorem 3.10** (Optimality and Invariance of the RDI Center). *Under Assumptions 3.5–3.7, assume class-conditional features are generated as $\mathbf{f}_{e,c} = Q_e U_c^* \mathbf{z} + \boldsymbol{\epsilon}_e$, where $U_c^* \in \mathrm{Gr}(k, d)$ is an invariant semantic subspace, $Q_e \in \mathcal{O}(d)$ are domain-specific orthogonal transformations, $\mathbf{z}$ is subGaussian with zero mean, and $\boldsymbol{\epsilon}_e$ is isotropic noise. Let $\{\widehat{U}_{e,c}\}$ be the empirical subspaces obtained from a minimizer of Eq. 15 with $n$ samples per domain. Then, with probability at least $1 - \delta$, for all classes $c$ and domain pairs $e, e'$:*

1. *(**Alignment**):*

$$d_{\mathrm{Gr}}(\widehat{U}_{e,c}, \widehat{U}_{e',c}) = \mathcal{O}\left(\sqrt{d/n}/c_0 + \epsilon_{\text{opt}}\right);$$

2. *(**Recovery**):*

$$d_{\mathrm{Gr}}(\widehat{U}_{e,c}, \bar{U}_c) = \mathcal{O}\left(\sqrt{d/n}/c_0 + \epsilon_{\text{opt}}\right),$$

*where $\bar{U}_c$ is the Fréchet mean of the population subspaces $\{Q_e U_c^*\}_{e \in \mathcal{E}}$.*

*When $d, c_0$ are treated as constants, both rates simplify to $\mathcal{O}(1/\sqrt{n})$. Proof in Appendix A.10.*

**From subspace recovery to generalization.** Theorem 3.10 ensures that empirical subspaces concentrate around their cross-domain centers at rate $\mathcal{O}(1/\sqrt{n})$. Combined with explicit rate control, this enables a finite-sample generalization bound via information-theoretic analysis. We adopt feature normalization (batch normalization in our projection head $g$, which feeds the rate term), keeping per-coordinate variances at $O(1)$ scale; this places the empirical rate $\widehat{\bar{R}}$ in the bounded regime assumed below.

**Theorem 3.11** (Generalization Bound via Information-Theoretic Analysis). *Let $\hat{\theta}$ minimize Eq. 15 over $M$ source domains with $n$ samples each (total $N = Mn$). Assume the loss $\ell$ is $L$-Lipschitz and bounded, Assumptions 3.5–3.8 hold, and the target domain $t$ satisfies $\max_{e,c} d_{\mathrm{Gr}}(U_{t,c}, U_{e,c}) \le \rho$. Then, with probability at least $1 - \delta$,*

$$\mathcal{R}_t(\hat{\theta}) \le \widehat{\mathcal{R}}_{\text{src}}(\hat{\theta}) + L\sqrt{\lambda_{\max}(\Lambda_z) \cdot \left(\widehat{\Delta}_{\text{geo}}(\hat{\theta}) + \rho\right)}$$

$$+ 2L\delta_{\text{res}} + B_\ell \sqrt{\frac{C_1 \cdot \widehat{\bar{R}}(\hat{\theta}) + C_2 + \log(2/\delta)}{Mn}},$$

*where $C_1 = 2d(1 + \log(1 + K)/\log(1 + \tau))$ and $C_2 = d \log(2\pi e/\tau) + 2\log|\mathcal{C}|$. Full statement with explicit constants in Appendix A.11.*

*Remark* 3.12 (Key features and practical interpretation). The bound exhibits: (i) empirical source risk $\widehat{\mathcal{R}}_{\text{src}}$; (ii) geometric discrepancy $\widehat{\Delta}_{\text{geo}} + \rho + \delta_{\text{res}}$; and (iii) statistical complexity $\sqrt{C_1 \widehat{\bar{R}}/n}$ where the rate term enters *linearly* — a key feature of the information-theoretic (MI-based) analysis. Under batch normalization, $\widehat{\bar{R}} = O(1)$, making $O(1/\sqrt{n})$ convergence practically achievable despite large $d$.

**Corollary 3.13** (Effective dimension via subspace projection). *A complementary Rademacher complexity argument, exploiting that RDI projects features onto a $k$-dimensional subspace (Appendix A.11, Step 3b), yields a complexity term scaling as $O(\sqrt{\exp(\widehat{\bar{R}})/n})$. Under batch normalization, $\widehat{\bar{R}} = O(1)$, this simplifies to $O(\sqrt{d_{\text{eff}}/n})$, where $d_{\text{eff}} \ll d$ is the operative dimension controlled by the rate term. Empirically, the rate term concentrates the feature spectrum onto a low-dimensional subspace: e.g., the stable rank of ResNet-50's top-32 subspace is 3–5 on PACS and OfficeHome, consistent with prior observations on intrinsic dimensionality in deep networks (Ansuini et al., 2019). The two proof routes are complementary: Theorem 3.11 (MI-based) gives tight $\widehat{\bar{R}}$-dependence (linear), while this Rademacher analysis exposes a favorable $d$-dependence in the bounded-rate regime.*

**Necessity of aggregated rate control.** Finally, we show that geometric alignment alone is insufficient for generalization.

**Proposition 3.14** (Necessity of Aggregated Rate Control). *Consider the ablated objective obtained by removing the aggregated rate term $\bar{R}(\mathcal{F})$ from Eq. 15. Even under norm-bounded features ($\|f\|_2 \leq B$ uniformly), there exists a sequence of feature distributions $\{\mathcal{F}^{(m)}\}_{m \geq 1}$ satisfying simultaneously: (i) perfect geometric alignment $\Delta_{\mathrm{geo}} = 0$; (ii) uniform norm bound $\sup_m \|f\|_2 \leq B$ for all $f \in \mathcal{F}^{(m)}$; yet (iii) $\kappa(\Sigma_c^{(m)}) \to \infty$ as $m \to \infty$. Consequently, standard norm-based regularization (e.g., weight decay) cannot prevent spectral degeneration; explicit rate control is necessary. Proof in Appendix A.12.*

# 4. Experiments

We empirically validate both facets of RDI's dual principle of geometric alignment and spectral control across five DomainBed benchmarks.

**Experimental Setup**

**Datasets and Protocol.** We evaluate on the **DomainBed** benchmark (Gulrajani & Lopez-Paz, 2021), comprising five datasets: PACS (Li et al., 2017) (4 domains, 7 classes), VLCS (Fang et al., 2013) (4 domains, 5 classes), Office-Home (Venkateswara et al., 2017) (4 domains, 65 classes), TerraIncognita (Beery et al., 2018) (4 domains, 10 classes), and DomainNet (Peng et al., 2019) (6 domains, 345 classes). We follow the *training-domain validation* protocol with 20 train-validation splits, each repeated 3 times.

**Implementation.** We use ResNet-50 (He et al., 2016) pretrained on ImageNet, the AdamW optimizer (Loshchilov & Hutter, 2019), and the standard DomainBed training schedule (5K iterations; 15K for DomainNet). Hyperparameters are selected via grid search using training-domain validation only. Additional details are provided in Appendix C.

## 4.1. Proof-of-Concept on Synthetic Data

To validate the geometric and information-theoretic motivations of RDI under controlled conditions, we construct a synthetic multi-domain subspace benchmark. Each domain is generated from a rotated $k$-dimensional subspace embedded in a $D$-dimensional ambient space, enabling explicit control over geometric discrepancies while preserving shared latent structure. Full details are provided in Appendix B.1.

**Geometric Behavior on the Grassmann Manifold.** Figure 3(a) illustrates three domain-specific subspaces $U_1, U_2, U_3$ on $\mathrm{Gr}(k, D)$ along with solutions under different ablation settings. Removing the geometric alignment term causes the estimate to bias toward the domain with larger energy (higher variance), as there is no cross-domain geometric coupling. Removing the fidelity term (data-fitting component) yields a degenerate solution that can reduce

---

**Algorithm 1** Training Procedure of RDI

**Require:** Training domains $\{\mathcal{D}_e\}$, backbone $f_\theta$, projection head $g_\phi$, classifier $h$

**Require:** Hyperparameters $\lambda_{\mathrm{geo}}, \lambda_{\mathrm{rec}}, \alpha$ (other parameters fixed at defaults, Table 4)

1: Initialize parameters $\theta, \phi$, subspaces $\{\bar{U}_c, \widehat{U}_{e,c}\}$
2: **for** $t = 1, 2, \ldots, T_{\max}$ **do**
3:     Sample batch $\mathfrak{B}$ from all domains
4:     Extract projected features: $Z = g_\phi(f_\theta(\mathfrak{B}))$
5:     Update subspaces $\{\bar{U}_c, \widehat{U}_{e,c}\}$ via EMA statistics on $Z$
6:     Compute classification loss $\mathcal{L}_{\mathrm{CE}} = \ell(h(Z), Y)$
7:     **if** $t > T_w$ **then**
8:         $\mathcal{L}_{\mathrm{geo}} = \sum_{c, e \neq e'} d_{\mathrm{Gr}}^2(\widehat{U}_{e,c}, \widehat{U}_{e',c})$ {Geometric (Eq. 8)}
9:         $\mathcal{L}_{\mathrm{rec}} = \sum_c \|Z_c - \Pi_{\bar{U}_c} Z_c\|_2^2$ {Reconstruction (Eq. 9)}
10:        $R_c = \frac{1}{d} \sum_i \log(1 + \tau \sigma_{c,i}^2 + \varepsilon)$ for each $c$ {Per-class rate (Eq. 12)}
11:        $\mathcal{L}_R = \sum_c n_c^{1-\alpha} R_c / \sum_c n_c^{1-\alpha}$ {Aggregated rate (Eq. 14)}
12:        $\mathcal{L} = \mathcal{L}_{\mathrm{CE}} + \gamma(\lambda_{\mathrm{geo}}\mathcal{L}_{\mathrm{geo}} + \lambda_{\mathrm{rec}}\mathcal{L}_{\mathrm{rec}} + \beta\mathcal{L}_R)$
13:     **else**
14:         $\mathcal{L} = \mathcal{L}_{\mathrm{CE}}$
15:     **end if**
16:     $(\theta, \phi) \leftarrow (\theta, \phi) - \eta\nabla\mathcal{L}$
17: **end for**
18: **Output:** Trained parameters $\theta, \phi$

---

geometric mismatch but no longer explains the generated observations. The full RDI objective recovers a subspace closest to the ground truth $U^*$, confirming that alignment and fidelity provide complementary constraints. Extended analyses are provided in Appendix B.2 and B.3.

**Rényi Aggregation as $\alpha$-Geometric Reweighting.** Figure 3(b) illustrates how the Rényi order $\alpha$ governs class aggregation through $\alpha$-geometric weighting induced by the escort distribution $\tilde{p}_c^{(\alpha)} \propto n_c^{1-\alpha}$. This defines an $\alpha$-geometry on the statistical manifold, interpolating between mixture ($\alpha \to 0$) and exponential ($\alpha \to \infty$) geometries. The aggregated risk $\Psi_\alpha$ implements a geometrically-aware reweighting: $\alpha < 1$ emphasizes the consensus-seeking mixture barycenter, whereas $\alpha > 1$ concentrates regularization on high-risk classes with fragile geometric structure. Thus, $\alpha$-geometry unifies risk-sensitive aggregation with the Riemannian structure of the representation manifold.

**Rate–Distortion Trade-off.** Figure 3(c) depicts the rate–distortion plane induced by our geometric objectives. Omitting the rate regularizer permits unconstrained spectral capacity, which can fit cross-domain alignment (low distortion)

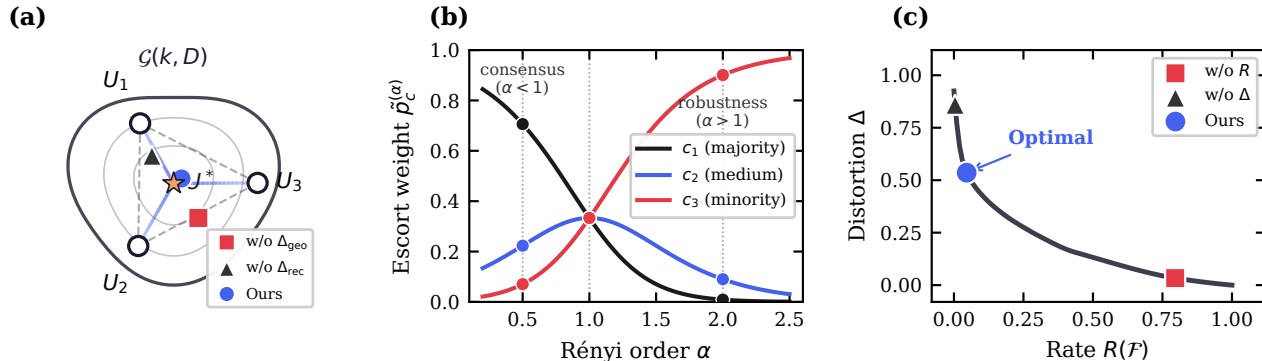

*Figure 3.* **Synthetic experiment.** **(a)** Geometric behavior on $\mathrm{Gr}(k, D)$: RDI recovers a subspace closest to ground truth $U^*$; ablations cause systematic deviations. **(b)** $\alpha$-geometric aggregation: the Rényi order $\alpha$ induces escort reweighting over classes, interpolating between consensus-driven ($\alpha < 1$) and robustness-driven ($\alpha > 1$) spectral regularization regimes. **(c)** Rate–distortion trade-off: full RDI achieves low distortion with moderate rate; ablations degrade either complexity or alignment.

but yields inflated complexity and unstable spectra. Omitting the distortion term enforces spectral control without geometric agreement, leading to compact but misaligned subspaces (high distortion). The full RDI objective attains an operating point near the efficient frontier, balancing representational capacity against cross-domain geometric alignment.

Overall, this experiment verifies that RDI instantiates a geometric form of rate–distortion trade-off, where invariant representations emerge from coupled control of subspace misalignment (distortion) and spectral capacity (rate).

### 4.2. Algorithmic Formulation and Training Procedure

**Geometric projection head.** Our theoretical formulation is stated on a representation space $z$, where geometric alignment and spectral regularization are enforced. To provide a controllable manifold for this regularization, we introduce a lightweight geometric projection head $g(\cdot)$ that maps backbone features into a compact space $z = g(f(x))$, avoiding rank-deficient covariance estimation in high dimensions. Implementation details and motivation are in Appendix C.

**Training procedure.** Algorithm 1 outlines RDI training. We maintain per-domain-class subspaces $\{\widehat{U}_{e,c}\}$ via exponential moving averages (EMA) of feature statistics, computing the top-$k$ eigenvectors periodically. The geometric distortion $\mathcal{L}_{\mathrm{geo}}$ uses geodesic distance on the Grassmannian; the projection loss $\mathcal{L}_{\mathrm{rec}}$ penalizes deviations of features from their estimated subspaces; and the rate term $\mathcal{L}_R$ combines capacity-regularized per-class rates via Rényi-weighted aggregation. A warm-up phase ($t \leq T_w$) with only classification loss precedes RDI constraints for optimization stability. RDI produces a fixed model generalizing directly to unseen domains without test-time adaptation.

### 4.3. Domain Generalization on DomainBed Benchmarks

All results use *train-domain validation* protocol where model selection uses only training domains, ensuring no target-domain information leakage.

**Overall performance.** Table 1 compares RDI with representative domain generalization (DG) methods under comparable training protocols. Following standard DomainBed practice, we compare against objective-level DG methods with comparable training pipelines. We explicitly exclude: (i) *flat-minima optimization methods* (e.g., SWAD (Cha et al., 2021), SFT (Li et al., 2025)), which seek loss-landscape regions robust to perturbations via dense weight averaging or loss-surface refinement, rather than introducing a new DG objective; (ii) *frozen-anchor methods* (e.g., MIRO (Cha et al., 2022)), which leverage frozen pretrained representations as external anchors, in contrast to RDI's learning of invariant structure from inter-domain geometry; (iii) *training-recipe methods* (e.g., ERM++ (Teterwak et al., 2025)), which improve ERM through parameter averaging, warm-starting, augmentation, and LR scheduling rather than a new DG algorithm. All three are orthogonal to and composable with RDI's geometric objective. RDI achieves **65.7%** average accuracy, the best performance among invariance-based and geometric alignment methods that do not rely on external anchors or test-time adaptation. Notably, RDI outperforms both classical invariance methods (IRM, VREx, GroupDRO) and recent geometric approaches (Fishr, CORAL, SagNet) that align gradient or moment statistics, rather than class-conditional subspace geometry.

**Dataset-wise analysis.** RDI achieves the best result on **PACS** (**87.4%**, +1.0% over GGA, with the largest gain on the challenging Sketch domain), **OfficeHome** (**69.9%**,

*Table 1.* **Accuracy (%) on DomainBed under training-domain validation (ResNet-50).** [†]: results from DomainBed. **Bold**: best; underline: second-best. Std approximated via error propagation from per-domain values (Appendix Tables 19–23).

| Method | PACS | VLCS | OfficeHome | TerraInc | DomainNet | Avg. |
|---|---|---|---|---|---|---|
| ERM | $85.5_{\pm 0.3}$ | $77.5_{\pm 0.4}$ | $66.5_{\pm 0.2}$ | $46.1_{\pm 1.6}$ | $40.9_{\pm 0.1}$ | 63.3 |
| IRM (Arjovsky et al., 2019)[†] | $83.5_{\pm 0.5}$ | $78.6_{\pm 0.4}$ | $64.3_{\pm 1.2}$ | $47.6_{\pm 0.8}$ | $33.9_{\pm 1.3}$ | 61.6 |
| VREx (Krueger et al., 2021)[†] | $84.9_{\pm 0.6}$ | $78.3_{\pm 0.5}$ | $66.4_{\pm 0.3}$ | $46.4_{\pm 1.4}$ | $33.6_{\pm 1.4}$ | 61.9 |
| GroupDRO (Sagawa et al., 2020)[†] | $84.4_{\pm 0.6}$ | $76.7_{\pm 0.4}$ | $66.0_{\pm 0.4}$ | $43.2_{\pm 0.8}$ | $33.3_{\pm 0.2}$ | 60.7 |
| DANN (Ganin et al., 2016)[†] | $83.7_{\pm 0.6}$ | $78.6_{\pm 0.4}$ | $65.9_{\pm 0.4}$ | $46.7_{\pm 1.0}$ | $38.3_{\pm 0.2}$ | 62.6 |
| CORAL (Sun & Saenko, 2016)[†] | $86.2_{\pm 0.4}$ | $\mathbf{78.8}_{\pm 0.4}$ | $68.7_{\pm 0.2}$ | $47.7_{\pm 1.0}$ | $41.5_{\pm 0.1}$ | 64.6 |
| Fishr (Rame et al., 2022) | $85.5_{\pm 0.5}$ | $77.8_{\pm 0.2}$ | $67.8_{\pm 0.2}$ | $47.4_{\pm 1.2}$ | $41.7_{\pm 0.1}$ | 64.0 |
| RSC (Huang et al., 2020)[†] | $85.2_{\pm 0.6}$ | $77.1_{\pm 0.4}$ | $65.5_{\pm 0.6}$ | $46.6_{\pm 0.8}$ | $38.9_{\pm 0.3}$ | 62.7 |
| SagNet (Nam et al., 2021)[†] | $86.3_{\pm 0.3}$ | $77.8_{\pm 0.4}$ | $68.1_{\pm 0.2}$ | $\underline{48.6}_{\pm 1.0}$ | $40.3_{\pm 0.1}$ | 64.2 |
| Mixup (Yan et al., 2020)[†] | $84.6_{\pm 0.5}$ | $77.4_{\pm 0.4}$ | $68.1_{\pm 0.3}$ | $47.9_{\pm 0.7}$ | $39.2_{\pm 0.2}$ | 63.4 |
| GGA (Ballas & Diou, 2025) | $\underline{86.4}_{\pm 0.9}$ | $\underline{78.7}_{\pm 0.6}$ | $67.0_{\pm 0.4}$ | $48.5_{\pm 1.1}$ | $\mathbf{44.4}_{\pm 0.1}$ | 65.0 |
| LFME (Chen et al., 2024) | $85.0_{\pm 0.5}$ | $78.4_{\pm 0.2}$ | $\underline{69.1}_{\pm 0.3}$ | $48.3_{\pm 0.9}$ | $42.1_{\pm 0.1}$ | 64.6 |
| RDI (Ours) | $\mathbf{87.4}_{\pm 0.5}$ | $78.2_{\pm 0.7}$ | $\mathbf{69.9}_{\pm 0.2}$ | $\mathbf{48.7}_{\pm 1.1}$ | $\underline{44.2}_{\pm 0.2}$ | $\mathbf{65.7}$ |

*Table 2.* **Ablation on PACS.** Std from 3 seeds per environment. ERM+$g$ exhibits high variance (up to 3.9% on Sketch); RDI losses reduce variance 3–4× (Prop. 3.14). All variants use default hyperparameters; Table 1 reports tuned results.

| Method | A | C | P | S | Avg. |
|---|---|---|---|---|---|
| ERM | $80.6_{\pm 0.4}$ | $72.3_{\pm 0.6}$ | $97.1_{\pm 0.3}$ | $70.0_{\pm 1.0}$ | 80.0 |
| ERM+$g$ | $84.3_{\pm 2.9}$ | $74.5_{\pm 1.8}$ | $97.2_{\pm 0.5}$ | $74.4_{\pm 3.9}$ | 82.6 |
| w/o Dist. | $86.9_{\pm 0.3}$ | $79.5_{\pm 0.4}$ | $\mathbf{97.9}_{\pm 0.4}$ | $81.4_{\pm 0.9}$ | 86.4 |
| w/o Rate | $86.0_{\pm 0.3}$ | $81.3_{\pm 1.7}$ | $97.8_{\pm 0.4}$ | $80.8_{\pm 1.8}$ | 86.5 |
| RDI | $\mathbf{88.4}_{\pm 1.0}$ | $\mathbf{81.5}_{\pm 1.0}$ | $97.4_{\pm 0.5}$ | $\mathbf{82.3}_{\pm 1.1}$ | $\mathbf{87.4}$ |

demonstrating spectral control under high intra-class diversity with 65 classes), and **TerraIncognita** (**48.7%**, despite severe class imbalance and camera-trap artifacts). On **DomainNet** (345 classes), RDI reaches **44.2%**, competitive with GGA and substantially above classical baselines. Architecture-agnostic results (DeiT-Small, ConvNeXt-Tiny) and robustness under low-data / class imbalance are in Appendices D.1, D.3, D.4.

### 4.4. Ablation Studies

Table 2 validates the necessity of each RDI component on PACS.

**Is the gain from the geometric projection head?** To isolate the contribution of the geometric projection head $g(\cdot)$ from our rate-distortion objectives, we include an **ERM + $g(\cdot)$** baseline that uses the identical projection architecture but disables both distortion and rate losses. The gap between ERM + $g(\cdot)$ (82.6%) and RDI (87.4%) demonstrates that the performance gain *cannot* be attributed to the geometric projection head alone; the rate-distortion objectives are essential. We further verify this on four representative baselines (ERM, CORAL, VREx, IB_ERM) with and without identical projection heads in Appendix D.2, where $\Delta$ ranges from $-2.16\%$ to $+0.69\%$, confirming that the projection head provides no consistent boost across methods.

**Component analysis.** Removing the distortion term ($\Delta_{\text{geo}}$) degrades average accuracy by 1.0%, with the largest drop on Art ($-1.5\%$); removing the rate term ($\bar{R}$) degrades by 1.1%, with the largest drop on Sketch ($-1.5\%$), supporting Proposition 3.14. The modest individual drops ($\sim 1\%$) relative to the full gain over ERM + $g$ (4.8%) reflect a shared geometric regularization substrate: each term partially compensates when the other is ablated, yet the full objective consistently outperforms both single-term variants. Hyperparameter sensitivity is analyzed in Appendix C.3.

### 4.5. Analysis of Optimization Dynamics

We empirically verify that RDI's joint objective consistently drives representations toward the lower-left region of the rate–distortion plane across all DomainBed benchmarks. Despite dataset-specific operating regimes (e.g., PACS converges to lower complexity than OfficeHome), all trajectories share this common convergence pattern. Full trajectories and Pareto-front interpretation appear in Appendix D.5.3.

## 5. Conclusion

We revisited DG from a geometric rate–distortion perspective and introduced RDI, a principled framework that extends rate–distortion theory to Grassmann manifolds with a joint objective for subspace alignment and spectral/volumetric control. We proved finite-sample stability under bounded shifts and empirically validated RDI on DomainBed, where ablations confirm the necessity of optimizing both terms.

**Take-home.** Effective DG requires *shaping the geometry* of class-conditional subspaces—aligning them while controlling their spectral complexity. RDI realizes this dual principle through a single rate–distortion objective, bridging information geometry, representation learning, and out-of-distribution generalization.

## Acknowledgements

This work was supported by the Strategic Priority Research Program of the Chinese Academy of Sciences (Grant No. XDB0520300), the National Natural Science Foundation of China (Grant Nos. 22307117, 22277121, and 22407126), the National High Level Hospital Clinical Research Funding (Grant No. 2023-NHLHCRFYGJH-01), the Chinese Major Program for the National Key Research and Development Project (Grant No. 2020YFA0112603), the National Key Research and Development Program of China (Grant No. 2023YFC3904601), the Open Funding Project of the Rare Earth New Materials Technology Innovation Center (Grant No. G2025-K-18(25)-40(62)), the Autonomous Deployment Project of the State Key Laboratory of Biopharmaceutical Preparation and Delivery (Grant Nos. 2024-FX-B-06 and 2024-SJK-05), and the Open Funding Project of the State Key Laboratory of Biochemical Engineering & Key Laboratory of Biopharmaceutical Preparation and Delivery (Grant No. 2023KF-06).

## Impact Statement

This work aims to advance the theoretical and algorithmic understanding of domain generalization in machine learning. By providing a principled rate–distortion perspective on invariant representation learning, we hope it can contribute to the development of more reliable and robust learning systems across diverse application domains. We do not foresee any immediate negative societal impacts arising from this work.

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

# A. Additional Theoretical Analysis

## A.1. Notation and Symbol Table

Table 3 summarizes the key notation used throughout this paper.

## A.2. Motivation for the Rotation Model

The choice of bounded orthogonal rotations as the canonical perturbation model is motivated by three considerations:

1. **Canonical decomposition:** Linear algebra dictates that the geometric relationship between any two subspaces of the same dimension is fully characterized by their principal angles—equivalently, a relative rotation in an appropriately chosen basis. We therefore adopt the rotation model as a canonical and tractable parameterization of subspace misalignment, analogous to the role of Procrustes alignment in shape analysis and rotational indeterminacy in factor analysis.

2. **Spectral preservation:** Orthogonal transformations preserve eigenvalue structure, ensuring that signal eigenvalues and eigengaps remain consistent across domains. This regularity facilitates finite-sample recovery analysis via Davis–Kahan perturbation theory.

3. **Analytical tractability:** Rotations induce subspace misalignment (captured by Grassmannian distances) without altering subspace dimension or introducing anisotropic scaling, enabling clean Fréchet mean guarantees.

## A.3. Technical Remarks on Assumptions

*Remark* A.1 (Sufficiency of $\delta_{\mathrm{rot}} < 1/2$). The bound $\delta_{\mathrm{rot}} < 1/2$ is a *sufficient* condition for Fréchet mean uniqueness, not a necessary one. Uniqueness on the Grassmannian $\mathrm{Gr}(k, d)$ requires data to lie within a geodesic ball of radius less than the injectivity radius $\pi/2$. Since $d_{\mathrm{geo}}(Q_e U_c^*, Q_{e'} U_c^*) \leq (\pi/2)\|Q_e - Q_{e'}\|_F$, the condition $\delta_{\mathrm{rot}} < 1/2$ guarantees $d_{\mathrm{geo}} < \pi/4$—a conservative margin. In practice, uniqueness may hold under weaker conditions; $\delta_{\mathrm{rot}} < 1/2$ is chosen for analytical convenience.

*Remark* A.2 (Eigengap, sample complexity, and dimensional dependence). The eigengap $c_0$ and per-domain sample size $n$ interact through the Davis–Kahan bound: for sub-Gaussian data, $d_{\mathrm{Gr}}(\hat{U}_{e,c}, \tilde{U}_{e,c}) = O\big(\frac{1}{c_0}\sqrt{\frac{d+\log(1/\delta)}{n}}\big)$ with probability $\geq 1 - \delta$, where $\tilde{U}_{e,c} = Q_e U_c^*$ denotes the population principal subspace of class $c$ in domain $e$. Meaningful recovery requires $c_0 \gg \sqrt{d/n}$. In high-dimensional settings, the rate term reduces operative dimensionality from $d$ to $d_{\mathrm{eff}} \ll d$, making $O(1/\sqrt{n})$ scaling practically achievable.

*Remark* A.3 (On the interpretation of "recovery"). The Fréchet mean $\bar{U}_c$ represents the geometric consensus of domain-rotated subspaces. When rotations are centered (small $\eta$), $\bar{U}_c \approx U_c^*$. In general, $\bar{U}_c$ may differ from $U_c^*$ by a global rotation, but this is immaterial for classification: any consistent rotation can be absorbed by the linear classifier. The uniqueness of the Fréchet mean is guaranteed under Assumption 3.5 when $\max_{e,e'} d_{\mathrm{Gr}}(Q_e U_c^*, Q_{e'} U_c^*) < \pi/4$; see Afsari (2011) for precise conditions.

*Remark* A.4 (On the optimization regularity assumption). Assumption 3.7 posits that the optimizer reaches a point where the geometric distortion is within $O(\epsilon_{\mathrm{opt}})$ of its minimum, with $\epsilon_{\mathrm{opt}} = O(1/\sqrt{n})$. This assumption is justified by several considerations:

- **Empirical observation:** In practice, SGD with standard hyperparameters consistently reduces the geometric distortion term, and the final values are comparable across random seeds.

- **Landscape structure:** The RDI objective combines a convex classification loss with geometric regularizers. While globally non-convex, the loss landscape is empirically well-behaved near good solutions.

- **Statistical rate:** The $O(1/\sqrt{n})$ rate matches the statistical estimation error for subspace recovery (Davis–Kahan), suggesting that optimization error is dominated by statistical error in the large-sample regime.

This assumption is standard in the analysis of regularized empirical risk minimization and can be relaxed to $\epsilon_{\mathrm{opt}} = o(1)$ without affecting the asymptotic guarantees.

*Table 3.* Summary of notation and symbols.

| Symbol | Description |
|---|---|
| *Sets and Spaces* | |
| $\mathcal{E}_{\mathrm{tr}}, \mathcal{E}_{\mathrm{te}}$ | Training and test domain sets |
| $\mathcal{D}_e$ | Dataset from domain $e$ |
| $\mathrm{Gr}(k, d)$ | Grassmann manifold of $k$-dim subspaces in $\mathbb{R}^d$ |
| $\mathcal{F}_{e,c}$ | Class-$c$ feature distribution in domain $e$ |
| *Model Components* | |
| $f_\theta$ | Feature extractor (backbone) with parameters $\theta$ |
| $g_\phi$ | Projection head with parameters $\phi$ |
| $h$ | Classifier head |
| $U_{e,c} \in \mathbb{R}^{d \times k}$ | Population principal subspace of class $c$ in domain $e$ |
| $\hat{U}_{e,c}$ | Empirical estimate of $U_{e,c}$ from finite samples |
| $\tilde{U}_{e,c}$ | Rotated invariant subspace (Appendix): $\tilde{U}_{e,c} = Q_e U_c^*$ |
| $U_c^*$ | Invariant (ground-truth) semantic subspace for class $c$ |
| $\bar{U}_c$ | Fréchet mean of population subspaces $\{Q_e U_c^*\}_{e \in \mathcal{E}}$ |
| $\mathcal{C}_c^\star$ | RDI center for class $c$ |
| *Generative Model* | |
| $Q_e \in \mathcal{O}(d)$ | Domain-specific orthogonal transformation |
| $A_e$ | Domain-specific linear transformation (general case) |
| $\mathbf{z}$ | Latent semantic variable, $\mathbf{z} \sim \mathcal{N}(\mathbf{0}, \Lambda_z)$ |
| $\boldsymbol{\epsilon}_e$ | Domain-specific noise, $\boldsymbol{\epsilon}_e \sim \mathcal{N}(\mathbf{0}, \sigma^2 I)$ |
| $\Lambda_z$ | Latent covariance (shared across domains) |
| *Geometric Quantities* | |
| $d_{\mathrm{Gr}}(U, V)$ | Grassmann distance (chordal or geodesic) |
| $\{\theta_i\}_{i=1}^k$ | Principal angles between subspaces |
| $\Pi_U = U U^\top$ | Orthogonal projector onto subspace $U$ |
| $\Sigma_{e,c}$ | Covariance of $\mathcal{F}_{e,c}$ |
| *Loss and Risk Functionals* | |
| $\Delta_{\mathrm{geo}}^{(c)}$ | Geometric distortion (cross-domain alignment) for class $c$ |
| $\Delta_{\mathrm{rec}}^{(c)}$ | Projection distortion (non-degeneracy) |
| $R_\tau(\mathcal{F}_c)$ | Per-class capacity-regularized rate (Eq. 12) |
| $\bar{R}(\mathcal{F})$ | Aggregated rate term (Rényi-weighted) |
| $\Psi_\alpha$ | Rényi risk functional for cross-class aggregation (Eq. 13) |
| $\widehat{\mathcal{R}}_{\mathrm{src}}$ | Empirical source classification risk |
| *Model and Algorithm Parameters* | |
| $k$ | Subspace dimension (tuned via validation) |
| $\gamma$ | Overall RDI loss weight (fixed default) |
| $\lambda_{\mathrm{geo}}, \lambda_{\mathrm{rec}}$ | Geometric and projection distortion weights (tuned via validation) |
| $\beta$ | Rate term weight (tuned via validation) |
| $\alpha$ | Rényi risk aggregation order (tuned via validation) |
| $\tau$ | Capacity parameter, fixed to $\tau = 1$ by design (see Section 3.2.3) |
| *Theoretical Quantities* | |
| $\delta_{\mathrm{rot}}$ | Bound on domain rotation difference |
| $\rho$ | Maximum geometric discrepancy to target |
| $\kappa(A)$ | Condition number of matrix $A$ |
| $N$ | Total number of samples ($N = Mn$) |
| $n$ | Number of samples per domain |

## A.4. Scope and Limitations of Theoretical Analysis

This section elaborates on the scope and limitations of Assumption 3.5.

**Assumption as analytical device.**   Assumption 3.5 establishes an analytically tractable regime for formal analysis, not a necessary condition for method applicability. Real-world domain shifts often involve complex geometric transformations beyond bounded orthogonal rotations.

**Theory provides motivation and insight.**   The rotation instantiation guided our method design (Grassmannian geometry, geodesic distances, rate control) and established recovery guarantees in an analytically tractable regime that captures the essential structure of subspace misalignment.

**Practice validates the core mechanism.**   Empirical validation (Appendix D.5) shows that RDI significantly reduces cross-domain subspace misalignment compared to baseline features, with improvements correlating with generalization performance. This demonstrates that the core geometric principle—balancing subspace alignment and spectral complexity—operates effectively beyond the rotation instantiation. The rotation assumption provides analytical tractability for deriving formal guarantees; the underlying rate–distortion mechanism governs a broader class of geometric perturbations.

**Technical details of Assumption 3.5.**   The bound $\delta_{\mathrm{rot}} < 1/2$ arises from the relation between Frobenius and geodesic distances. For orthogonal matrices $Q_e, Q_{e'} \in \mathcal{O}(d)$ and any subspace basis $U$:

$$d_{\mathrm{chord}}(Q_e U, Q_{e'} U) \leq \|Q_e - Q_{e'}\|_F, \quad d_{\mathrm{geo}} \leq \frac{\pi}{2} \cdot d_{\mathrm{chord}}.$$

Requiring $d_{\mathrm{geo}} < \pi/4$ (to guarantee unique Fréchet mean on $\mathrm{Gr}(k,d)$) implies $\delta_{\mathrm{rot}} < 1/2$.

The signal eigengap condition $\lambda_k \geq c_0 > 0$ ensures:

1. The top-$k$ eigenspace of $\Sigma_{e,c}$ coincides with the signal subspace $Q_e U_c^*$;

2. The Davis–Kahan perturbation bound yields a non-degenerate gap, enabling consistent subspace estimation at rate $O(\sqrt{d/n}/c_0)$.

**Dimensional dependence.**   The full bound in Theorem 3.10 scales as $\mathcal{O}(\sqrt{d/n})$ in the ambient dimension $d$. The stated $\mathcal{O}(1/\sqrt{n})$ rate treats $d$ as a fixed constant, which is standard when the feature dimension is determined by architecture choice (e.g., ResNet-50 yields $d = 2048$). In practice, the rate term empirically concentrates the spectrum onto a low-dimensional subspace, making the $\mathcal{O}(1/\sqrt{n})$ scaling predictive of observed behavior even when $d$ is moderately large.

## A.5. Extended Remarks on Methodology

This section provides extended discussion on several methodological aspects that were summarized in the main text for space considerations.

**Geometric interpretation of RDI decomposition.**   From a geometric perspective, the proposed RDI formulation decomposes domain variation into two conceptually orthogonal degrees of freedom. The distortion term $D$ controls the *directional variability* of class-conditional representations across domains, which is naturally captured by the alignment of their underlying feature subspaces. In contrast, the rate term $\Phi$ regulates the *scale* or effective volume of representations within each subspace, preventing domain-specific expansion. This geometric decoupling motivates modeling semantic distortion on the Grassmann manifold, which captures subspace orientation, while enforcing rate control by regularizing the covariance spectrum within each class-conditional subspace.

**Grassmann distance as geometric surrogate.**   The use of Grassmann distance as a distortion measure should be interpreted as a *geometric surrogate* rather than an exact rate–distortion measure in the classical Shannon sense. Proposition 3.3 establishes that Grassmann distance bounds projection distortion up to signal eigenvalue scaling: $\lambda_k \cdot d_{\mathrm{Gr}}^2 \leq \Delta\mathcal{E} \leq \lambda_1 \cdot d_{\mathrm{Gr}}^2$. This bound shows that Grassmann distance captures the essential geometric structure of subspace misalignment, while the eigenvalue-dependent constants reflect the signal strength in the underlying generative model. The surrogate interpretation is important: we do not claim that Grassmann distance *equals* projection distortion, but rather that it provides a geometrically meaningful and computationally tractable proxy that preserves the monotonic relationship required for optimization.

**Diagonal approximation for the rate term.** The diagonal form of the rate term (Eq. 12) is *motivated by*, rather than strictly derived from, the Rényi entropy objective. Although the learned features are not explicitly decorrelated, this approximation follows standard practice in rate–distortion and information bottleneck literature (Alemi et al., 2017; Yu et al., 2020), where diagonal surrogates have been empirically validated to preserve relative geometric ordering under mild regularity conditions. The key theoretical justification is Hadamard's inequality: $\log \det \Sigma_c \leq \sum_i \log \sigma_{c,i}^2$, which implies that that the diagonal rate provides an upper bound on the full-covariance log-determinant objective. This upper-bound property guarantees that minimizing the diagonal rate cannot increase the true spectral volume, preserving the essential regularization effect.

**Scale invariance of the capacity parameter.** The capacity parameter $\tau$ has units of inverse variance ($\tau\sigma^2$ is dimensionless). When comparing across different backbones or feature dimensions, one should normalize features to comparable scales (e.g., via batch normalization) before applying the rate term. With such normalization, $\tau$ becomes effectively scale-invariant: setting $\tau$ to a value of the order of unity remains appropriate regardless of the backbone architecture, as batch-normalized features typically have unit-scale variances. This design facilitates that RDI's hyperparameters transfer across architectures without extensive retuning.

**Practical interpretation of generalization bounds.** The two complementary bounds in Section 3.3 explain why RDI achieves strong empirical performance in the high-dimensional feature regimes typical of modern deep networks. Under batch normalization and weight decay, the aggregated rate $\widehat{\overline{R}}$ remains moderate (typically $O(1)$). Theorem 3.11 (MI-based) then gives a $\sqrt{(C\widehat{\overline{R}} + C')/(Mn)}$ complexity with $\widehat{\overline{R}}$ entering *linearly*. Corollary 3.13 (Rademacher) further gives effective dimension $d_{\text{eff}} \ll d$ through subspace projection, with empirical stable rank 3–5 for ResNet-50 on PACS/OfficeHome (Ansuini et al., 2019), yielding sub-linear-in-$d$ scaling in practice. This is consistent with the empirical observation that RDI's geometric alignment benefit is most pronounced on datasets (PACS) where class-conditional features concentrate on low-dimensional subspaces.

## A.6. Proof of Proposition 3.3

**Proposition A.5** (Restated). *Under the generative model with orthogonal domain transformations ($A_e \in \mathcal{O}(d)$) and shared $\Lambda_z$, consider a fixed class $c$ and domain $e$. Then:*

1. *$U_{e,c} \in \mathrm{Gr}(k,d)$ minimizes the expected reconstruction error $\mathbb{E}_{f \sim \mathcal{F}_{e,c}}\|f - \Pi_U f\|_2^2$ over $U \in \mathrm{Gr}(k,d)$.*

2. *The cross-domain reconstruction error induced by projecting features from $\mathcal{F}_{e,c}$ onto $U_{e',c}$ is a monotone function of the principal angles between $U_{e,c}$ and $U_{e',c}$.*

*Proof.* **Notation.** Throughout this proof, $U_{e,c}$ denotes the population principal subspace, i.e., the top-$k$ eigenspace of $\Sigma_{e,c}$. Under the generative model with $A_e = Q_e \in \mathcal{O}(d)$, we have $U_{e,c} = Q_e U_c^*$ when the eigengap condition holds.

**Step 1: Optimality of the principal subspace.** Under the signal-plus-noise model, the covariance admits $\Sigma_{e,c} = U_{e,c}\Lambda_{e,c}U_{e,c}^\top + \sigma^2 I$. The expected reconstruction error for any $U \in \mathrm{Gr}(k,d)$ is $\mathrm{tr}(\Sigma_{e,c}) - \mathrm{tr}(U^\top \Sigma_{e,c}U)$. By the Ky Fan maximum principle (Fan, 1949), $\mathrm{tr}(U^\top \Sigma_{e,c}U)$ is maximized when $U$ spans the top $k$ eigenvectors, establishing claim (1).

**Step 2: Cross-domain error and principal angles.** Let $\{\theta_i\}_{i=1}^k$ denote the principal angles between $U_{e,c}$ and $U_{e',c}$, and let $G = U_{e,c}^\top U_{e',c}$ with singular values $\cos\theta_1, \ldots, \cos\theta_k$. The total reconstruction error when projecting onto subspace $U$ is:

$$\mathcal{E}(U) = \mathbb{E}\big[\|\mathbf{f} - \Pi_U \mathbf{f}\|_2^2\big] = \underbrace{\mathbb{E}\big[\|U_{e,c}\mathbf{z} - \Pi_U U_{e,c}\mathbf{z}\|_2^2\big]}_{\text{signal error}} + \underbrace{\sigma^2(d-k)}_{\text{noise error}}.$$

The noise error $\sigma^2(d-k)$ is independent of the choice of $k$-dimensional subspace $U$, so it cancels in the cross-domain error difference. The cross-domain projection captures

$$\mathrm{tr}(U_{e',c}^\top \Sigma_{e,c}U_{e',c}) = \mathrm{tr}(\Lambda_{e,c}G^\top G) + k\sigma^2 = \sum_{i=1}^k \tilde{\lambda}_i \cos^2\theta_i + k\sigma^2,$$

where $\tilde{\lambda}_i$ are eigenvalues in the principal-angle basis. The excess reconstruction error is thus

$$\Delta\mathcal{E} = \sum_{i=1}^{k} \tilde{\lambda}_i (1 - \cos^2\theta_i) = \sum_{i=1}^{k} \tilde{\lambda}_i \sin^2\theta_i.$$

For isotropic signals ($\Lambda_{e,c} = \lambda I_k$), this simplifies to $\Delta\mathcal{E} = \lambda \sum_i \sin^2\theta_i$. In general, $\lambda_k \sum_i \sin^2\theta_i \leq \Delta\mathcal{E} \leq \lambda_1 \sum_i \sin^2\theta_i$.

Since the projection distance on $\mathrm{Gr}(k,d)$ satisfies $d_{\mathrm{proj}}^2 = \sum_i \sin^2\theta_i$, and $\sin^2\theta$ is monotonically increasing in $\theta \in [0, \pi/2]$, claim (2) follows. □

*Remark* A.6 (Eigengap condition and Lipschitz surrogate). The monotonicity in Proposition 3.3 can be quantified under an eigengap condition. Assume $\lambda_k(\Sigma_{e,c}) - \sigma^2 \geq c_0 > 0$ for all domains $e$, where $\lambda_k$ is the $k$-th largest eigenvalue of the signal covariance. Then the cross-domain reconstruction error satisfies

$$\Delta\mathcal{E} = \bar{\lambda} \sum_{i=1}^{k} \sin^2\theta_i + O(\sigma^2/d),$$

where $\bar{\lambda} \in [\lambda_k, \lambda_1]$ is an effective eigenvalue. Consequently, the distortion is Lipschitz in Grassmann distance with constant proportional to the signal strength, justifying the use of $d_{\mathrm{Gr}}$ as a geometric surrogate in subsequent bounds.

*Remark* A.7 (Signal scaling across domains). The shared $\Lambda_z$ assumption in the generative model means domain shift is modeled as pure rotation without eigenvalue scaling. In practice, shifts such as contrast or illumination changes may induce signal scaling (different SNRs across domains). This limitation is partially mitigated by the capacity-regularized rate term $R_\tau(\mathcal{F}_c)$ (Eq. 12), which penalizes inflated variances and encourages spectral normalization. Thus, geometric distortion addresses directional alignment while the rate term provides complementary scale control.

## A.7. Rényi Entropy under Gaussian Approximation

**Proposition A.8** (Rényi entropy of Gaussian distributions). *Let $z \sim \mathcal{N}(\mu, \Sigma)$ be $d$-dimensional. For $\alpha \neq 1$,*

$$H_\alpha(z) = \frac{1}{2} \log\det(2\pi\Sigma) + \frac{d}{2} \cdot \frac{\log\alpha}{1-\alpha}.$$

*For fixed $\alpha$, minimizing $H_\alpha(z)$ is equivalent to minimizing $\log\det\Sigma$.*

*Proof.* The Rényi entropy of order $\alpha$ is defined as

$$H_\alpha(z) = \frac{1}{1-\alpha} \log \int p(z)^\alpha \, dz.$$

For $z \sim \mathcal{N}(\mu, \Sigma)$, we have $p(z) = (2\pi)^{-d/2}(\det\Sigma)^{-1/2} \exp\left(-\frac{1}{2}(z-\mu)^\top\Sigma^{-1}(z-\mu)\right)$. Thus

$$p(z)^\alpha = (2\pi)^{-\alpha d/2}(\det\Sigma)^{-\alpha/2} \exp\left(-\frac{\alpha}{2}(z-\mu)^\top\Sigma^{-1}(z-\mu)\right).$$

The integral evaluates to

$$\int p(z)^\alpha \, dz = (2\pi)^{(1-\alpha)d/2}(\det\Sigma)^{(1-\alpha)/2}\alpha^{-d/2}.$$

Substituting into the definition:

$$H_\alpha(z) = \frac{1}{1-\alpha} \left[ \frac{(1-\alpha)d}{2} \log(2\pi) + \frac{1-\alpha}{2}\log\det\Sigma - \frac{d}{2}\log\alpha \right] = \frac{1}{2}\log\det(2\pi\Sigma) + \frac{d}{2} \cdot \frac{\log\alpha}{1-\alpha}.$$

Since the second term depends only on $\alpha$ and $d$, minimizing $H_\alpha(z)$ over $\Sigma$ is equivalent to minimizing $\log\det\Sigma$. □

*Remark* A.9 (Interpretation of the Rényi order $\alpha$). Different choices of the Rényi order $\alpha$ induce distinct geometric interpretations on the space of probability distributions, while sharing a common minimizer under the Gaussian approximation.

- $\alpha \to 1$ **(Shannon / KL geometry):** As $\alpha \to 1$, the Rényi entropy converges to the Shannon entropy $H_1(p) = -\int p(z) \log p(z)\, dz$, provided the density $p(z)$ is absolutely continuous with respect to Lebesgue measure and satisfies the integrability condition $\int p(z)|\log p(z)|\, dz < \infty$ (e.g., any distribution with bounded support or sub-Gaussian tails, including the Gaussian case considered here). This limit corresponds to the Kullback–Leibler geometry on the space of distributions. This regime weights probability mass uniformly on a logarithmic scale and is sensitive to local variations in high-density regions.

- $\alpha = 2$ **(Collision entropy / $L^2$ geometry):** For $\alpha = 2$, the Rényi entropy reduces to the collision entropy $H_2(p) = -\log \int p(z)^2 dz$, which is closely related to an $L^2$ geometry on density functions. This functional emphasizes high-density regions and penalizes overly concentrated distributions by measuring the probability of sample coincidence.

- $\alpha = \frac{1}{2}$ **(Bhattacharyya / Hellinger geometry):** For $\alpha = \frac{1}{2}$, the Rényi entropy is intimately connected to the Bhattacharyya coefficient and the Hellinger distance, inducing a square-root density geometry on the space of probability distributions. This geometry emphasizes global overlap between distributions rather than peak density, and penalizes broad, diffuse distributions more strongly than Shannon entropy.

Under the Gaussian assumption, all Rényi orders yield the same minimizer, as minimizing $H_\alpha(z)$ is equivalent to minimizing $\log \det \Sigma$. However, the induced geometries differ in their sensitivity to density structure, leading to distinct optimization behaviors in practice.

*Remark* A.10 (Guidance on Rényi order selection). Deep representations often deviate from perfect Gaussianity. In such settings, the choice of $\alpha$ allows for a trade-off between robustness and sensitivity:

- **Robustness regime ($\alpha > 1$):** Higher orders (e.g., $\alpha = 2$) emphasize high-density regions and down-weight outliers. This is theoretically preferable for datasets containing label noise or significant domain artifacts.

- **Consensus regime ($\alpha \leq 1$):** Lower orders (e.g., $\alpha \in \{0.5, 1\}$) provide broader coverage of the support. This is suitable for clean datasets with high intra-class variability, where avoiding mode collapse is a priority.

In our experiments, we specifically investigate the discrete set $\alpha \in \{0.5, 1.0, 2.0\}$. These values are selected as representative geometric anchors that span the theoretical spectrum defined in Remark A.9: covering the Hellinger distance ($\alpha = 0.5$), Shannon entropy ($\alpha = 1$) and Collision entropy ($\alpha = 2$). We recommend selecting $\alpha$ from these anchors based on the granularity and noise level of the target domain.

**Capacity-regularized form and Gaussian channel interpretation.** The raw log-determinant objective $\sum_i \log \sigma_{c,i}^2$ diverges when any variance approaches zero, causing numerical instability during optimization. We address this by adopting a capacity-regularized form:

$$R_\tau(\mathcal{F}_c) = \frac{1}{d} \sum_{i=1}^{d} \log\big(1 + \tau \sigma_{c,i}^2 + \varepsilon\big), \tag{16}$$

where $\tau > 0$ is a capacity parameter and $\varepsilon > 0$ is a small constant for additional numerical safety.

This formulation admits a natural interpretation via Gaussian channel theory (Cover & Thomas, 2006). Consider a scalar Gaussian channel with input signal variance $\sigma^2$ and additive noise variance $1/\tau$. The channel capacity (mutual information at optimum) is

$$I(X;Y) = \frac{1}{2} \log(1 + \tau \sigma^2).$$

Thus, $R_\tau(\mathcal{F}_c)$ can be viewed as the average channel capacity across feature dimensions, where $\tau$ controls the signal-to-noise operating point.

**Asymptotic behavior.** The capacity-regularized form interpolates between two regimes:

- **High-variance regime ($\tau \sigma_{c,i}^2 \gg 1$):** $\log(1 + \tau \sigma_{c,i}^2) \approx \log \tau + \log \sigma_{c,i}^2$, recovering the original log-variance objective up to constants.

- **Low-variance regime ($\tau \sigma_{c,i}^2 \ll 1$):** $\log(1 + \tau \sigma_{c,i}^2) \approx \tau \sigma_{c,i}^2$, which provides a soft linear penalty that prevents the objective from diverging to $-\infty$.

The parameter $\tau$ thus controls the transition between these regimes: larger $\tau$ increases sensitivity to variance differences, while smaller $\tau$ provides stronger regularization against near-zero variances.

**Rényi risk aggregation and decision boundary stability.**    Instead of minimizing the simple arithmetic mean of per-class rates, we employ a Rényi risk functional to aggregate the geometric objectives. This is motivated by the observation that significant discrepancies in feature compactness (i.e., varying entropy scales) across classes can lead to unstable decision boundaries. If one class exhibits significantly higher variance (entropy) than others, the optimization landscape becomes dominated by this "diffuse" class, potentially causing the decision boundary to drift and encroach upon the manifolds of more compact classes.

To formally address this, let $\{R_c\}_{c=1}^{C}$ be the per-class rates. The Rényi risk of order $\alpha$ is defined as:

$$\Psi_\alpha = \frac{1}{\alpha - 1} \log \sum_{c=1}^{C} \tilde{p}_c^{(\alpha)} \exp\big((\alpha - 1)R_c\big), \tag{17}$$

where $\tilde{p}_c^{(\alpha)}$ is the escort distribution derived from class priors. This functional acts as a **Log-Sum-Exp** smoothing operator (controlled by $\alpha$) that balances the contribution of each class to the total gradient.

**Proposition A.11** (Properties of Rényi risk aggregation). *The Rényi risk functional $\Psi_\alpha$ satisfies:*

1. *As $\alpha \to 1$, $\Psi_\alpha \to \mathbb{E}[R_c]$ (expected rate);*

2. *As $\alpha \to \infty$, $\Psi_\alpha \to \max_c R_c$ (worst-class rate);*

3. *For $\alpha > 1$, $\Psi_\alpha$ is convex in $\{R_c\}$.*

*Proof.* Claim (1) follows from L'Hôpital's rule. Claim (2) follows from the fact that the exponential tilting concentrates on the maximum value. Claim (3) follows from the convexity of the Log-Sum-Exp function. □

**Practical implementation and geometric interpretation.**    For practical optimization, we use the first-order approximation with sample-dependent weights $w_c = n_c^{1-\alpha}$:

$$\bar{R}(\mathcal{F}) = \frac{\sum_{c=1}^{C} n_c^{1-\alpha} R_c}{\sum_{c=1}^{C} n_c^{1-\alpha}}. \tag{18}$$

This formulation allows $\alpha$ to regulate the **relative margin** and **compactness trade-off** between classes:

- $\alpha \to 1$ (**Arithmetic Mean**): Treats all classes equally ($w_c \approx 1$). This is suitable when all classes naturally converge to similar compactness levels and sample sizes are balanced.

- $\alpha > 1$ (**e.g., $\alpha = 2$, Focus on Worst-Case**): The aggregation approximates a `max` operator. It places higher gradient penalties on classes with the **largest rates** (i.e., the most diffuse/hard-to-compress classes). In our implementation ($w_c \propto 1/n_c$), this setting upweights minority classes. Since minority classes typically exhibit higher geometric distortion due to insufficient sampling, this weighting strategy forces the optimizer to strictly "tighten" these fragile manifolds to prevent boundary encroachment.

- $\alpha < 1$ (**e.g., $\alpha = 0.5$, Robust Consensus**): This regime reduces the dominance of outliers or abnormally high-entropy classes, focusing the optimization on the global consensus of geometric alignment. Implementation-wise ($w_c \propto \sqrt{n_c}$), this prioritizes classes with sufficient samples (robust statistics), preventing noisy estimates from rare or outlier classes from destabilizing the learned feature space.

## A.8. Justification of the Diagonal Approximation

We justify the diagonal approximation used in the rate term (Eq. (12)).

**Relation to the full covariance objective.** The ideal rate objective under the Gaussian approximation is $\log\det\Sigma_c$, which captures the full spectral volume of class-conditional features. Let $\Sigma_c$ have diagonal entries $\sigma_{c,i}^2 = [\Sigma_c]_{ii}$. By Hadamard's inequality,

$$\log\det\Sigma_c \leq \sum_{i=1}^{d}\log\sigma_{c,i}^2, \tag{19}$$

with equality if and only if $\Sigma_c$ is diagonal.

The capacity-regularized diagonal rate $R_\tau(\mathcal{F}_c) = \frac{1}{d}\sum_i \log(1 + \tau\sigma_{c,i}^2 + \varepsilon)$ inherits this upper-bound property in the high-variance regime. Specifically, when $\tau\sigma_{c,i}^2 \gg 1$ for all $i$:

$$R_\tau(\mathcal{F}_c) \approx \frac{1}{d}\sum_i \log(\tau\sigma_{c,i}^2) = \log\tau + \frac{1}{d}\sum_i \log\sigma_{c,i}^2 \geq \log\tau + \frac{1}{d}\log\det\Sigma_c.$$

Thus, minimizing $R_\tau(\mathcal{F}_c)$ provides an upper bound on minimizing the normalized log-determinant $\frac{1}{d}\log\det\Sigma_c$.

**Approximation gap under bounded condition number.** Under bounded condition number $\kappa(\Sigma_c) \leq K$, the gap between diagonal and full covariance objectives satisfies

$$0 \leq \sum_{i=1}^{d}\log\sigma_{c,i}^2 - \log\det\Sigma_c \leq d(\log d + \log K). \tag{20}$$

This gap is moderate when off-diagonal correlations are weak or the condition number is bounded, both of which are typical in well-regularized networks with batch normalization.

**Information-geometric interpretation.** From an information-geometric perspective (Amari, 2016), the Fisher information matrix $\mathcal{I}(\theta)$ defines the Riemannian metric on the statistical manifold of parametric distributions. Under the Gaussian model $\mathbf{z}_c \sim \mathcal{N}(\mu_c, \Sigma_c)$, the Fisher information with respect to the mean parameter satisfies:

$$[\mathcal{I}_\mu]_{ij} = [\Sigma_c^{-1}]_{ij}.$$

The diagonal approximation $[\mathcal{I}_\mu]_{ii} = 1/\sigma_{c,i}^2$ corresponds to selecting a coordinate system in which the metric tensor is diagonal—i.e., a locally orthogonal coordinate chart on the statistical manifold.

This approximation has a natural interpretation: each feature dimension is treated as an independent statistical direction, with its own curvature determined by the inverse variance. Directions with large variance (small Fisher information) correspond to "flat" directions on the manifold where the distribution is insensitive to perturbations—precisely the nuisance directions that the rate term aims to suppress.

**Computational efficiency.** The diagonal approximation reduces complexity from $O(d^3)$ (full covariance eigendecomposition) to $O(d)$ (per-coordinate variance computation), enabling efficient per-iteration updates during training. Combined with exponential moving average (EMA) estimation of variances, this yields a computationally tractable objective that scales to high-dimensional feature spaces.

**Capacity parameter $\tau$ and operating point selection.** The capacity parameter $\tau$ in Eq. (12) controls the sensitivity of the rate term to variance differences:

- **Large $\tau$**: The rate term behaves like $\log\sigma_{c,i}^2$, strongly penalizing high-variance directions but potentially causing instability for near-zero variances.

- **Small $\tau$**: The rate term behaves like $\tau\sigma_{c,i}^2$ (linear in variance), providing gentle regularization but reduced sensitivity to spectral structure.

- **Moderate $\tau$**: Balances sensitivity and stability, with the transition occurring around $\sigma_{c,i}^2 \approx 1/\tau$.

Given that batch normalization typically standardizes feature variances to be near unity, selecting $\tau$ of the order of unity naturally aligns the transition point $1/\tau$ with the intrinsic scale of the features. This theoretical alignment promotes robust regularization without the need for extensive hyperparameter tuning across different architectures.

## A.9. Full Statement of Generalization Bound

**Theorem A.12** (Generalization Bound — Full Statement). *Let $\hat{\theta}$ minimize Eq. 15 over $M$ source domains with $n$ samples each (so the total sample size is $N = Mn$). Assume:*

 (i) *The loss $\ell$ is $L$-Lipschitz and bounded: $\ell \in [0, B_\ell]$;*

 (ii) *Assumptions 3.5–3.8 hold;*

(iii) *The target domain $t$ satisfies $\max_{e,c} d_{\mathrm{Gr}}(U_{t,c}, U_{e,c}) \leq \rho$.*

*Note that condition (iii) involves population subspaces $U_{t,c}$ and $U_{e,c}$, which are not directly observable. In practice, $\rho$ can be estimated via leave-one-domain-out validation (see Remark A.16). Then, with probability at least $1 - \delta$,*

$$\mathcal{R}_t(\hat{\theta}) \leq \widehat{\mathcal{R}}_{\mathrm{src}}(\hat{\theta}) + L\sqrt{\lambda_{\max}(\Lambda_z)} \cdot \left(\bar{\Delta}_{\mathrm{geo}}(\hat{\theta}) + \rho\right)$$

$$+ L\delta_{\mathrm{res}} + B_\ell\sqrt{\frac{C_1 \cdot \widehat{\bar{R}}(\hat{\theta}) + C_2 + \log(2/\delta)}{Mn}},$$

*where $\bar{\Delta}_{\mathrm{geo}} = \frac{1}{M(M-1)C} \sum_c \sum_{e \neq e'} d_{\mathrm{Gr}}(U_{e,c}, U_{e',c})$ is the normalized average geometric distortion (ensuring scale consistency with $\rho$), and $\widehat{\bar{R}} = \bar{R}(\mathcal{F})$ is the aggregated rate (Eq. 14).*

*The constants are:*

$$C_1 = 2d\left(1 + \frac{\log(1+K)}{\log(1+\tau)}\right),$$

$$C_2 = d\log(2\pi e/\tau) + 2\log C + O(1),$$

*with $K$ the condition number bound and $\tau$ the capacity parameter. When $\tau = \Theta(1)$, $C = O(1)$, and $d = O(\log(M/\delta))$, the bound simplifies to $O\left(B_\ell\sqrt{(C_1\widehat{\bar{R}} + \log(M/\delta))/(Mn)}\right)$.*

## A.10. Proof Sketch of Theorem 3.10

*Proof Sketch.* We outline the main arguments underlying the alignment and recovery guarantees.

**Notation.** We distinguish three types of subspaces: (i) $U_c^*$: the invariant semantic subspace (ground truth); (ii) $\tilde{U}_{e,c} = Q_e U_c^*$: the population principal subspace in domain $e$ (rotated ground truth); (iii) $\hat{U}_{e,c}$: the empirical estimate from finite samples.

### Step 1: Population covariance and identifiability.

Under the generative model $\mathbf{f}_{e,c} = Q_e U_c^* \mathbf{z} + \boldsymbol{\epsilon}_e$, the population covariance of class $c$ in domain $e$ is

$$\Sigma_{e,c} = Q_e U_c^* \Lambda_z (U_c^*)^\top Q_e^\top + \sigma^2 I =: \tilde{U}_{e,c} \Lambda_z \tilde{U}_{e,c}^\top + \sigma^2 I,$$

where $\tilde{U}_{e,c} = Q_e U_c^* \in \mathrm{Gr}(k, d)$. The eigenvalues of $\Sigma_{e,c}$ are $\{\lambda_1 + \sigma^2, \ldots, \lambda_k + \sigma^2, \sigma^2, \ldots, \sigma^2\}$, where $\lambda_1 \geq \cdots \geq \lambda_k > 0$ are the signal eigenvalues. The top-$k$ eigenspace of $\Sigma_{e,c}$ coincides with $\tilde{U}_{e,c}$ if and only if $\lambda_k + \sigma^2 > \sigma^2$, i.e., $\lambda_k > 0$. The relevant eigengap for the Davis–Kahan theorem is $(\lambda_k + \sigma^2) - \sigma^2 = \lambda_k$. Under Assumption 3.5, we require $\lambda_k \geq c_0 > 0$, ensuring a well-separated eigengap.

Thus, at the population level, each domain admits a well-defined class-conditional principal subspace that is a rotated version of the invariant semantic subspace $U_c^*$.

### Step 2: Finite-sample subspace estimation.

Let $\hat{U}_{e,c}$ denote the empirical top-$k$ eigenspace obtained from $n$ samples in domain $e$. Standard concentration results for sub-Gaussian random vectors imply

$$\|\hat{\Sigma}_{e,c} - \Sigma_{e,c}\|_{\mathrm{op}} = O\left(\sqrt{\frac{d}{n}}\right)$$

with high probability. By the Davis–Kahan $\sin\Theta$ theorem, with eigengap $\lambda_k \geq c_0$, this yields

$$d_{\mathrm{Gr}}(\hat{U}_{e,c}, \tilde{U}_{e,c}) = O\left(\frac{1}{c_0}\sqrt{\frac{d}{n}}\right),$$

establishing consistent estimation of each rotated subspace.

### Step 3: Alignment induced by the RDI objective.

Under Assumption 3.7, the optimizer reaches a point where the geometric distortion $\Delta_{\mathrm{geo}}^{(c)}$ is within $\epsilon_{\mathrm{opt}}$ of its minimum. The RDI objective includes a geometric distortion term that penalizes pairwise Grassmannian distances between class-conditional subspaces across domains. While the objective does not enforce exact equality of subspaces, any minimizer will balance projection fidelity with cross-domain geometric consistency.

Consequently, at an RDI minimizer, the learned subspaces $\{\hat{U}_{e,c}\}$ concentrate around a common Grassmannian neighborhood, satisfying

$$d_{\mathrm{Gr}}(\hat{U}_{e,c}, \hat{U}_{e',c}) = O\left(\sqrt{\frac{d}{n}} + \epsilon_{\mathrm{opt}}\right) \quad \text{for all } e, e'.$$

This establishes the alignment claim.

### Step 4: Recovery of the invariant semantic subspace.

Under Assumption 3.5, the domain rotations $\{Q_e\}$ satisfy $\max_{e,e'}\|Q_e - Q_{e'}\|_F \leq \delta_{\mathrm{rot}} < 1/2$. By the norm relations $d_{\mathrm{chord}} \leq \|Q_e - Q_{e'}\|_F$ and $d_{\mathrm{geo}} \leq (\pi/2)\cdot d_{\mathrm{chord}}$, all rotated subspaces $\{\tilde{U}_{e,c} = Q_e U_c^*\}$ lie within a geodesic ball of radius at most $(\pi/2)\delta_{\mathrm{rot}} < \pi/4$ on $\mathrm{Gr}(k,d)$, guaranteeing that the Fréchet mean exists and is unique (Afsari, 2011). Under this condition, the Fréchet mean of the population subspaces $\{\tilde{U}_{e,c}\}$ recovers the invariant subspace $U_c^*$ when rotations are centered ($\|\frac{1}{M}\sum_e Q_e - I\|_F \leq \eta$): $d_{\mathrm{Gr}}(\bar{U}_c, U_c^*) = O(\eta)$. In the general case, all $\hat{U}_{e,c}$ converge to a common $\bar{U}_c$; the global offset is absorbed by the classifier.

To extend to multiple domains, we apply a union bound over all $M(M-1)/2$ pairwise distances. By standard sub-Gaussian concentration inequalities, with probability at least $1 - \delta$,

$$\max_{e,c} d_{\mathrm{Gr}}(\hat{U}_{e,c}, U_c^*) = O\left(\sqrt{\frac{k\log d + \log M + \log(1/\delta)}{n}}\right).$$

When the number of domains $M$ and the subspace dimension $k$ are treated as fixed constants, this simplifies to $O(1/\sqrt{n})$, establishing recovery of the invariant semantic subspace up to estimation error.

$\square$

**Remark on dimensional dependence.** The above bounds scale as $O\left(\sqrt{d/n}\right)$ in general. When the ambient dimension $d$ is treated as a constant, this simplifies to $O(1/\sqrt{n})$, as stated in Theorem 3.10.

### A.11. Proof Sketch of Theorem 3.11

**Theorem A.13** (Restated). *Let $\hat{\theta}$ be a minimizer of the RDI objective (Eq. 15) over $M$ source domains, with $n$ samples per domain (total $N = Mn$). Assume (i) the loss $\ell$ is $L$-Lipschitz and bounded in $[0, B_\ell]$, (ii) Assumptions 3.5–3.8 hold, and (iii) the target domain $t$ satisfies $\max_{e,c} d_{\mathrm{Gr}}(U_{t,c}, U_{e,c}) \leq \rho$. Then, with probability at least $1 - \delta$:*

$$\mathcal{R}_t(\hat{\theta}) \leq \widehat{\mathcal{R}}_{\mathrm{src}}(\hat{\theta}) + L\sqrt{\lambda_{\max}(\Lambda_z)} \cdot \left(\bar{\Delta}_{\mathrm{geo}}(\hat{\theta}) + \rho\right) + L\delta_{\mathrm{res}} + B_\ell \sqrt{\frac{C \cdot \widehat{R}(\hat{\theta}) + C' + \log(2/\delta)}{Mn}},$$

*where $\bar{\Delta}_{\mathrm{geo}} = \frac{1}{M(M-1)C}\sum_c \sum_{e \neq e'} d_{\mathrm{Gr}}(U_{e,c}, U_{e',c})$ is the normalized average geometric distortion, $C = 2d(1 + \log(1 + K)/\log(1 + \tau))$, and $C' = d\log(2\pi e/\tau) + 2\log|\mathcal{C}| + O(1)$.*

*Proof Sketch.* We outline the main arguments underlying the target risk bound, highlighting the distinct roles of geometric alignment and statistical complexity.

**Step 1: Decomposition of target risk.**

For any hypothesis $h = g \circ f_\theta$, the target risk admits the decomposition

$$\mathcal{R}_t(h) = \mathcal{R}_{\mathrm{src}}(h) + \big[\mathcal{R}_t(h) - \mathcal{R}_{\mathrm{src}}(h)\big],$$

where the second term captures the effect of domain shift. Controlling this discrepancy is the central challenge in domain generalization.

**Step 2: Geometric control of domain discrepancy (detailed).**

We bound $|\mathcal{R}_t(h|c) - \mathcal{R}_e(h|c)|$ for class $c$ using properties of the class-conditional feature distributions. For features $f \sim \mathcal{F}_{e,c}$ with covariance $\Sigma_{e,c}$, the class-conditional risk depends on the feature distribution's geometry.

Under the $L$-Lipschitz assumption on the loss and bounded features (Assumption 3.8):

$$|\mathcal{R}_t(h|c) - \mathcal{R}_e(h|c)| \leq L \cdot W_2(\mathcal{F}_{t,c}, \mathcal{F}_{e,c}),$$

where $W_2$ denotes the 2-Wasserstein distance.

Under the generative model, the Wasserstein distance between class-conditional distributions can be bounded by their subspace geometry. For zero-mean Gaussians, the 2-Wasserstein distance is given by the Bures metric on covariance matrices: $W_2^2(\mathcal{N}(0, \Sigma_1), \mathcal{N}(0, \Sigma_2)) = \mathrm{tr}(\Sigma_1 + \Sigma_2 - 2(\Sigma_1^{1/2} \Sigma_2 \Sigma_1^{1/2})^{1/2})$. When covariances are concentrated on $k$-dimensional subspaces $U_1, U_2$ with signal strength $\Lambda_z$, this reduces to a function of principal angles. Specifically:

$$W_2(\mathcal{F}_{t,c}, \mathcal{F}_{e,c}) \leq \sqrt{\lambda_{\max}(\Lambda_z)} \cdot d_{\mathrm{Gr}}(U_{t,c}, U_{e,c}) + \delta_{\mathrm{res}},$$

where the first term captures subspace misalignment and $\delta_{\mathrm{res}}$ bounds the residual error outside the principal subspaces (Assumption 3.6).

Aggregating over classes and source domains yields

$$|\mathcal{R}_t(h) - \mathcal{R}_{\mathrm{src}}(h)| \; \leq \; L\sqrt{\lambda_{\max}(\Lambda_z)} \cdot \big(\bar{\Delta}_{\mathrm{geo}}(h) + \rho\big) + L\delta_{\mathrm{res}},$$

where $\bar{\Delta}_{\mathrm{geo}}$ measures normalized source-domain misalignment and $\rho$ bounds the source-to-target geometric discrepancy.

**Step 3: Statistical complexity — two proof routes.**

We establish complexity control via two complementary routes.

**Step 3a (MI-based bound, used for Theorem 3.11).** We apply the information-theoretic generalization framework of Xu & Raginsky (2017) as an alternative to covering number arguments. Let $S = \{(x_i, y_i)\}_{i=1}^{Mn}$ be the training data and $W = \hat{\theta}$ the learned parameters. For $(B_\ell/2)$-sub-Gaussian losses:

$$\mathbb{E}_{S,W}\big[\mathcal{R}_{\mathrm{src}}(h_W) - \widehat{\mathcal{R}}_{\mathrm{src}}(h_W)\big] \leq \sqrt{\frac{B_\ell^2 I(S;W)}{2Mn}}.$$

The key step is bounding $I(S;W)$ via the rate term. By chain rule: $I(S;W) \leq I(S;Z) + I(S;W|Z)$, where $I(S;W|Z) = O(1)$ for regularized training (Pensia et al., 2018). Further decomposing: $I(S;Z) = I(X;Z) + I(Y;Z|X) \leq I(X;Z) + \log|\mathcal{C}|$. The interaction information satisfies $I(X;Z;Y) = I(X;Y) - I(X;Y|Z) \leq H(Y) \leq \log|\mathcal{C}|$. **This looseness only affects the additive constant $C'$, not the rate-dependent term.**

Using the Gaussian approximation for $Z|Y = c$ and partitioning eigenvalues into small ($\sigma^2 \leq 1/\tau$) and large ($\sigma^2 > 1/\tau$) sets:

$$I(S;W) \leq C \cdot \widehat{R}(\hat{\theta}) + C',$$

where $C = 2d(1 + \log(1 + K)/\log(1 + \tau))$ and $C' = d\log(2\pi e/\tau) + 2\log|\mathcal{C}| + O(1)$. The two $\log|\mathcal{C}|$ terms arise from $I(Y; Z|X)$ and the interaction information bound. Substituting this MI bound back yields the $\sqrt{(C\widehat{\bar{R}} + C' + \log(2/\delta))/Mn}$ complexity term in Theorem 3.11, *linear* in $\widehat{\bar{R}}$.

**Step 3b (Rademacher complexity, used for Corollary 3.13).** The rate term in the RDI objective constrains the spectral volume of feature representations. In particular, under the Gaussian approximation, the empirical aggregated rate $\widehat{\bar{R}} = \bar{R}(\mathcal{F})$ (Eq. 14) with per-class rates $R_c = R_\tau(\mathcal{F}_c)$ (Eq. 12) upper bounds the normalized log-determinant of the feature covariance in the high-variance regime, via Hadamard's inequality (see Appendix A.8). The capacity-regularized form provides numerical stability while preserving the essential spectral control property.

To formalize the capacity control, we establish a covering number bound. Let $W$ denote the final linear classifier with $\|W\|_F \leq B$, and assume the feature map $f_\theta$ is $L_f$-Lipschitz.

We first relate the $\|\cdot\|_\infty$-covering number to the $\|\cdot\|_2$-covering number. For a hypothesis class $\mathcal{H}$ of functions mapping to $\mathbb{R}^C$ (with $C$ classes), standard norm equivalence yields

$$\log\mathcal{N}(\epsilon, \mathcal{H}, \|\cdot\|_\infty) \leq \log\mathcal{N}(\epsilon/\sqrt{C}, \mathcal{H}, \|\cdot\|_2).$$

For the $\|\cdot\|_2$-covering number of linear classifiers composed with Lipschitz feature maps, standard covering number arguments for linear classifiers give

$$\log\mathcal{N}(\epsilon/\sqrt{C}, \mathcal{H}, \|\cdot\|_2) \lesssim \frac{C \cdot B^2 L_f^2}{\epsilon^2} \cdot \mathrm{tr}(\Sigma).$$

Combining these two inequalities, and absorbing $C$ into the constant (since $C$ is fixed), we obtain

$$\log\mathcal{N}(\epsilon, \mathcal{H}, \|\cdot\|_\infty) \lesssim \frac{B^2 L_f^2}{\epsilon^2} \cdot \mathrm{tr}(\Sigma).$$

We now bound $\mathrm{tr}(\Sigma)$ in terms of $\widehat{\bar{R}}$. Under Assumption 3.8 (bounded condition number $\kappa(\Sigma) \leq K$),

$$\mathrm{tr}(\Sigma) = \sum_{i=1}^{d} \lambda_i \leq d \cdot \lambda_{\max} = d \cdot \kappa(\Sigma) \cdot \lambda_{\min} \leq d \cdot K \cdot \lambda_{\min}.$$

It remains to bound $\lambda_{\min}$. By the AM-GM inequality applied to eigenvalues,

$$\lambda_{\min} \leq \left(\prod_{i=1}^{d} \lambda_i\right)^{1/d} = (\det\Sigma)^{1/d}.$$

To connect with the capacity-regularized rate $R_\tau$ (Eq. 12), we establish the following bound. For any $x > 0$, we have the identity

$$\log x = \log(1 + \tau x) + \log\frac{x}{1 + \tau x}.$$

Since $\frac{x}{1+\tau x} < \frac{1}{\tau}$ for all $x > 0$, we obtain $\log\frac{x}{1+\tau x} < -\log\tau$. Applying this with $x = \sigma_{c,i}^2$ and summing over coordinates:

$$\sum_{i=1}^{d} \log\sigma_{c,i}^2 < \sum_{i=1}^{d} \log(1 + \tau\sigma_{c,i}^2) - d\log\tau = d\,R_\tau(\mathcal{F}_c) - d\log\tau.$$

Combined with Hadamard's inequality (Eq. 19 in Appendix A.8):

$$\log\det\Sigma_c \leq \sum_{i=1}^{d} \log\sigma_{c,i}^2 \leq d\,R_\tau(\mathcal{F}_c) - d\log\tau.$$

Aggregating via the Rényi-weighted rate (Eq. 14) yields $\log\det\Sigma \lesssim d\widehat{\bar{R}} - d\log\tau$. Therefore,

$$(\det\Sigma)^{1/d} = \exp\big((\log\det\Sigma)/d\big) \leq \frac{1}{\tau}\exp(\widehat{\bar{R}}).$$

Combining these inequalities yields

$$\lambda_{\min} \ \leq \ (\det \Sigma)^{1/d} \ \leq \ \frac{1}{\tau} \exp(\widehat{\overline{R}}),$$

and therefore

$$\mathrm{tr}(\Sigma) \ \leq \ d \cdot K \cdot \lambda_{\min} \ \leq \ \frac{dK}{\tau} \exp(\widehat{\overline{R}}).$$

Plugging into Dudley's entropy integral yields

$$\hat{\mathfrak{R}}_n(\mathcal{H}) = O\left( \sqrt{\frac{\exp(\widehat{\overline{R}}(\hat{\theta}))}{n}} \right).$$

While this dependence on $\widehat{\overline{R}}$ is exponential and thus looser than Step 3a, the route exposes tight $d$-dependence: under bounded rate $\widehat{\overline{R}} = O(1)$ (e.g., via batch normalization), this simplifies to $O(\sqrt{(k \log d)/n})$, giving the effective dimension reduction $d_{\mathrm{eff}} = k + \log d \ll d$ of Corollary 3.13.

**Step 4: Combining the bounds for Theorem 3.11.**

Applying the information-theoretic generalization framework (Xu & Raginsky, 2017) with the bound from Step 3a, and combining with the geometric discrepancy control from Steps 1–2, yields

$$\mathcal{R}_t(\hat{\theta}) \ \leq \ \widehat{\mathcal{R}}_{\mathrm{src}}(\hat{\theta}) + L\sqrt{\lambda_{\max}(\Lambda_z)} \cdot \left( \bar{\Delta}_{\mathrm{geo}}(\hat{\theta}) + \rho \right) + L\delta_{\mathrm{res}} + B_\ell \sqrt{\frac{C \cdot \widehat{\overline{R}}(\hat{\theta}) + C' + \log(2/\delta)}{Mn}},$$

which completes the proof sketch. □

*Remark* A.14 (Comparison between MI and Rademacher routes). The two routes used in our analysis offer complementary strengths: (i) $\widehat{\overline{R}}$-**dependence:** Step 3a (MI-based) yields $\sqrt{\widehat{\overline{R}}/n}$, linear in $\widehat{\overline{R}}$; Step 3b (Rademacher) yields $\sqrt{\exp(\widehat{\overline{R}})/n}$, exponential in $\widehat{\overline{R}}$. (ii) $d$-**dependence:** Step 3a's constant $C = 2d(1 + \log(1 + K)/\log(1 + \tau))$ is linear in $d$ (with logarithmic dependence on condition number $K$ and inverse-logarithmic dependence on $\tau$); Step 3b, exploiting subspace projection, gives $d_{\mathrm{eff}} \ll d$, empirically validated by stable rank measurements (3–5 for ResNet-50 on PACS/OfficeHome) (Ansuini et al., 2019). Under batch normalization, $\widehat{\overline{R}} = O(1)$, the Rademacher route's exponential becomes $O(1)$, allowing both routes to give sub-linear-in-$n$ scaling, with Rademacher providing tighter $d$-dependence.

*Remark* A.15 (Gaussianity as a tractable proxy). We adopt $Z|Y = c \sim \mathcal{N}(\mu_c, \Sigma_c)$ to obtain closed-form entropy expressions. This approximation is motivated by: (i) **Central limit effects:** Deep network features aggregate many nonlinear transformations, inducing approximate Gaussianity (Neal, 1996); (ii) **Self-consistency:** The RDI objective directly penalizes $\log \det \Sigma_c$ via the rate term; (iii) **Robustness:** For sub-Gaussian distributions, differential entropy satisfies the same scaling up to constants.

*Remark* A.16 (On the target-domain discrepancy $\rho$). The quantity $\rho = \max_{e,c} d_{\mathrm{Gr}}(U_{t,c}, U_{e,c})$ involves the unseen target domain and is not directly computable. However, $\rho$ can be bounded empirically via leave-one-domain-out validation: treating each source domain as a held-out "pseudo-target" yields an upper confidence estimate. The bound degrades gracefully with $\rho$; when the target domain is drastically different (e.g., infrared images vs. natural photos), $\rho$ may be large and the bound loose—reflecting the fundamental difficulty of generalizing to distant domains.

## A.12. Proof of Proposition 3.14

**Proposition A.17** (Restated). *Consider an ablated objective obtained by removing the aggregated rate term $\bar{R}(\mathcal{F})$ (Eq. 14) from the RDI objective (Eq. 15), **even when feature norms are bounded** ($\|f\|_2 \leq B$). There exist representations achieving: (i) perfect geometric alignment $\Delta_{\mathrm{geo}} = 0$; (ii) bounded feature norms; yet (iii) arbitrarily ill-conditioned class covariances $\kappa(\Sigma_c) \to \infty$. Consequently, standard norm-based regularization (e.g., weight decay) is insufficient; explicit spectral rate control is necessary.*

*Proof.* We provide an explicit construction demonstrating that geometric alignment combined with bounded feature norms does not control the conditioning of class covariances.

**Construction.** Fix a $k$-dimensional subspace $U \in \mathrm{Gr}(k, d)$ shared across all domains and classes, ensuring $\Delta_{\mathrm{geo}} = 0$. Define features as $\mathbf{f} = U\mathbf{z}$ where $\mathbf{z} = (z_1, \dots, z_k)^\top \in \mathbb{R}^k$ with:

$$z_1 \sim \mathrm{Uniform}[-1, 1], \quad \mathrm{Var}(z_1) = 1/3,$$
$$z_i \sim \mathrm{Uniform}[-\epsilon, \epsilon] \text{ for } i \geq 2, \quad \mathrm{Var}(z_i) = \epsilon^2/3,$$

where $\epsilon > 0$ is a free parameter.

**Verification of conditions.**

*(i) Perfect alignment:* All domains share the same subspace $U$, so $d_{\mathrm{Gr}}(U_{e,c}, U_{e',c}) = 0$ for all $e, e'$, yielding $\Delta_{\mathrm{geo}} = 0$.

*(ii) Bounded norms:* Since $U$ has orthonormal columns,

$$\|\mathbf{f}\|_2 = \|\mathbf{z}\|_2 \leq \sqrt{1 + (k-1)\epsilon^2} < \sqrt{k} =: B.$$

Thus all features satisfy $\|\mathbf{f}\|_2 \leq B$ uniformly.

*(iii) Ill-conditioned covariance:* The covariance in the latent space is

$$\mathrm{Cov}(\mathbf{z}) = \mathrm{diag}\left(\tfrac{1}{3}, \tfrac{\epsilon^2}{3}, \dots, \tfrac{\epsilon^2}{3}\right),$$

with eigenvalues $\lambda_{\max} = 1/3$ and $\lambda_{\min} = \epsilon^2/3$. The condition number is

$$\kappa(\Sigma) = \frac{\lambda_{\max}}{\lambda_{\min}} = \frac{1}{\epsilon^2} \xrightarrow{\epsilon \to 0} \infty.$$

**Geometric interpretation.** The construction places all features in an *extremely flat ellipsoid*: the variance along the $z_1$ direction is $O(1)$ while variances along other directions are $O(\epsilon^2)$. As $\epsilon \to 0$, features concentrate on a lower-dimensional submanifold while remaining norm-bounded.

**Why weight decay is insufficient.** Weight decay constrains classifier weights $\|W\|_F \leq B_W$, but when class covariances are ill-conditioned: (i) signal along small-eigenvalue directions is dominated by noise; (ii) the classifier is forced to amplify these directions with large weights to extract signal; (iii) this amplification makes predictions sensitive to noise, degrading generalization. Formally, for a linear classifier $h(f) = W^\top f$, the signal-to-noise ratio along direction $v$ scales as $\sqrt{\lambda_v}$. When $\lambda_{\min} \to 0$, robust classification becomes impossible regardless of $\|W\|_F$.

**Why spectral rate control is necessary.** The rate term $R_\tau(\mathcal{F}_c) = \frac{1}{d}\sum_i \log(1 + \tau\sigma_{c,i}^2)$ penalizes both excessively large and imbalanced eigenvalues. Minimizing $R_\tau$ while maintaining projection fidelity (via $\Delta_{\mathrm{rec}}$) encourages *balanced spectral profiles* where no eigenvalue is negligibly small. This prevents the pathological ill-conditioning exhibited in the construction above. $\square$

*Remark* A.18 (Comparison with prior constructions). The classical argument shows that *unbounded* variance leads to unbounded Rademacher complexity. Our strengthened construction shows that even with *bounded norms*, geometric alignment alone permits arbitrarily poor conditioning—a subtler failure mode that weight decay cannot address. The rate term $R_\tau$ directly regularizes the eigenspectrum, providing control that neither geometric alignment nor norm constraints can achieve alone.

# B. Synthetic Data Experiments

These synthetic experiments validate the geometric and information-theoretic behavior of RDI under controlled conditions, complementing the real-data evaluations in the main text.

## B.1. Data Generation Protocol

We employ a standard factor model for multi-domain subspace learning, following the probabilistic PCA framework (Tipping & Bishop, 1999):

1. **Ground Truth:** A latent invariant subspace $U^* \in \mathrm{Gr}(k, D)$ is sampled uniformly from the Grassmannian via QR decomposition of a $D \times k$ random Gaussian matrix (Edelman et al., 1998).

2. **Domain Shift:** Each domain $e$ has a rotated subspace $U_e = \exp_{U^*}(\theta_e V_e)$ obtained by geodesic rotation of $U^*$ by angle $\theta_e$ along a random tangent direction $V_e \in T_{U^*}\mathrm{Gr}(k, D)$, following standard subspace perturbation models (Stewart & Sun, 1990).

3. **Observations:** Data are generated as $x = s_e \cdot z U_e^\top + \sigma \epsilon$ where $z \sim \mathcal{N}(0, I_k)$ and $\epsilon \sim \mathcal{N}(0, I_D)$. The scale $s_e$ controls domain-specific signal strength.

This setup provides explicit control over geometric discrepancy (via $\theta_e$) and statistical heterogeneity (via $s_e$), enabling targeted validation of our theoretical claims. Unlike classification-focused benchmarks (Aubin et al., 2021), our protocol directly tests subspace recovery on the Grassmann manifold, which aligns with the geometric nature of RDI.

**Parameter Ranges.** Across all synthetic experiments: $D \in [32, 128]$, $k \in [4, 10]$, $E \in [3, 5]$, $\theta_e \in [0°, 140°]$, $\sigma \in [0.03, 0.1]$, and $n_e \in [500, 2000]$ samples per domain. Lower dimensions are used for grid search experiments; higher dimensions for robustness tests. All results report mean $\pm$ std over 5–10 random seeds.

### B.2. Experiment A: Robustness to Domain Rotation

Figure 4 examines how subspace recovery degrades as domain rotation increases.

**Setup.** Three domains are generated with symmetric rotations $\{-\theta, 0, +\theta\}$ around the ground truth $U^*$, with equal variance across domains. We sweep $\theta_{\max}$ from $0°$ to $60°$.

**Results.** Joint optimization (Ours) maintains consistently higher affinity $\kappa$ with $U^*$ across all rotation angles. The gap widens as $\theta_{\max}$ increases, demonstrating that combining geometric and projection objectives provides robustness to larger domain shifts. Both ablations (w/o $\Delta_{\mathrm{geo}}$ and w/o $\Delta_{\mathrm{rec}}$) degrade more rapidly, confirming that neither objective alone suffices.

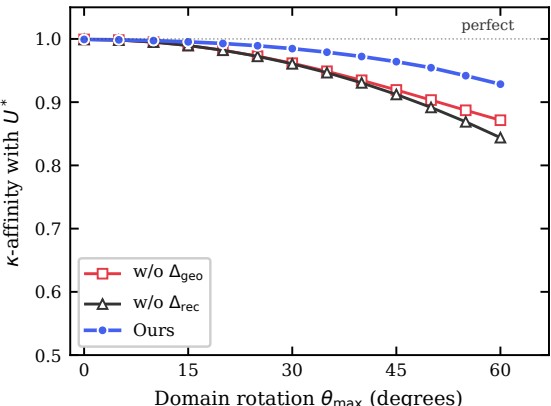

*Figure 4.* **Subspace affinity under increasing domain rotation.** Joint optimization (Ours) consistently recovers subspaces closest to ground truth $U^*$, with the advantage growing as rotation angle increases. Results averaged over 5 random seeds.

### B.3. Experiment B: Conflicting Bias Scenario

Figure 5 presents a more challenging scenario where geometric and statistical biases conflict.

**Setup.** We use $D = 50$, $k = 5$, $E = 3$. In the **symmetric** scenario, domains have equal variance with rotations $\theta_e \in \{-30°, 0°, +30°\}$. In the **asymmetric** scenario, we construct a configuration where geometric and statistical biases conflict:

| Domain | $\theta_e$ | Scale $s_e$ | Samples $n_e$ |
|---|---|---|---|
| 1 (dominant) | $15°$ | 8.0 | 400 |
| 2 | $50°$ | 1.5 | 150 |
| 3 | $75°$ | 1.0 | 100 |

This creates a tension:

- **Pooled PCA** (w/o $\Delta_{\text{geo}}$): Biased toward Domain 1 (high variance, many samples), ignoring geometric consensus.

- **Fréchet mean** (w/o $\Delta_{\text{rec}}$): Treats all domains equally, potentially overfitting to unreliable subspace estimates from Domains 2–3.

**Results.** Panel (a) compares two metrics—geometric distance ($\Delta_{\text{geo}}$) and projection error ($\Delta_{\text{rec}}$)—under symmetric and asymmetric scenarios. In the symmetric case, all methods perform similarly. In the asymmetric (conflicting) case, each ablation excels on its own objective but fails on the other, while Joint achieves balanced performance on both.

Panel (b) visualizes the trade-off curve as $\lambda_{\text{geo}}/\lambda_{\text{rec}}$ varies. The curve traces a Pareto front between geometric alignment and projection fidelity. Neither extreme ($\lambda \to 0$ or $\lambda \to \infty$) achieves the best overall recovery, empirically validating the necessity of balancing both objectives.

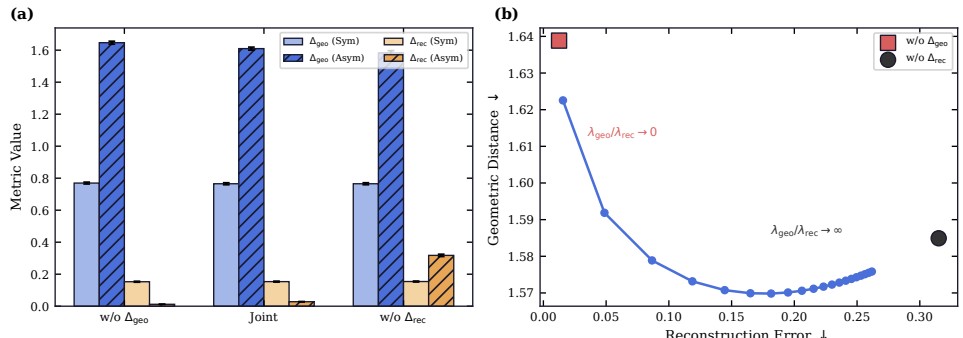

*Figure 5.* **Dual-metric evaluation under conflicting biases.** **(a)** Geometric distance and projection error across methods. Joint optimization avoids the "specialization" trap of ablations. **(b)** Trade-off curve: optimal subspace recovery requires intermediate $\lambda_{\text{geo}}/\lambda_{\text{rec}}$ values.

### B.4. Experiment C: Necessity of Rate Control

Figure 6 validates Proposition 3.14 by demonstrating that successful alignment does not guarantee well-conditioned representations.

**Setup.** We use $D = 32$, $k = 4$, $E = 3$ with rotations $\theta_e \in \{0°, 25°, 50°\}$ and $n_e = 400$ samples per domain. We construct a "hostile" scenario where domain covariances differ substantially despite sharing a common invariant subspace. Specifically, we introduce a hostility parameter $\xi \in [0, 1]$ that controls cross-domain covariance heterogeneity: at $\xi = 0$, all domains share identical covariance structure; at $\xi = 1$, one domain has amplified variance along certain directions (scale factor $s_1 = 5$), creating tension between alignment and spectral health. We compare two methods:

- **w/o $\bar{R}$**: Alignment objective only (geometric distortion minimization).

- **w/ $\bar{R}$ (RDI)**: Full objective with rate control.

**Results.** Panel (a) shows the condition number $\kappa(\Sigma)$ of learned representations. Under friendly conditions, both methods achieve low $\kappa$. However, as hostility increases, w/o $\bar{R}$ suffers spectral collapse ($\kappa \approx 15$), while w/ $\bar{R}$ maintains well-conditioned covariances ($\kappa \approx 1.5$)—a $10\times$ improvement.

Panel (b) shows that both methods achieve comparably low alignment error across all hostility levels. This confirms the core claim of Proposition 3.14: *alignment can succeed while spectral health fails.* Theoretical analysis and empirical evidence thus jointly indicate that rate control is necessary, not merely beneficial, for robust representation learning.

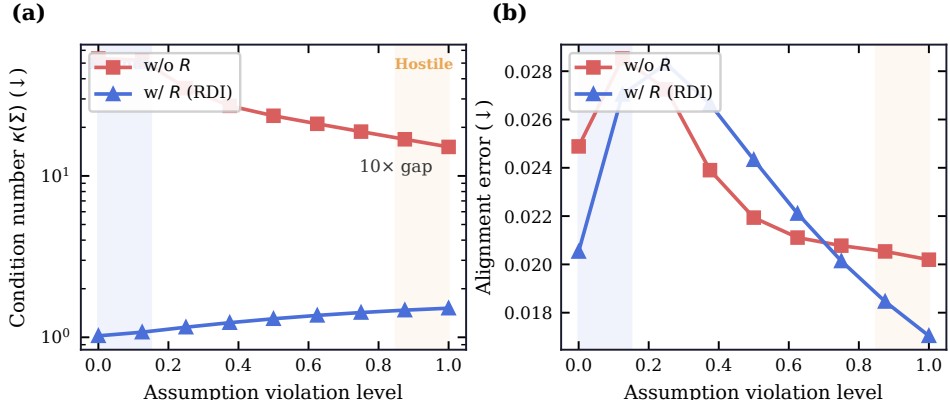

*Figure 6.* **Necessity of rate control under hostile conditions.** **(a)** Condition number $\kappa(\Sigma)$: w/o $\bar{R}$ degrades severely as assumption violations increase, while w/ $\bar{R}$ remains stable. **(b)** Alignment error: both methods succeed, confirming that alignment alone is insufficient. Results averaged over 5 random seeds.

### B.5. Experiment D: Domain Efficiency in Subspace Recovery

Figure 7 examines how the number of source domains $M$ affects invariant subspace recovery, providing direct empirical validation of RDI's domain efficiency.

**Setup.** We use the same protocol as Appendix B.1, sampling source domains $M \in \{2, 3, 4, 5, 6, 8\}$ with controlled rotational shifts around a fixed ground-truth invariant subspace $U^*$. For each $M$, we train ERM, CORAL, and RDI on the $M$ source domains and evaluate the recovered subspace $\hat{U}$ via the Grassmann distance $d_{\mathrm{Gr}}(\hat{U}, U^*)$ (lower is better). Results are averaged over 5 random seeds per configuration.

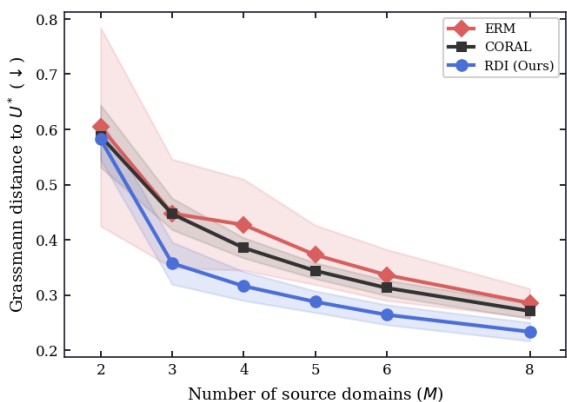

*Figure 7.* **Invariant subspace recovery vs. number of source domains.** Grassmann distance from recovered subspace $\hat{U}$ to ground-truth $U^*$ (lower is better), as a function of the number of source domains $M$. RDI achieves consistently lower Grassmann distance than ERM and CORAL across all $M \geq 3$, with markedly narrower variance bands. Shaded bands show $\pm 1$ standard deviation across 5 random seeds.

**Results.** Three patterns emerge:

- **Consistent recovery quality.** RDI achieves the lowest Grassmann distance across all $M \geq 3$, with the margin over ERM and CORAL preserved as $M$ varies.

- **Domain efficiency.** RDI requires substantially fewer source domains to reach a given recovery quality. RDI with $M = 3$ matches ERM with $M = 5$ to 6, and RDI with $M = 4$ matches CORAL with $M = 5$.

- **Stability.** RDI exhibits the narrowest variance bands across all $M$, indicating that the geometric rate–distortion objective yields more stable subspace estimates than purely statistical alignment (CORAL) or unregularized empirical risk (ERM).

**Interpretation.** This experiment provides direct evidence for the theoretical motivation in Section 3.3: by explicitly minimizing Grassmann distances between class-conditional subspaces, RDI converges to the underlying invariant geometry more efficiently than methods relying on second-order statistical alignment. The domain-efficiency advantage suggests that the geometric formulation extracts shared structure from limited domain diversity more effectively, which is consistent with RDI's strong performance on real-world benchmarks where the number of available source domains is necessarily small.

### B.6. Effective Dimension under Rényi Order Variation

Figure 8 illustrates how the Rényi order $\alpha$ modulates effective dimensionality.

**Definition.** The effective dimension is $d_{\text{eff}}^{(\alpha)} = \exp(H_\alpha)$ where $H_\alpha$ is the Rényi entropy of normalized eigenvalues.

**Interpretation.** As $\alpha$ increases, $d_{\text{eff}}^{(\alpha)}$ decreases monotonically: $\alpha \to 0$ counts support size, $\alpha = 1$ yields perplexity (Shannon), $\alpha = 2$ gives the participation ratio, and $\alpha \to \infty$ reflects max-eigenvalue dominance. This justifies using $\alpha > 1$ in the rate term to emphasize compression of high-variance directions.

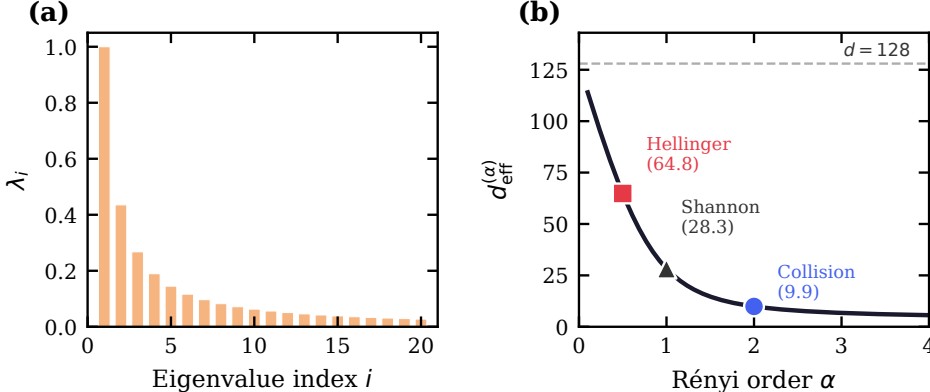

*Figure 8.* **Effective dimension vs. Rényi order.** (a) Power-law eigenvalue spectrum. (b) $d_{\text{eff}}^{(\alpha)}$ transitions from support counting to max-eigenvalue dominance as $\alpha$ increases.

## C. Implementation Details

### C.1. Geometric Projection Head Architecture

As discussed in Section 4.2, we introduce a geometric projection head $g(\cdot)$ to map backbone features into a compact space where our geometric objectives are applied. The projection head consists of a 2-layer MLP. For the ResNet-50 backbone used in our main experiments:

$$g(f) = \text{BN}\big(\text{Linear}_{1024 \to 512}\big(\text{Dropout}_{0.3}\big(\text{ReLU}\big(\text{BN}\big(\text{Linear}_{2048 \to 1024}(f)\big)\big)\big)\big)\big),$$

where BN denotes batch normalization. For other backbones, the intermediate and output dimensions are scaled proportionally to the backbone's feature dimension $d$. This architecture is lightweight ($\sim$2.6M parameters) compared to the ResNet-50 backbone ($\sim$23M parameters).

*Table 4*. **Hyperparameter configuration for RDI (ResNet-50 backbone).** Search ranges and fixed values used in main experiments with ResNet-50. All selections follow the DomainBed training-domain validation protocol.

| Hyperparameter | Symbol | Search Range | Default / Notes |
|---|---|---|---|
| *Geometric projection head $g(\cdot)$* | | | |
| Hidden dimension | – | – | 1024 (fixed) |
| Output dimension | $d_z$ | – | 512 (fixed) |
| Dropout rate | – | – | 0.3 (fixed) |
| *RDI-specific hyperparameters* | | | |
| Subspace dimension | $k$ | $\{8, 16, 32, 64, 128\}$ | – |
| Overall RDI loss weight | $\gamma$ | – | 0.5 (fixed) |
| Geometric distortion weight | $\lambda_{\mathrm{geo}}$ | $\{0.1, 0.2, 0.3, 0.4, 0.5, 0.6\}$ | – |
| Projection distortion weight | $\lambda_{\mathrm{rec}}$ | $\{0.1, 0.2, 0.3, 0.4, 0.5, 0.6\}$ | – |
| Capacity parameter | $\tau$ | – | 1.0 (fixed) |
| Rényi order | $\alpha$ | $\{0.5, 1.0, 2.0\}$ | – |
| Warm-up iterations | $T_w$ | – | 500 (fixed) |
| Rate term weight | $\beta$ | $10^{\mathrm{uniform}(-1,0)}$ | – |
| EMA momentum | $\mu_{\mathrm{EMA}}$ | $\{0.85, 0.90, 0.95\}$ | – |
| *Training hyperparameters* | | | |
| Learning rate | $\eta$ | $10^{\mathrm{uniform}(-5,-3.5)}$ | – |
| Batch size per domain | – | $2^{\mathrm{uniform}(3,7)}$ | – |
| Weight decay | $\lambda_{\mathrm{wd}}$ | $10^{\mathrm{uniform}(-5.0,-2.5)}$ | – |
| Dropout | – | $\{0, 0.1, 0.5\}$ | – |
| Total iterations | $T_{\max}$ | – | 5000 (15000 for DomainNet) |

**Design rationale.** The projection head is a *statistical prerequisite* for the class-conditional spectral analysis at the core of RDI, not merely an architectural choice for added capacity. A non-degenerate empirical class-conditional covariance $\widehat{\Sigma}_{e,c} \in \mathbb{R}^{d \times d}$ requires at least $n_{e,c} \geq d$ samples to be full rank, and well-conditioned spectral estimation typically requires $n_{e,c} \geq 10\,d$. Modern deep backbones produce feature dimensions in this challenging regime: $d = 384$ for DeiT-Small, 768 for ConvNeXt-Tiny, and 2048 for ResNet-50, all of which we evaluate (Appendix D.1). For *any* of these backbones, the well-conditioned threshold demands thousands of samples per (domain, class) bucket. **No standard DG benchmark comes close.** The average per-bucket sample count across DomainBed datasets ranges from roughly 60 (OfficeHome: 15,588 images / 4 domains / 65 classes) to roughly 620 (TerraIncognita), with PACS ($\sim$357), DomainNet ($\sim$283), and VLCS ($\sim$536) in between. Against ResNet-50 ($d = 2048$), all benchmarks fall short of even the bare-rank threshold; against the smallest backbone we evaluate (DeiT-Small, $d = 384$), only TerraIncognita marginally exceeds it, and all fall short of the well-conditioned threshold by an order of magnitude or more. Consequently, the empirical covariance is severely rank-deficient: at most $\min(n_{e,c}, d)$ of $d$ possible directions carry signal, with the remaining dimensions reflecting estimation noise. Two consequences follow: (i) the top-$k$ eigenspace estimate is dominated by sampling noise rather than signal, destabilizing the Grassmannian alignment objective; and (ii) the rate term $R_\tau(\mathcal{F}_c) = \frac{1}{d} \sum_i \log(1 + \tau\sigma_{c,i}^2 + \varepsilon)$ is overwhelmingly dominated by the $d - \mathrm{rank}(\widehat{\Sigma}_{e,c})$ noise dimensions. Even shrinkage regularization cannot resolve this—it improves conditioning, but cannot recover signal directions that were never sampled. We address this by applying a two-layer MLP projection head that progressively reduces feature dimensionality (intermediate and output dimensions scale with each backbone). This substantially improves the sample-to-dimension ratio at the layer where the rate–distortion objectives operate, partially mitigating the rank deficiency while preserving each backbone's native feature scale. To verify that the projection head alone does not explain RDI's gains, we include an ERM + $g(\cdot)$ baseline in our ablation study (Table 2), which uses identical architecture but disables both rate and distortion losses; further controlled experiments across multiple baselines are in Appendix D.2.

## C.2. Hyperparameter Configuration

Table 4 summarizes all hyperparameters used in RDI, including their search ranges and fixed values. We adopt the DomainBed protocol for hyperparameter selection: all choices are made using training-domain validation only, with no access to target domain data.

**Search protocol.** We follow the standard DomainBed random search protocol (Gulrajani & Lopez-Paz, 2021). For each dataset and test environment, we sample 20 hyperparameter configurations uniformly from the search ranges specified in Table 4. Model selection uses **training-domain validation**: we hold out 20% of each source domain as validation data and select the configuration with the highest average validation accuracy across source domains. The target domain is never accessed during hyperparameter selection.

**Selected hyperparameters.** Table 5 reports the hyperparameters selected by the above protocol for each dataset. For datasets with multiple test environments, we report the most frequently selected value across splits; when values vary substantially across environments, we indicate the range.

*Table 5.* **Selected hyperparameters for each dataset.** Following the DomainBed protocol, hyperparameters are tuned for each test environment using the **training-domain validation set** (a random 20% split of each source domain). We report the **mode** (most frequent configuration) across these splits.

| Hyperparameter | PACS | VLCS | OfficeHome | TerraInc | DomainNet |
|---|---|---|---|---|---|
| Learning rate $\eta$ | 4e-5 | 2e-5 | 1e-5 | 3e-5 | 5e-5 |
| Weight decay $\lambda_{\mathrm{wd}}$ | 0.01 | 3e-5 | 3e-4 | 2e-4 | 0.0 |
| Batch size | 32 | 29 | 25 | 37 | 32 |
| Subspace dim $k$ | 32 | 32 | 32 | 32 | 64 |
| $\lambda_{\mathrm{geo}}$ | 0.4 | 0.2 | 0.3 | 0.4 | 0.3 |
| $\lambda_{\mathrm{rec}}$ | 0.4 | 0.3 | 0.3 | 0.6 | 0.3 |
| $\tau$ | 2.0 | 1.0 | 1.0 | 2.0 | 2.0 |
| $\alpha$ | 0.5 | 2.0 | 2.0 | 2.0 | 0.5 |
| $\mu_{\mathrm{EMA}}$ | 0.90 | 0.85 | 0.85 | 0.85 | 0.90 |

**Dataset-specific notes.** For DomainNet, we extend training to 15,000 iterations and use $k = 64$ to accommodate the larger number of classes (345 classes). All other datasets use the default 5,000 iterations.

### C.3. Hyperparameter Sensitivity Analysis

We analyze the sensitivity of RDI to its key hyperparameters on PACS. Since optimal hyperparameters may vary across test environments due to different domain shift characteristics, the **"Selected"** row represents the configuration chosen by the validation protocol for each split (averaged across all four splits, achieving 87.44%). We compare this adaptive baseline against fixed parameter values applied uniformly across all environments.

**Sensitivity Results.** Table 6 summarizes the results.

- **Stability:** RDI remains robust across a wide range of $k$, $\lambda_{\mathrm{rec}}$, $\lambda_{\mathrm{geo}}$, and $\tau$, with most variations staying within $1.5\%$ of the selected baseline.

- **Potential Headroom:** Notably, fixed $\tau = 1.0$ yields slightly higher accuracy (**87.62%**, $+0.18\%$) compared to the validation-selected average, suggesting further tuning potential.

**Implementation robustness.** EMA momentum $\mu_{\mathrm{EMA}}$ for center updates shows negligible sensitivity: both 0.85 and 0.95 achieve $\approx 87.25\%$ (deviation $< 0.05\%$), confirming that this engineering choice does not significantly affect performance.

**Rényi Order $\alpha$.** Table 7 reports per-environment sensitivity to the Rényi order $\alpha$, which parametrizes the $\alpha$-geometric reweighting over class-conditional rate terms.

The most consistent finding is that $\alpha = 1.0$ **(uniform aggregation) is suboptimal across all four test environments**, with performance drops ranging from $0.5\%$ to $2.8\%$ compared to the environment-specific optimum. This validates the utility of non-uniform $\alpha$-escort weighting in the Rényi-based rate aggregation.

*Table 6.* **Sensitivity to core hyperparameters on PACS (averaged across all four splits).** "Selected" denotes the per-split validation-chosen configuration. Although $\tau$ is fixed at 1 throughout, we report sensitivity over $\{1.0, 2.0, 5.0\}$ as robustness evidence.

| Parameter | Value | Avg Acc (%) | Std (%) | $\Delta$ (%) |
|---|---|---|---|---|
| ***Reference: Validation Selected*** | | **87.44** | 0.90 | — |
| $k$ (subspace dim.) | 8 | 86.50 | 0.80 | $-0.94$ |
| | 16 | 85.38 | 0.90 | $-2.06$ |
| | 32 | 87.44 | 0.90 | — |
| | 64 | 86.96 | 0.60 | $-0.48$ |
| | 128 | 86.35 | 0.80 | $-1.09$ |
| $\lambda_{\mathrm{rec}}$ (recon.) | 0.1 | 86.32 | 0.78 | $-1.12$ |
| | 0.2 | 85.91 | 0.74 | $-1.53$ |
| | 0.3 | 87.05 | 0.25 | $-0.39$ |
| | 0.4 | 86.69 | 0.70 | $-0.75$ |
| | 0.5 | 86.54 | 0.51 | $-0.90$ |
| | 0.6 | 86.42 | 0.66 | $-1.02$ |
| $\lambda_{\mathrm{geo}}$ (geo.) | 0.1 | 85.97 | 0.81 | $-1.47$ |
| | 0.2 | 86.86 | 0.58 | $-0.58$ |
| | 0.3 | **87.62** | 0.53 | $+0.18$ |
| | 0.4 | 87.47 | 0.55 | $+0.03$ |
| | 0.5 | 85.84 | 0.63 | $-1.60$ |
| | 0.6 | 86.43 | 0.57 | $-1.01$ |
| $\tau$ (capacity) | 1.0 | **87.62** | 0.57 | $+0.18$ |
| | 2.0 | 87.44 | 0.90 | — |
| | 5.0 | 87.35 | 0.57 | $-0.09$ |

The optimal $\alpha$ exhibits environment dependence: Art Painting, Photo, and Sketch favor $\alpha = 0.5$, while Cartoon benefits from $\alpha = 2.0$. This domain-specific preference may reflect differences in class-conditional geometric structure across target distributions, suggesting that adaptive $\alpha$ selection via validation is preferable to a fixed universal choice.

*Table 7.* Sensitivity to Rényi order $\alpha$ on PACS. The parameter $\alpha$ governs $\alpha$-geometric reweighting over class-conditional rate terms. Bold indicates the best result per environment.

| $\alpha$ | Art Painting | Cartoon | Photo | Sketch |
|---|---|---|---|---|
| 0.5 | **88.44**$_{\pm 1.00}$ | 80.72$_{\pm 0.77}$ | **97.68**$_{\pm 0.22}$ | **82.36**$_{\pm 1.06}$ |
| 1.0 | 87.86$_{\pm 0.82}$ | 80.19$_{\pm 1.19}$ | 97.40$_{\pm 0.28}$ | 79.61$_{\pm 2.00}$ |
| 2.0 | 87.63$_{\pm 0.91}$ | **81.53**$_{\pm 0.98}$ | 97.65$_{\pm 0.30}$ | 79.58$_{\pm 2.33}$ |

## C.4. Computational Overhead

Table 8 summarizes the computational cost of RDI relative to a standard ERM baseline.

*Table 8.* **Computational overhead of RDI.** Training time and GPU memory compared to ERM baseline. All experiments conducted on a single NVIDIA RTX 5090 GPU.

| Dataset | Training Time | | | GPU Memory | | |
|---|---|---|---|---|---|---|
| | ERM | RDI | Ratio | ERM | RDI | Ratio |
| PACS | 12 min | 20 min | $1.66\times$ | 10.1 GB | 10.2 GB | $1.01\times$ |
| OfficeHome | 15 min | 23 min | $1.47\times$ | 11.6 GB | 12.0 GB | $1.03\times$ |
| DomainNet | 256 min | 302 min | $1.18\times$ | 16.3 GB | 17.0 GB | $1.04\times$ |

**Training complexity.** The training overhead of RDI arises from three geometric operations:

- **Subspace estimation:** For each class with $N$ samples, computing the principal subspace requires eigendecomposition of either the Gram matrix ($N < D$) or the covariance matrix ($N \geq D$), yielding complexity $O(\min(N,D)^3 + ND \cdot \min(N,D))$.

- **Grassmannian distance:** Computing the geodesic distance between two $k$-dimensional subspaces in $\mathbb{R}^D$ requires an SVD of a $k \times k$ matrix, with complexity $O(Dk^2 + k^3) \approx O(Dk^2)$ since $k \ll D$.

- **center updates:** For $M$ source domains, constructing the pairwise distance matrix and updating class centers costs $O(M^2 Dk^2 + Dk^2)$ per class.

- Additionally, the rate constraint requires $O(BD)$ operations per batch for within-class variance computation via vectorized scatter, which is dominated by the geometric operations above.

In practice, these operations are amortized over mini-batches and executed via vectorized primitives (`torch.vmap`, `torch.scatter`), resulting in wall-clock overhead of $1.2\times$–$1.6\times$ relative to ERM (see Table 8). Memory overhead remains minimal ($< 5\%$) as we maintain only $O(Dk)$ parameters per class for subspace storage.

**Inference complexity.** At inference time, RDI uses a standard linear classifier on the projected features $z = g(f(x))$. The additional overhead from the projection head $g(\cdot)$ is negligible: for a 2-layer MLP with dimensions $2048 \rightarrow 1024 \rightarrow 512$, this amounts to approximately 1.5 MFLOPs per sample—less than $0.04\%$ of the ResNet-50 backbone ($\sim 4$ GFLOPs). Consequently, RDI achieves inference latency indistinguishable from standard ERM.

# D. Additional Experiments

### D.1. Cross-Architecture Results

**Motivation.** The main results (Table 1) use ResNet-50 as the backbone, which is the DomainBed standard. A natural concern is whether RDI's geometric mechanism generalizes beyond this convolutional baseline. To verify architecture-agnostic effectiveness, we evaluate RDI on two additional backbone families: a Vision Transformer (DeiT-Small) and a modern CNN (ConvNeXt-Tiny). Both experiments follow the DomainBed training-domain validation protocol with 3 random seeds.

**Results on DeiT-Small.** Table 9 reports four-dataset results with DeiT-Small. RDI achieves the highest average on all four datasets, outperforming both ERM and SDViT (a recent ViT-tailored DG method). Notably, the gain on PACS ($87.3\%$ vs. ERM's $83.9\%$) closely matches that of ResNet-50 ($87.4\%$ vs. $85.5\%$), confirming that the geometric rate–distortion mechanism transfers across architecture families despite their distinct inductive biases.

*Table 9.* **ViT Results:** Accuracy (%) with DeiT-Small backbone under training-domain validation (3 seeds). RDI achieves the highest average on all four datasets, confirming architecture-agnostic effectiveness.

(a) PACS

| Method | Art | Cartoon | Photo | Sketch | Avg. |
|--------|-----|---------|-------|--------|------|
| ERM | $87.2_{\pm1.4}$ | $80.1_{\pm1.0}$ | $98.7_{\pm0.2}$ | $69.6_{\pm1.0}$ | 83.9 |
| SDViT | $87.6_{\pm0.3}$ | $81.3_{\pm0.4}$ | $98.0_{\pm0.3}$ | $75.8_{\pm1.0}$ | 85.7 |
| **RDI** | $\mathbf{89.6}_{\pm1.5}$ | $\mathbf{84.4}_{\pm0.6}$ | $\mathbf{98.8}_{\pm0.3}$ | $\mathbf{76.4}_{\pm2.6}$ | **87.3** |

(b) OfficeHome

| Method | Art | Clipart | Product | Real | Avg. |
|--------|-----|---------|---------|------|------|
| ERM | $67.7_{\pm0.3}$ | $55.2_{\pm0.4}$ | $79.5_{\pm0.4}$ | $81.7_{\pm0.1}$ | 71.0 |
| SDViT | $68.3_{\pm0.8}$ | $56.3_{\pm0.2}$ | $79.5_{\pm0.3}$ | $81.8_{\pm0.1}$ | 71.5 |
| **RDI** | $\mathbf{69.6}_{\pm0.4}$ | $\mathbf{57.2}_{\pm0.2}$ | $\mathbf{80.7}_{\pm0.1}$ | $\mathbf{82.8}_{\pm0.1}$ | **72.6** |

(c) VLCS

| Method | Caltech | LabelMe | SUN09 | VOC2007 | Avg. |
|--------|---------|---------|-------|---------|------|
| ERM | $98.1_{\pm0.7}$ | $62.7_{\pm1.0}$ | $72.9_{\pm1.7}$ | $77.6_{\pm0.6}$ | 77.8 |
| SDViT | $96.8_{\pm0.5}$ | $63.2_{\pm0.8}$ | $\mathbf{76.2}_{\pm0.4}$ | $78.3_{\pm0.4}$ | 78.6 |
| **RDI** | $\mathbf{98.3}_{\pm0.6}$ | $\mathbf{64.3}_{\pm0.6}$ | $75.7_{\pm0.3}$ | $\mathbf{78.8}_{\pm1.0}$ | **79.3** |

(d) TerraIncognita

| Method | L100 | L38 | L43 | L46 | Avg. |
|--------|------|-----|-----|-----|------|
| ERM | $52.8_{\pm2.9}$ | $28.8_{\pm0.8}$ | $50.7_{\pm0.7}$ | $37.1_{\pm2.7}$ | 42.4 |
| SDViT | $\mathbf{54.2}_{\pm1.7}$ | $29.1_{\pm2.6}$ | $52.2_{\pm0.3}$ | $37.4_{\pm0.6}$ | 43.2 |
| **RDI** | $53.0_{\pm1.5}$ | $\mathbf{29.5}_{\pm1.2}$ | $\mathbf{54.4}_{\pm1.2}$ | $\mathbf{37.6}_{\pm1.3}$ | **43.6** |

**Results on ConvNeXt-Tiny.** Table 10 reports PACS results with ConvNeXt-Tiny, a representative modern CNN. Both methods use DomainBed's default hyperparameters without ConvNeXt-specific tuning. RDI achieves 91.4% average accuracy, a +6.4% improvement over ERM (85.0%), with consistent gains across all four environments. Combined with the DeiT-Small results above, the cross-architecture experiments demonstrate that RDI's gains transfer across CNN (ResNet-50), Transformer (DeiT-Small), and modern CNN (ConvNeXt-Tiny) architectures.

*Table 10.* **ConvNeXt Results:** Accuracy (%) with ConvNeXt-Tiny backbone on PACS under training-domain validation (3 seeds). RDI's gains transfer to modern CNN architectures.

| Method | Art | Cartoon | Photo | Sketch | Avg. |
|--------|-----|---------|-------|--------|------|
| ERM | $83.8_{\pm3.5}$ | $79.1_{\pm4.7}$ | $97.1_{\pm0.2}$ | $80.0_{\pm2.8}$ | 85.0 |
| **RDI** | $\mathbf{93.6}_{\pm1.4}$ | $\mathbf{87.5}_{\pm1.0}$ | $\mathbf{99.6}_{\pm0.1}$ | $\mathbf{84.8}_{\pm1.8}$ | **91.4** |

**Summary.** The cross-architecture experiments confirm that RDI's geometric rate–distortion mechanism is architecture-agnostic: it depends only on feature distributions rather than specific architectural inductive biases. This generality is consistent with our theoretical formulation (Section 3.3), which makes no architecture-specific assumptions.

### D.2. Projection Head Fairness

**Motivation.** RDI adds a projection head $g$ on top of the feature extractor $f_\theta$. A natural concern is whether the gains attributed to RDI's geometric objectives might instead come from the projection head itself, which provides additional capacity and dimensionality reduction. To isolate this effect, we re-evaluate four representative baselines (ERM, CORAL, VREx, IB_ERM) with and without an identical projection head, under DomainBed default hyperparameters on PACS (3

seeds, training-domain validation). The projection head is the sole variable.

*Table 11.* **Projection head fairness on PACS.** All methods use DomainBed default hyperparameters and the same projection head $g$ (when applicable); no method-specific tuning. $\Delta$ denotes the accuracy change introduced by adding $g$.

| Method | Art | Cartoon | Photo | Sketch | Avg. | $\Delta$ |
|---|---|---|---|---|---|---|
| ERM | $82.84_{\pm1.2}$ | $78.64_{\pm1.2}$ | $96.48_{\pm1.1}$ | $78.94_{\pm2.6}$ | 84.23 | — |
| ERM $+ g$ | $82.69_{\pm2.8}$ | $75.75_{\pm1.9}$ | $96.73_{\pm0.9}$ | $78.01_{\pm1.6}$ | 83.30 | $-0.93$ |
| CORAL | $83.93_{\pm2.3}$ | $78.45_{\pm2.5}$ | $94.61_{\pm0.8}$ | $76.53_{\pm2.5}$ | 83.38 | — |
| CORAL $+ g$ | $83.08_{\pm0.9}$ | $77.11_{\pm2.3}$ | $95.66_{\pm0.4}$ | $76.07_{\pm2.0}$ | 82.98 | $-0.40$ |
| VREx | $82.79_{\pm2.6}$ | $76.21_{\pm3.6}$ | $96.21_{\pm1.1}$ | $76.53_{\pm2.6}$ | 82.93 | — |
| VREx $+ g$ | $84.06_{\pm1.4}$ | $76.88_{\pm3.5}$ | $96.88_{\pm0.5}$ | $76.68_{\pm3.0}$ | 83.62 | $+0.69$ |
| IB_ERM | $83.71_{\pm1.3}$ | $77.88_{\pm2.4}$ | $96.03_{\pm0.8}$ | $73.44_{\pm3.1}$ | 82.77 | — |
| IB_ERM $+ g$ | $81.74_{\pm3.9}$ | $72.26_{\pm2.8}$ | $97.21_{\pm0.4}$ | $71.24_{\pm0.5}$ | 80.61 | $-2.16$ |
| **RDI** | $\mathbf{86.88_{\pm0.4}}$ | $\mathbf{81.21_{\pm1.7}}$ | $\mathbf{97.19_{\pm0.1}}$ | $\mathbf{79.61_{\pm0.9}}$ | **86.22** | — |

**Findings.** Across four methods under identical conditions, $\Delta$ ranges from $-2.16\%$ to $+0.69\%$—the projection head provides no consistent or universal boost. Some methods benefit slightly (VREx $+0.69\%$), others degrade (ERM $-0.93\%$, CORAL $-0.40\%$, IB_ERM $-2.16\%$). RDI uses the same defaults, the same projection head, and no method-specific tuning, yet outperforms the best vanilla baseline (ERM, $84.23\%$) by $+1.99\%$ and the best $+g$ variant (VREx$+g$, $83.62\%$) by $+2.60\%$. The RDI advantage therefore cannot be attributed to the projection head itself.

**Why does the effect vary across methods?** A projection head performs dimensionality reduction on the feature space. Xue et al. (2024) show theoretically and empirically that the benefit of a projection head is not universal but depends on the interaction between the training objective, data distribution, and the feature weighting in the projected space. Our results are consistent with this finding: different methods respond differently to the same projection head. For IB_ERM, which already compresses representations via an information bottleneck, the additional projection head may discard task-relevant directions ($\Delta = -2.16\%$). For RDI, whose rate–distortion objective explicitly shapes the spectral structure of the projected subspace, the projection head is *actively optimized* as part of the method rather than passively imposed.

**Methodology note.** All methods—including RDI—use identical DomainBed default hyperparameters with no method-specific tuning of any kind. This is essential by design: tuning any method or its $+g$ variant would conflate the projection head's effect with hyperparameter optimization, defeating the purpose of isolating $g$ as the sole variable. If the projection head required method-specific tuning to yield gains, this would itself confirm that it is not a universal architectural boost.

### D.3. Robustness Under Data Scarcity

**Motivation.** A natural question is whether RDI's geometric mechanism remains effective when training data is limited, i.e., when class-conditional covariance estimates become noisier. To examine this, we evaluate RDI, ERM, and CORAL on PACS and OfficeHome with $25\%$ and $50\%$ of training samples per class per domain, using DomainBed training-domain validation (3 seeds).

*Table 12.* **Low-data results on OfficeHome.** Accuracy (%) under reduced training data ratios.

| Ratio | Method | Art | Clipart | Product | Real | Avg. |
|---|---|---|---|---|---|---|
| 0.50 | **RDI** | $\mathbf{63.1_{\pm0.1}}$ | $\mathbf{54.9_{\pm0.3}}$ | $\mathbf{75.6_{\pm0.2}}$ | $\mathbf{78.6_{\pm0.3}}$ | **68.0** |
| | ERM | $60.3_{\pm0.4}$ | $52.0_{\pm0.2}$ | $74.4_{\pm0.7}$ | $75.6_{\pm0.3}$ | 65.6 |
| | CORAL | $56.4_{\pm0.9}$ | $53.2_{\pm1.0}$ | $73.5_{\pm0.4}$ | $73.7_{\pm1.0}$ | 64.2 |
| 0.25 | **RDI** | $\mathbf{62.0_{\pm0.6}}$ | $\mathbf{51.9_{\pm0.5}}$ | $\mathbf{71.8_{\pm0.9}}$ | $\mathbf{75.8_{\pm0.2}}$ | **65.4** |
| | ERM | $58.5_{\pm0.3}$ | $47.5_{\pm1.2}$ | $70.5_{\pm0.1}$ | $72.7_{\pm0.1}$ | 62.3 |
| | CORAL | $57.9_{\pm0.3}$ | $48.6_{\pm0.6}$ | $69.0_{\pm0.6}$ | $70.8_{\pm1.8}$ | 61.6 |

*Table 13.* **Low-data results on PACS.** Accuracy (%) under reduced training data ratios.

| Ratio | Method | Art | Cartoon | Photo | Sketch | Avg. |
|-------|--------|-----|---------|-------|--------|------|
| | **RDI** | $85.8_{\pm1.2}$ | $80.3_{\pm0.9}$ | $97.4_{\pm0.2}$ | $80.7_{\pm0.7}$ | **86.05** |
| 0.50 | ERM | $81.6_{\pm2.8}$ | $74.6_{\pm3.5}$ | $95.7_{\pm0.5}$ | $69.8_{\pm0.8}$ | 80.42 |
| | CORAL | $78.8_{\pm0.9}$ | $78.1_{\pm0.1}$ | $95.0_{\pm0.3}$ | $77.4_{\pm1.8}$ | 82.33 |
| | **RDI** | $84.2_{\pm0.9}$ | $77.4_{\pm1.4}$ | $96.5_{\pm0.4}$ | $76.0_{\pm2.2}$ | **83.52** |
| 0.25 | ERM | $85.9_{\pm1.0}$ | $68.7_{\pm2.1}$ | $95.2_{\pm0.5}$ | $67.5_{\pm4.6}$ | 79.32 |
| | CORAL | $78.5_{\pm0.4}$ | $73.9_{\pm3.5}$ | $92.3_{\pm0.4}$ | $64.7_{\pm1.0}$ | 77.34 |

**Findings.** RDI achieves consistent gains over ERM and CORAL across both benchmarks and both data ratios. On OfficeHome, RDI outperforms ERM by $+2.4\%$ at 50% ratio and $+3.1\%$ at 25%; on PACS, by $+5.6\%$ and $+4.2\%$ respectively. Notably, CORAL, which relies on cross-domain covariance alignment without spectral control, drops below ERM at the 25% ratio on both datasets ($-0.7\%$ on OfficeHome, $-1.98\%$ on PACS), demonstrating a consistent failure mode of second-order alignment under data scarcity.

**Mechanism.** This divergence reflects a key design difference between RDI and CORAL, two methods that both operate on second-order statistics. CORAL requires reliable cross-domain covariance estimation, which degrades rapidly under small samples; its alignment target becomes noisy and potentially harmful. In contrast, RDI's rate term penalizes spectral volume, suppressing variance dispersion and concentrating representations into a compact spectral region, which acts as implicit regularization against overfitting. The consistent pattern across PACS and OfficeHome indicates that as sample size shrinks, CORAL's covariance estimates deteriorate and its performance drops below ERM, while RDI's spectral compression maintains stable gains. We note that these limitations are inherent to any method operating on class-conditional statistics; the relative robustness of RDI compared to CORAL underscores the stabilizing role of the rate term in low-data regimes.

### D.4. Robustness Under Class Imbalance

**Built-in mechanism.** RDI's Rényi-aggregated rate term provides a built-in mechanism that naturally addresses class imbalance. As shown in Section 3.2.3, the escort distribution $\tilde{p}_c^{(\alpha)} \propto n_c^{1-\alpha}$ reduces to *inverse-frequency weighting* at $\alpha = 2$. In an imbalanced setting, minority classes inherently become the most geometrically vulnerable due to noisier covariance estimation from scarce samples. The escort weighting at $\alpha = 2$ therefore automatically upweights these fragile minority classes in the rate regularization, without requiring explicit class-frequency information.

**Empirical evidence on TerraIncognita.** TerraIncognita serves as a natural testbed for this property, as its camera-trap collection protocol produces severe class imbalance across locations. Under this setting, RDI achieves the best average accuracy ($48.7\%$) among all compared methods (Table 22), and notably minimizes the cross-environment performance range: **15.6 pp** (RDI), versus **25.7 pp** (Mixup) and **21.2 pp** (ERM). This stability under cross-location shift, combined with the highest average accuracy, indicates that the Rényi-weighted rate term effectively prevents minority-class representations from being neglected in any single environment.

**Inherent limitations.** We emphasize that these robustness properties do not eliminate the fundamental challenges of class imbalance. Methods operating on class-conditional statistics, including RDI, all share a vulnerability: minority classes contribute fewer samples per batch, potentially making their eigendecomposition noisy and geometric alignment less reliable. In extreme cases, a minority class may not appear in every batch, causing its EMA statistics to update infrequently. The Rényi reweighting partly mitigates but cannot fully resolve such structural limitations. A dedicated treatment of severe class imbalance, e.g., combining RDI with sample-level rebalancing techniques, is left for future work.

### D.5. Representational Analysis: Validating the Rate-Distortion Mechanism

**Motivation and research question.** The RDI objective (Eq. 15) jointly optimizes two geometrically orthogonal components: the distortion term $\Delta$ controls cross-domain subspace alignment, while the rate term $\Phi$ regulates the *spectral volume* within each subspace. To empirically validate this mechanism, we ask: *How does RDI's joint rate-distortion optimization manifest in learned representations?*

We examine two complementary aspects corresponding to the two RDI components:

- **Cross-domain consistency of class-discriminative structure** (related to the distortion term $\Delta_{\mathrm{geo}}$): measured via Centered Kernel Alignment (CKA) of class prototypes.

- **Spectral conditioning of feature distributions** (related to the rate term $R_\tau$): measured via Stable Rank, Condition Number, and Participation Ratio.

Finally, we visualize the optimization trajectory in the $(R, \Delta)$ plane to verify convergence toward the Pareto front of the rate-distortion trade-off.

### D.5.1. Representational Similarity Analysis (CKA)

**Relation to RDI distortion term.** The geometric distortion $\Delta_{\mathrm{geo}}$ (Eq. 8) minimizes cross-domain Grassmannian distances between class-conditional subspaces. At the prototype level, this should manifest as consistent class-center arrangements across domains—if subspaces align, the centroids within them should also exhibit similar geometric relationships. Centered Kernel Alignment (CKA)[1] directly measures this property.

**Methodology.** We define two CKA variants to capture different aspects of cross-domain representational consistency:

- **Prototype CKA**: Measures similarity between class prototype layouts across domains. Higher values indicate that class centers maintain consistent geometric arrangements regardless of domain.

- **Covariance CKA**: Measures similarity between feature covariance structures across domains. Values near 1.0 indicate dimensional collapse (all features concentrated in a single direction), while moderate values suggest healthy feature diversity.

We report Test-Train CKA, which measures representational similarity between the held-out test domain and training domains—directly relevant to generalization performance. We compare three feature extractors:

- **Untrained**: Randomly initialized ResNet-50 (baseline for dimensional collapse);

- **ERM**: Standard empirical risk minimization on source domains;

- **RDI (Ours)**: Our proposed method with geometric distortion + rate control.

**Results on PACS.** Table 14 presents the CKA analysis on PACS. The Untrained baseline exhibits extremely low Prototype CKA (0.248) and near-perfect Covariance CKA (0.999), indicating dimensional collapse where all class representations point in the same direction. Both ERM and RDI dramatically improve Prototype CKA, with RDI achieving the highest average (0.940 vs. 0.925 for ERM), indicating better preservation of class-discriminative geometry across domains.

*Table 14.* **CKA analysis on PACS (Test-Train).** For fair comparison, all methods use backbone features. Prototype CKA ($\uparrow$) measures class layout consistency; Covariance CKA captures feature diversity (moderate values preferred over $\approx 1.0$ which indicates collapse). Results are reported as the mean of 3 runs.

| Method | Metric | Test Domain | | | | Avg |
| --- | --- | --- | --- | --- | --- | --- |
| | | Art | Cartoon | Photo | Sketch | |
| Untrained | Proto CKA | 0.347 | 0.086 | 0.318 | 0.239 | 0.248 |
| | Cov CKA | 0.999 | 0.999 | 1.000 | 1.000 | 0.999 |
| ERM | Proto CKA | $0.943_{\pm 0.004}$ | $0.904_{\pm 0.011}$ | $0.961_{\pm 0.005}$ | $0.892_{\pm 0.017}$ | 0.925 |
| | Cov CKA | $0.879_{\pm 0.029}$ | $0.782_{\pm 0.012}$ | $0.839_{\pm 0.014}$ | $0.760_{\pm 0.031}$ | 0.815 |
| RDI (Ours) | Proto CKA | $\mathbf{0.962}_{\pm 0.001}$ | $0.889_{\pm 0.008}$ | $\mathbf{0.986}_{\pm 0.002}$ | $\mathbf{0.924}_{\pm 0.016}$ | $\mathbf{0.940}$ |
| | Cov CKA | $0.597_{\pm 0.001}$ | $0.690_{\pm 0.002}$ | $0.721_{\pm 0.009}$ | $0.509_{\pm 0.031}$ | 0.629 |

[1]S. Kornblith et al., "Similarity of Neural Network Representations Revisited," ICML 2019.

**Results on OfficeHome.** Table 15 presents the CKA analysis on OfficeHome. Similar patterns emerge: the Untrained baseline shows severe dimensional collapse (Prototype CKA = 0.083, Covariance CKA = 0.992). RDI consistently outperforms ERM in Prototype CKA across all test domains, with an average improvement of $+0.034$ (0.880 vs. 0.846).

*Table 15.* **CKA analysis on OfficeHome (Test-Train).** For fair comparison, all methods use backbone features. Prototype CKA ($\uparrow$) measures class layout consistency; Covariance CKA captures feature diversity. Results are reported as the mean of 3 runs.

| | | Test Domain | | | | |
|---|---|---|---|---|---|---|
| **Method** | **Metric** | Art | Clipart | Product | Real | **Avg** |
| Untrained | Proto CKA | 0.093 | 0.028 | 0.084 | 0.128 | 0.083 |
| | Cov CKA | 0.993 | 0.990 | 0.990 | 0.993 | 0.992 |
| ERM | Proto CKA | $0.860_{\pm 0.004}$ | $0.761_{\pm 0.009}$ | $0.870_{\pm 0.005}$ | $0.894_{\pm 0.005}$ | 0.846 |
| | Cov CKA | $0.473_{\pm 0.006}$ | $0.486_{\pm 0.006}$ | $0.542_{\pm 0.002}$ | $0.590_{\pm 0.005}$ | 0.523 |
| RDI (Ours) | Proto CKA | $\mathbf{0.870}_{\pm 0.001}$ | $\mathbf{0.812}_{\pm 0.008}$ | $\mathbf{0.903}_{\pm 0.006}$ | $\mathbf{0.936}_{\pm 0.001}$ | **0.880** |
| | Cov CKA | $\mathbf{0.573}_{\pm 0.006}$ | $\mathbf{0.571}_{\pm 0.004}$ | $\mathbf{0.649}_{\pm 0.008}$ | $\mathbf{0.608}_{\pm 0.002}$ | **0.600** |

**Key findings.** **Finding 1: Training eliminates dimensional collapse.** The Untrained baseline exhibits near-perfect Covariance CKA ($\approx 1.0$) with extremely low Prototype CKA ($< 0.25$), indicating that randomly initialized networks produce collapsed representations where all classes are indistinguishable. Task-specific training (both ERM and RDI) dramatically improves class separability.

**Finding 2: RDI achieves superior prototype consistency.** RDI consistently outperforms ERM in Prototype CKA (PACS: $+0.015$; OfficeHome: $+0.034$), indicating that RDI learns class representations whose geometric layout is more consistent across domains. This aligns with the theoretical prediction that minimizing $\Delta_{\mathrm{geo}}$ promotes domain-invariant semantic geometry.

### D.5.2. SPECTRAL ANALYSIS: FEATURE DISTRIBUTION HEALTH

**Relation to RDI rate term.** The rate term $R_\tau(\mathcal{F}_c)$ (Eq. 12) regularizes spectral complexity via the log-determinant of class-conditional covariances. Under the Gaussian approximation, minimizing $R_\tau$ is equivalent to controlling $\log \det \Sigma_c$, which directly governs the eigenspectrum of the feature distribution. Spectral metrics such as Stable Rank and Condition Number reflect the quantities this term controls.

**Methodology.** For each class $c$ and domain $e$, we compute the empirical feature covariance matrix $\Sigma_{c,e} = \frac{1}{n} \sum_{i=1}^{n} (z_i - \bar{z})(z_i - \bar{z})^\top$ and analyze its eigenspectrum $\{\lambda_i\}_{i=1}^{d}$. We report three complementary metrics:

- **Stable Rank** $= \frac{\mathrm{tr}(\Sigma)}{\|\Sigma\|_{\mathrm{op}}} = \frac{\sum_i \lambda_i}{\lambda_{\max}}$ measures the *effective dimensionality* relative to the dominant direction. Higher values indicate that features utilize more dimensions.

- **Subspace Condition Number** $= \frac{\lambda_1^{(k)}}{\lambda_k^{(k)}}$ (computed on top-$k$ principal components) measures the *spectral uniformity* within the effective subspace. Lower values indicate more balanced eigenvalue distribution.

- **Participation Ratio** $= \frac{(\sum_{i=1}^{k} \lambda_i)^2}{\sum_{i=1}^{k} \lambda_i^2}$ (computed on top-$k$ subspace) quantifies spectral diversity with range $[1, k]$, where $k$ indicates perfect uniformity and 1 indicates a single dominant direction.

All metrics are averaged across classes and training domains for each held-out test domain configuration.

**Results on PACS.** Table 16 presents the spectral analysis on PACS backbone features.

*Table 16.* **Spectral analysis on PACS (backbone layer).** We report Stable Rank (↑), Subspace Condition Number $\log_{10}(\kappa)$ on top-32 PCs (↓), and Participation Ratio (↑) for each test domain. All metrics are averaged over classes and training domains. Results are reported as the mean of 3 runs.

| Method | Metric | Test Domain | | | | Avg |
|---|---|---|---|---|---|---|
| | | Art | Cartoon | Photo | Sketch | |
| ERM | Stable Rank | $2.44_{\pm 0.10}$ | $2.72_{\pm 0.08}$ | $2.76_{\pm 0.08}$ | $2.78_{\pm 0.07}$ | 2.68 |
| | Sub. $\log_{10}(\kappa)$ | $2.30_{\pm 0.05}$ | $2.21_{\pm 0.01}$ | $2.19_{\pm 0.03}$ | $2.20_{\pm 0.05}$ | 2.23 |
| | Part. Ratio | $4.11_{\pm 0.21}$ | $4.72_{\pm 0.19}$ | $4.86_{\pm 0.14}$ | $4.90_{\pm 0.15}$ | 4.65 |
| RDI (Ours) | Stable Rank | $\mathbf{4.66}_{\pm 0.07}$ | $\mathbf{3.88}_{\pm 0.07}$ | $\mathbf{4.58}_{\pm 0.11}$ | $\mathbf{3.52}_{\pm 0.04}$ | **4.16** |
| | Sub. $\log_{10}(\kappa)$ | $\mathbf{1.70}_{\pm 0.02}$ | $\mathbf{1.89}_{\pm 0.03}$ | $\mathbf{1.68}_{\pm 0.02}$ | $\mathbf{2.06}_{\pm 0.03}$ | **1.83** |
| | Part. Ratio | $\mathbf{8.46}_{\pm 0.17}$ | $\mathbf{7.06}_{\pm 0.18}$ | $\mathbf{8.63}_{\pm 0.20}$ | $\mathbf{6.06}_{\pm 0.08}$ | **7.55** |
| *RDI Improvement* | | +91% | +43% | +66% | +27% | **+55%** |

On PACS, RDI demonstrates **substantial spectral improvements** over ERM:

- Stable Rank increases by **55%** on average (2.68 → 4.16), indicating that RDI utilizes nearly twice as many effective dimensions.

- Subspace Condition Number decreases by **18%** ($\log_{10} \kappa$: 2.23 → 1.83), indicating more balanced eigenvalue distribution within the effective subspace.

- Participation Ratio increases by **62%** (4.65 → 7.55), confirming improved spectral diversity.

**Results on OfficeHome.** Table 17 presents the spectral analysis on OfficeHome backbone features.

*Table 17.* **Spectral analysis on OfficeHome (backbone layer).** We report Stable Rank (↑), Subspace Condition Number $\log_{10}(\kappa)$ (↓), and Participation Ratio (↑) for each test domain. Results are reported as the mean of 3 runs.

| Method | Metric | Test Domain | | | | Avg |
|---|---|---|---|---|---|---|
| | | Art | Clipart | Product | Real | |
| ERM | Stable Rank | $4.79_{\pm 0.07}$ | $4.66_{\pm 0.07}$ | $4.49_{\pm 0.03}$ | $4.49_{\pm 0.06}$ | 4.61 |
| | Sub. $\log_{10}(\kappa)$ | $2.97_{\pm 0.01}$ | $2.97_{\pm 0.01}$ | $2.99_{\pm 0.01}$ | $2.99_{\pm 0.01}$ | 2.98 |
| | Part. Ratio | $10.39_{\pm 0.16}$ | $10.02_{\pm 0.13}$ | $9.74_{\pm 0.06}$ | $9.78_{\pm 0.14}$ | 9.98 |
| RDI (Ours) | Stable Rank | $4.01_{\pm 0.02}$ | $3.83_{\pm 0.06}$ | $3.63_{\pm 0.07}$ | $\mathbf{4.74}_{\pm 0.09}$ | 4.05 |
| | Sub. $\log_{10}(\kappa)$ | $\mathbf{2.95}_{\pm 0.01}$ | $2.98_{\pm 0.01}$ | $3.00_{\pm 0.01}$ | $\mathbf{2.85}_{\pm 0.01}$ | **2.95** |
| | Part. Ratio | $8.94_{\pm 0.01}$ | $8.38_{\pm 0.20}$ | $7.95_{\pm 0.18}$ | $\mathbf{10.49}_{\pm 0.18}$ | 8.94 |
| *RDI Change* | | -16% | -18% | -19% | +6% | **-12%** |

On OfficeHome, RDI exhibits a **different spectral pattern**:

- Stable Rank *decreases* by 12% on average (4.61 → 4.05), suggesting that RDI compresses the feature distribution.

- Subspace Condition Number remains comparable (2.98 → 2.95, a marginal 1% improvement).

- Participation Ratio decreases by 10% (9.98 → 8.94), consistent with the Stable Rank trend.

**Key finding: Adaptive spectral regularization.** The contrasting patterns on PACS vs. OfficeHome reveal that **RDI performs adaptive spectral regularization based on task complexity**, consistent with the rate-distortion principle of finding the optimal compression level for each task.

*Table 18.* **Summary: Adaptive spectral behavior of RDI.** RDI expands collapsed representations (PACS) while compressing redundant ones (OfficeHome), maintaining comparable subspace condition numbers in both cases.

| Dataset | Classes | ERM Stable Rank | RDI Behavior | Change |
|---|---|---|---|---|
| PACS | 7 | 2.68 (collapsed) | **Expand** | +55% |
| OfficeHome | 65 | 4.61 (distributed) | **Compress** | -12% |

**PACS (7 classes):** ERM exhibits dimensional collapse (Stable Rank = 2.68), utilizing only ∼3 effective dimensions to distinguish 7 classes. RDI's rate term encourages the model to utilize more dimensions (Stable Rank → 4.16), providing sufficient capacity for robust cross-domain generalization.

**OfficeHome (65 classes):** ERM already learns a relatively distributed representation (Stable Rank = 4.61) to discriminate 65 classes. Here, RDI *compresses* the feature space (Stable Rank → 4.05) while maintaining similar condition numbers, suggesting that ERM's representation contains redundancy that RDI removes.

**Interpretation:** RDI does not mechanically expand or compress feature dimensionality. Instead, it learns an *appropriately sized* representation dictated by the rate-distortion trade-off: expanding when capacity is insufficient (PACS), compressing when redundancy exists (OfficeHome). This adaptive behavior is a hallmark of rate-distortion optimization.

### D.5.3. RATE-DISTORTION TRAJECTORY ANALYSIS

**Relation to Pareto optimality.** Classical rate-distortion theory establishes a fundamental trade-off: for any source, there exists a *rate-distortion curve $R(D)$* representing the minimum achievable rate for a given distortion level. This curve constitutes the *Pareto front*—points on the curve are Pareto optimal in the sense that neither rate nor distortion can be reduced without increasing the other. RDI's joint optimization should drive representations toward this Pareto front.

**Methodology.** For each training run, we record the rate term $R = R_\tau(\mathcal{F})$ and the geometric distortion term $\Delta_{\text{geo}}$ at initialization and convergence. We plot the optimization trajectory in the $(R, \Delta)$ plane, where the starting point (red circle) corresponds to initialization and the final point (blue circle) corresponds to convergence.

**Results.** Figure 9 presents the domain-wise optimization trajectories on PACS and OfficeHome.

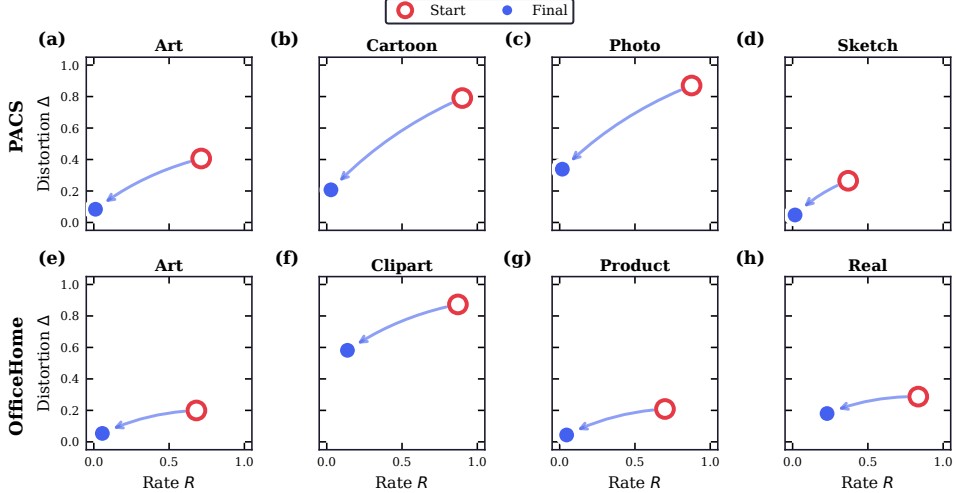

*Figure 9.* **Domain-wise rate-distortion optimization trajectories.** Each panel shows one domain's trajectory from initialization (red circle) to convergence (blue circle). All domains exhibit consistent movement toward the lower-left region, indicating simultaneous reduction of rate $R$ and distortion $\Delta$—consistent with convergence toward the Pareto front. Results are reported as the mean of 3 runs.

**Key finding: Convergence toward the Pareto front.** Although different domains follow geometrically distinct paths depending on their initial conditions, all consistently optimize toward lower distortion and lower rate. This demonstrates that RDI's combined objective provides a coherent optimization target: rather than trading off one term for the other, the joint optimization discovers representations that improve both simultaneously until reaching the Pareto front where further improvement in one necessitates degradation in the other.

### D.5.4. SUMMARY: CONVERGENCE TO THE RATE-DISTORTION PARETO FRONT

The analyses in this section validate both components of the RDI objective and their joint optimization:

- **Distortion term → CKA analysis (Section D.5.1):** RDI achieves higher Prototype CKA than ERM (PACS: $+0.015$; OfficeHome: $+0.034$), confirming that $\Delta_{\text{geo}}$ promotes cross-domain consistency of class-discriminative geometry. The improvement in prototype-level alignment validates that minimizing Grassmannian distances induces semantically meaningful invariance.

- **Rate term → Spectral analysis (Section D.5.2):** RDI adaptively regulates spectral complexity—expanding collapsed representations (PACS: Stable Rank $+55\%$) while compressing redundant ones (OfficeHome: $-12\%$). This task-adaptive behavior is consistent with $R_\tau$'s role as capacity regularization: the rate term finds the appropriate complexity level for each task rather than imposing a fixed constraint.

- **Joint optimization → R-D trajectory (Section D.5.3):** All domains converge toward the lower-left region of the $(R, \Delta)$ plane, consistent with convergence to the *Pareto front* of the rate-distortion trade-off. The hyperparameter $\beta$ modulates the operating point along this front, allowing practitioners to balance alignment and compression based on task requirements.

**Integrated interpretation.** These findings support the hypothesis that RDI's generalization advantage stems from finding Pareto-optimal representations that balance cross-domain geometric consistency (distortion) with spectral efficiency (rate). Neither component alone is sufficient: geometric alignment without rate control can lead to ill-conditioned representations (Proposition 3.14), while rate control without geometric constraints may sacrifice cross-domain consistency. RDI's joint optimization navigates this trade-off, converging to operating points on the Pareto front that achieve both properties simultaneously.

### D.5.5. QUALITATIVE VISUALIZATION VIA T-SNE

To provide a qualitative complement to the quantitative analyses above (CKA, spectral metrics, R–D trajectory), we visualize learned representations via t-SNE (van der Maaten & Hinton, 2008) on PACS.

**Setup.** We train ERM and RDI with the ResNet-50 backbone on PACS using leave-Art-out training-domain validation (i.e., Art is the unseen test domain). After training, we extract backbone features for randomly sampled examples across all four domains (Art, Cartoon, Photo, Sketch), and project them to 2D via t-SNE with default perplexity. Points are colored by their source domain to expose domain-related structure in the representation space.

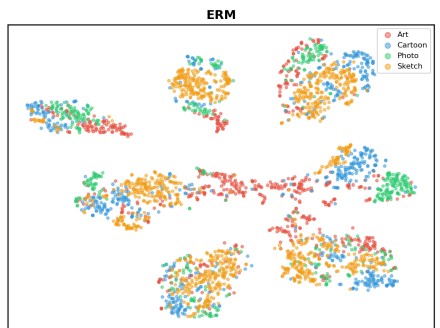 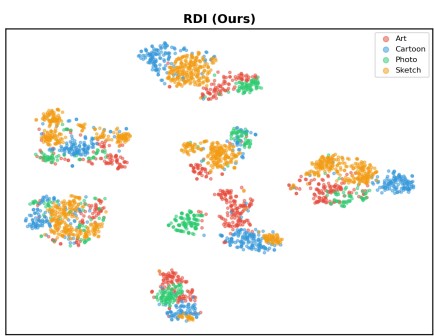

*Figure 10.* **t-SNE visualization of backbone features on PACS** (test domain: Art). Points are colored by source domain. **Left (ERM):** clusters exhibit irregular shapes with visible substructure within individual clusters. **Right (RDI):** clusters are tighter and more distinctly separated, with source domains coexisting within each cluster rather than forming domain-specific sub-regions. The unseen test domain (Art, red) is consistently aligned with the training-domain clusters.

**Interpretation.** The visual pattern is consistent with the quantitative findings above. The tighter and more distinct RDI clusters align with the spectral analysis (Appendix D.5.2), which showed improved spectral conditioning of class-conditional features. The co-occurrence of multiple source domains within the same cluster aligns with the higher cross-domain prototype CKA reported in Appendix D.5.1. Together, the quantitative and qualitative evidence indicate that RDI's geometric rate–distortion objective produces representations whose macroscopic structure is more consistent across domains.

## D.6. Relation Between Theoretical Assumptions and Empirical Observations

This section elaborates on the relationship between the rotation assumption (Assumption 3.5) and the empirical findings in Appendix D.5.

**Perturbation model vs. perturbation instance.** Our theoretical framework adopts bounded rotations as a tractable model for domain shift, enabling rigorous analysis. While real-world shifts involve more complex transformations, the empirical results demonstrate that the core principle of geometric alignment is effectively operative, as evidenced below:

1. **Validation via Semantic Alignment:** Instead of relying on rigid geometric metrics, we utilize **Prototype CKA** (Tables 14 and 15) to capture semantic consistency. The consistent improvement in Prototype CKA across all domains confirms that RDI aligns the geometric arrangement of class centroids more effectively than ERM. This validates that the method recovers semantically invariant subspaces, achieving the goal of alignment even under complex, non-rotational shifts.

2. **Instantiation vs. Principle:** The rotation assumption provides a tractable instantiation for deriving generalization bounds. The underlying rate-distortion principle, however, governs a broader class of shifts. The empirical success of RDI indicates that minimizing our objective effectively regularizes representation geometry, leading to invariance properties consistent with the theory, beyond the strict rotation model.

3. **Holistic Representation Control:** The combined analysis of CKA and spectral properties (Section D.5.2) demonstrates that RDI achieves alignment at multiple levels. By navigating the trade-off between spectral capacity (via the rate term) and geometric consistency (via the distortion term), RDI induces representations that are both compact and invariant. This provides empirical support for the necessity of the joint optimization mechanism in our framework.

**Role of projection regularization.** The theoretical guaranties operate under a structure of "semantic preservation + geometric optimization": classification loss $\widehat{\mathcal{R}}_{\mathrm{src}}$ provides discriminativeness, while the projection term $\Delta_{\mathrm{rec}}$ prevents information collapse during geometric alignment. Given these preconditions, geometric alignment ($\Delta_{\mathrm{geo}}$) coupled with rate control ($\bar{R}$) recovers invariant subspaces. This mechanism, not the specific rotation model, is what allows generalization beyond the theoretical regime.

## D.7. Extension to Affine Perturbations (Sketch)

The rotation instantiation (Assumption 3.5) represents one point in a hierarchy of geometric perturbation models. Here we sketch the extension to bounded-condition-number affine transformations:

$$\mathbf{f}_{e,c} = A_e U_c^* \mathbf{z} + \boldsymbol{\epsilon}_e, \quad \text{where } \kappa(A_e) \le K.$$

This broader model captures domain shifts involving both directional misalignment (rotation component) and anisotropic scaling (condition number), while preserving the core geometric structure that RDI exploits.

**Required modifications.**

1. **Distance metric.** Replace the Grassmann distance $d_{\mathrm{Gr}}(\cdot, \cdot)$ with a joint metric capturing both angular and spectral discrepancy:
$$d_{\mathrm{joint}}(U, \Sigma; U', \Sigma') = d_{\mathrm{Gr}}(U, U') + \mu \|\log \Sigma - \log \Sigma'\|_F,$$
where $\Sigma, \Sigma'$ are the within-subspace covariances and $\mu > 0$ balances the two components.

2. **Recovery bound.** Theorem 3.10 would generalize with an additional $\kappa(A_e)^2$ factor in the constants. Specifically, the Davis–Kahan perturbation bound depends on the eigengap $\lambda_k - \lambda_{k+1}$, which scales inversely with the condition number when $A_e$ is non-orthogonal.

3. **Generalization bound.** Theorem 3.11 remains structurally similar, with the geometric discrepancy $\rho$ replaced by a condition-number-aware quantity $\tilde{\rho} = \max_{e,c} d_{\mathrm{joint}}(U_{t,c}, \Sigma_{t,c}; U_{e,c}, \Sigma_{e,c})$.

4. **Rate term as implicit condition-number control.** The existing rate term $\bar{R}(\mathcal{F})$ (Eq. 14) with per-class rates $R_c = R_\tau(\mathcal{F}_c)$ (Eq. 12) naturally penalizes spectral expansion induced by non-orthogonal $A_e$. Specifically, if $A_e$ has condition number $\kappa(A_e) > 1$, then $A_e U_c^* \mathbf{z}$ has inflated variance along certain directions, which the rate term penalizes. This provides implicit regularization against ill-conditioned transformations even without explicit condition-number constraints in the objective. The rate term thus serves a dual role: controlling generalization complexity (Theorem 3.11) and implicitly regularizing against pathological affine perturbations.

**Challenges.** The main technical challenge is that the Fréchet mean on the extended space (Grassmannian $\times$ positive-definite cone) lacks the clean uniqueness guarantees available for the pure Grassmannian under small rotations. A full treatment requires either stronger separation assumptions or a hierarchical optimization that first aligns subspaces, then matches spectral profiles. We leave a rigorous analysis of this extension to future work.

## E. Full Comparison Results

We report the complete leave-one-domain-out results on each DomainBed dataset using the training-domain validation protocol. †: results from DomainBed (Gulrajani & Lopez-Paz, 2021); other baselines are from their original papers. LFME reports only average accuracy.

*Table 19.* Accuracy (%) on **PACS** under train-domain validation. †: from DomainBed. **Bold**: best, underline: second best.

| Method | Art Painting | Cartoon | Photo | Sketch | Avg. |
|---|---|---|---|---|---|
| ERM[†] | $84.7 \pm 0.4$ | $80.8 \pm 0.6$ | $97.2 \pm 0.3$ | $79.3 \pm 1.0$ | 85.5 |
| IRM[†] | $84.8 \pm 1.3$ | $76.4 \pm 1.1$ | $96.7 \pm 0.6$ | $76.1 \pm 1.0$ | 83.5 |
| VREx[†] | $86.0 \pm 1.6$ | $79.1 \pm 0.6$ | $96.9 \pm 0.5$ | $77.7 \pm 1.7$ | 84.9 |
| GroupDRO[†] | $83.5 \pm 0.9$ | $79.1 \pm 0.6$ | $96.7 \pm 0.3$ | $78.3 \pm 2.0$ | 84.4 |
| DANN[†] | $86.4 \pm 0.8$ | $77.4 \pm 0.8$ | $97.3 \pm 0.4$ | $73.5 \pm 2.3$ | 83.7 |
| CORAL[†] | $88.3 \pm 0.2$ | $80.0 \pm 0.5$ | $97.5 \pm 0.3$ | $78.8 \pm 1.3$ | 86.2 |
| Fishr | **$88.4 \pm 0.2$** | $78.7 \pm 0.7$ | $97.0 \pm 0.1$ | $77.8 \pm 2.0$ | 85.5 |
| RSC[†] | $85.4 \pm 0.8$ | $79.7 \pm 1.8$ | **$97.6 \pm 0.3$** | $78.2 \pm 1.2$ | 85.2 |
| SagNet[†] | $87.4 \pm 1.0$ | $80.7 \pm 0.6$ | $97.1 \pm 0.1$ | $80.0 \pm 0.4$ | 86.3 |
| Mixup[†] | $86.1 \pm 0.5$ | $78.9 \pm 0.8$ | **$97.6 \pm 0.1$** | $75.8 \pm 1.8$ | 84.6 |
| GGA | $86.5 \pm 1.8$ | $81.2 \pm 3.0$ | $97.1 \pm 0.9$ | $80.8 \pm 0.9$ | 86.4 |
| LFME | – | – | – | – | 85.0 |
| RDI (Ours) | **$88.4 \pm 1.0$** | **$81.5 \pm 1.0$** | $97.4 \pm 0.5$ | **$82.3 \pm 1.1$** | **87.4** |

*Table 20.* Accuracy (%) on **VLCS** under train-domain validation. †: from DomainBed. **Bold**: best, underline: second best.

| Method | Caltech | LabelMe | SUN | VOC | Avg. |
|---|---|---|---|---|---|
| ERM[†] | $97.7 \pm 0.4$ | $64.3 \pm 0.9$ | $73.4 \pm 0.5$ | $74.6 \pm 1.3$ | 77.5 |
| IRM[†] | $98.6 \pm 0.1$ | $64.9 \pm 0.9$ | $73.4 \pm 0.6$ | $77.3 \pm 0.9$ | 78.6 |
| VREx[†] | $98.4 \pm 0.3$ | $64.4 \pm 1.4$ | **$74.1 \pm 0.4$** | $76.2 \pm 1.3$ | 78.3 |
| GroupDRO[†] | $97.3 \pm 0.3$ | $63.4 \pm 0.9$ | $69.5 \pm 0.8$ | $76.7 \pm 0.7$ | 76.7 |
| DANN[†] | **$99.0 \pm 0.3$** | $65.1 \pm 1.4$ | $73.1 \pm 0.3$ | $77.2 \pm 0.6$ | 78.6 |
| CORAL[†] | $98.3 \pm 0.1$ | **$66.1 \pm 1.2$** | $73.4 \pm 0.3$ | **$77.5 \pm 1.2$** | **78.8** |
| Fishr | $98.9 \pm 0.3$ | $64.0 \pm 0.5$ | $71.5 \pm 0.2$ | $76.8 \pm 0.7$ | 77.8 |
| RSC[†] | $97.9 \pm 0.1$ | $62.5 \pm 0.7$ | $72.3 \pm 1.2$ | $75.6 \pm 0.8$ | 77.1 |
| SagNet[†] | $97.9 \pm 0.4$ | $64.5 \pm 0.5$ | $71.4 \pm 1.3$ | **$77.5 \pm 0.5$** | 77.8 |
| Mixup[†] | $98.3 \pm 0.6$ | $64.8 \pm 1.0$ | $72.1 \pm 0.5$ | $74.3 \pm 0.8$ | 77.4 |
| GGA | $98.4 \pm 0.2$ | $65.4 \pm 0.1$ | $73.8 \pm 1.6$ | $77.4 \pm 1.9$ | 78.7 |
| LFME | – | – | – | – | 78.4 |
| RDI (Ours) | $98.0 \pm 0.8$ | $64.7 \pm 0.8$ | $73.7 \pm 1.5$ | $76.4 \pm 1.8$ | 78.2 |

*Table 21.* Accuracy (%) on **OfficeHome** under train-domain validation. †: from DomainBed. **Bold**: best, underline: second best.

| Method | Art | Clipart | Product | Real World | Avg. |
|---|---|---|---|---|---|
| ERM[†] | 61.3 ± 0.7 | 52.4 ± 0.3 | 75.8 ± 0.1 | 76.6 ± 0.3 | 66.5 |
| IRM[†] | 58.9 ± 2.3 | 52.2 ± 1.6 | 72.1 ± 2.9 | 74.0 ± 2.5 | 64.3 |
| VREx[†] | 60.7 ± 0.9 | 53.0 ± 0.9 | 75.3 ± 0.1 | 76.6 ± 0.5 | 66.4 |
| GroupDRO[†] | 60.4 ± 0.7 | 52.7 ± 1.0 | 75.0 ± 0.7 | 76.0 ± 0.7 | 66.0 |
| DANN[†] | 59.9 ± 1.3 | 53.0 ± 0.3 | 73.6 ± 0.7 | 76.9 ± 0.5 | 65.9 |
| CORAL[†] | $\underline{65.3}$ ± 0.4 | 54.4 ± 0.5 | 76.5 ± 0.1 | $\underline{78.4}$ ± 0.5 | 68.7 |
| Fishr | 62.4 ± 0.5 | 54.4 ± 0.4 | 76.2 ± 0.5 | 78.3 ± 0.1 | 67.8 |
| RSC[†] | 60.7 ± 1.4 | 51.4 ± 0.3 | 74.8 ± 1.1 | 75.1 ± 1.3 | 65.5 |
| SagNet[†] | 63.4 ± 0.2 | $\underline{54.8}$ ± 0.4 | 75.8 ± 0.4 | 78.3 ± 0.3 | 68.1 |
| Mixup[†] | 62.4 ± 0.8 | $\underline{54.8}$ ± 0.6 | 76.9 ± 0.3 | 78.3 ± 0.2 | 68.1 |
| GGA | 61.7 ± 0.1 | 52.5 ± 0.5 | $\underline{77.1}$ ± 1.3 | 77.0 ± 0.1 | 67.0 |
| LFME | – | – | – | – | $\underline{69.1}$ |
| RDI (Ours) | **65.7** ± 0.5 | **56.4** ± 0.1 | **77.2** ± 0.4 | **80.4** ± 0.2 | **69.9** |

*Table 22.* Accuracy (%) on **TerraIncognita** under train-domain validation. †: from DomainBed. **Bold**: best, underline: second best.

| Method | Location 100 | Location 38 | Location 43 | Location 46 | Avg. |
|---|---|---|---|---|---|
| ERM[†] | 49.8 ± 4.4 | 42.1 ± 1.4 | 56.9 ± 1.8 | 35.7 ± 3.9 | 46.1 |
| IRM[†] | $\underline{54.6}$ ± 1.3 | 39.8 ± 1.9 | 56.2 ± 1.8 | 39.6 ± 0.8 | 47.6 |
| VREx[†] | 48.2 ± 4.3 | 41.7 ± 1.3 | 56.8 ± 0.8 | 38.7 ± 3.1 | 46.4 |
| GroupDRO[†] | 41.2 ± 0.7 | 38.6 ± 2.1 | 56.7 ± 0.9 | 36.4 ± 2.1 | 43.2 |
| DANN[†] | 51.1 ± 3.5 | 40.6 ± 0.6 | 57.4 ± 0.5 | 37.7 ± 1.8 | 46.7 |
| CORAL[†] | 51.6 ± 2.4 | 42.2 ± 1.0 | 57.0 ± 1.0 | 39.8 ± 2.9 | 47.7 |
| Fishr | 50.2 ± 3.9 | **43.9** ± 0.8 | 55.7 ± 2.2 | 39.8 ± 1.0 | 47.4 |
| RSC[†] | 50.2 ± 2.2 | 39.2 ± 1.4 | 56.3 ± 1.4 | 40.8 ± 0.6 | 46.6 |
| SagNet[†] | 53.0 ± 2.9 | $\underline{43.0}$ ± 2.5 | $\underline{57.9}$ ± 0.6 | 40.4 ± 1.3 | $\underline{48.6}$ |
| Mixup[†] | **59.6** ± 2.0 | 42.2 ± 1.4 | 55.9 ± 0.8 | 33.9 ± 1.4 | 47.9 |
| GGA | 50.9 ± 2.2 | 42.5 ± 1.0 | **59.7** ± 1.4 | **41.5** ± 3.5 | 48.5 |
| LFME | – | – | – | – | 48.3 |
| RDI (Ours) | 53.5 ± 3.6 | $\underline{43.0}$ ± 1.5 | 57.0 ± 0.5 | $\underline{41.4}$ ± 1.8 | **48.7** |

*Table 23.* Accuracy (%) on **DomainNet** under train-domain validation. †: from DomainBed. **Bold**: best, underline: second best.

| Method | Clipart | Infograph | Painting | Quickdraw | Real | Sketch | Avg. |
|---|---|---|---|---|---|---|---|
| ERM[†] | 58.1 ± 0.3 | 18.8 ± 0.3 | 46.7 ± 0.3 | 12.2 ± 0.4 | 59.6 ± 0.1 | 49.8 ± 0.4 | 40.9 |
| IRM[†] | 48.5 ± 2.8 | 15.0 ± 1.5 | 38.3 ± 4.3 | 10.9 ± 0.5 | 48.2 ± 5.2 | 42.3 ± 3.1 | 33.9 |
| VREx[†] | 47.3 ± 3.5 | 16.0 ± 1.5 | 35.8 ± 4.6 | 10.9 ± 0.3 | 49.6 ± 4.9 | 42.0 ± 3.0 | 33.6 |
| GroupDRO[†] | 47.2 ± 0.5 | 17.5 ± 0.4 | 33.8 ± 0.5 | 9.3 ± 0.3 | 51.6 ± 0.4 | 40.1 ± 0.6 | 33.3 |
| DANN[†] | 53.1 ± 0.2 | 18.3 ± 0.1 | 44.2 ± 0.7 | 11.8 ± 0.1 | 55.5 ± 0.4 | 46.8 ± 0.6 | 38.3 |
| CORAL[†] | 59.2 ± 0.1 | 19.7 ± 0.2 | 46.6 ± 0.3 | 13.4 ± 0.4 | 59.8 ± 0.2 | 50.1 ± 0.6 | 41.5 |
| Fishr | 58.2 ± 0.5 | 20.2 ± 0.2 | 47.7 ± 0.3 | 12.7 ± 0.2 | 60.3 ± 0.2 | 50.8 ± 0.1 | 41.7 |
| RSC[†] | 55.0 ± 1.2 | 18.3 ± 0.5 | 44.4 ± 0.6 | 12.2 ± 0.2 | 55.7 ± 0.7 | 47.8 ± 0.9 | 38.9 |
| SagNet[†] | 57.7 ± 0.3 | 19.0 ± 0.2 | 45.3 ± 0.3 | 12.7 ± 0.5 | 58.1 ± 0.5 | 48.8 ± 0.2 | 40.3 |
| Mixup[†] | 55.7 ± 0.3 | 18.5 ± 0.5 | 44.3 ± 0.5 | 12.5 ± 0.4 | 55.8 ± 0.3 | 48.2 ± 0.5 | 39.2 |
| GGA | **63.7** ± 0.2 | **21.3** ± 0.3 | **50.4** ± 0.1 | **14.1** ± 0.2 | $\underline{63.8}$ ± 0.4 | **53.5** ± 0.3 | **44.4** |
| LFME | – | – | – | – | – | – | 42.1 |
| RDI (Ours) | $\underline{63.4}$ ± 0.1 | $\underline{21.0}$ ± 0.3 | $\underline{50.0}$ ± 0.2 | $\underline{13.7}$ ± 0.3 | **64.9** ± 0.7 | $\underline{52.3}$ ± 0.4 | $\underline{44.2}$ |

