# OpenReview forum: "Geometric Rate–Distortion Invariance for Domain Generalization"
_ICML.cc/2026/Conference — ICML 2026 regular_

### Official Review · Reviewer_yXjS · 2026-03-09

**Soundness:** 2
**Presentation:** 3
**Significance:** 3
**Originality:** 3
**Overall Recommendation:** 4
**Confidence:** 2

**Summary:**

The paper introduces Geometric Rate–Distortion Invariance (RDI), a domain generalization (DG) framework that represents class-conditional feature distributions as low-dimensional subspaces on the Grassmann manifold. The method jointly optimizes a geometric alignment objective, interpreted as distortion, together with a spectral complexity term corresponding to rate. It further incorporates a capacity-controlled Rényi rate formulation using a Fisher-diagonal Gaussian surrogate, along with an escort-based mechanism for class-wise aggregation. Theoretical analysis provides finite-sample guarantees under bounded rotational domain shifts. Empirically, RDI is evaluated on the DomainBed benchmark using a ResNet-50 backbone and demonstrates competitive performance compared with representative DG baselines. Ablation studies further show that both the geometric alignment component and the complexity regularization contribute to improved generalization.

**Compliance With Llm Reviewing Policy:**

Affirmed.

**Final Justification:**

I am not an expert in domain generalization. Although the authors' rebuttal addressed many of my concerns, after further checking the latest papers I discovered a clear issue in the experimental section: in Table 1 the authors create their own narrow category “invariance-based and geometric alignment methods” (a classification that does not appear in prior DG literature, including GGA) and claim RDI (65.7%) is the best within it. This allows them to exclude stronger methods under the exact same protocol that were publicly available before submission, such as SFT (67.1%) and ERM++ (68.9%). Moreover, GGA itself is explicitly designed as a plug-in module that can be combined with any DG method.
This selective baseline comparison makes the empirical claims less convincing. Therefore, I am inclined to reject.

---
Thank you for the detailed and thoughtful response, particularly the clear explanation regarding the exclusion of SFT and ERM++ as orthogonal categories (flat-minima optimization and training recipes). The commitment to revise Table 1 to explicitly state the exclusion criteria in the camera-ready version fully addresses my primary remaining concern about baseline selection. Combined with the additional results on computational efficiency, ViT backbones, and standard deviations, I now consider my concerns resolved. I will therefore raise my score to Weak Accept (4).

I appreciate the constructive discussion and look forward to the revised manuscript.

**Key Questions For Authors:**

1.How are gradients handled for the geometric alignment component in the objective (i.e., the loss term corresponding to the geometric alignment term)? When computing the class–domain subspaces and the Grassmannian distance, is the eigendecomposition differentiated through during backpropagation, or are the subspaces treated as stop-gradient statistics updated via EMA? If the latter is the case, how does the geometric alignment loss provide a direct learning signal for feature representations beyond the EMA updates?
2.How sensitive is the method to several key hyperparameters, such as the subspace dimension parameter kkk and the parameters in the Rényi-based rate regularization term (e.g., those controlling class aggregation and variance scaling in the rate formulation)? Providing sensitivity curves or recommended default ranges would help clarify the robustness and stability of the method.
3.The method relies heavily on second-order statistics (e.g., covariance-related quantities and subspace eigendecomposition). Such operations can be computationally expensive in practice. It would therefore be helpful for the authors to clarify the computational cost and efficiency of the approach during training. For example, does the method significantly increase training time or memory usage compared to standard ERM or other DG methods? Providing a complexity analysis or empirical training-time comparisons would help assess the practicality of the proposed approach.

**Limitations:**

See weakness

**Strengths And Weaknesses:**

Strength：
1.The paper introduces a novel perspective by formulating domain generalization as a geometric rate–distortion optimization problem on the Grassmann manifold.
2.The experiments follow the widely adopted DomainBed protocol and datasets, enabling fair comparisons with prior domain generalization methods. Ablation studies investigate the contributions of both the geometric alignment term and the rate regularization term, while a synthetic experiment illustrates the rate–distortion trade-off implied by the proposed objective.
3.Overall, the paper is well-written and clearly structured.

Weakness：
1.The method relies heavily on second-order information throughout the formulation and optimization process. While such statistics may provide richer geometric or distributional structure, they typically introduce substantial computational and memory overhead. The paper does not analyze the resulting impact on training or inference efficiency, nor does it provide complexity comparisons with standard first-order methods. It would be important for the authors to clarify whether the use of extensive second-order information significantly affects computational efficiency in practice.
2.The experimental comparison omits several strong DG baselines, such as MIRO and ERM++, which are commonly included in recent DomainBed evaluations. The absence of these methods weakens the claim of competitiveness. Moreover, experiments are conducted only with ResNet backbones, while recent studies show that domain generalization performance can vary significantly when using Vision Transformer architectures. Finally, the main comparison table does not report standard deviations or confidence intervals, making it difficult to assess the statistical significance of the reported results.
3.Some implementation details are insufficiently specified. For example, the update frequency and computational cost of the subspace eigendecomposition are not clearly described, and the selection and tuning strategy for the subspace dimension parameter k is not explained. In addition, the memory overhead required to maintain per-domain-class subspaces on large datasets such as DomainNet is not discussed. The practical computation of the Fréchet mean estimator during training also remains unclear.

---

> ### Author Rebuttal · Authors · 2026-03-31
>
> We sincerely thank Reviewer yXjS for the constructive feedback and for recognizing our geometric formulation as a novel perspective for domain generalization. We comprehensively address your questions below, aiming to fully clarify the computational efficiency and gradient mechanisms of our approach.
>
> **1. Computational efficiency.** We have conducted a detailed empirical analysis. Training time increases by only $1.18-1.66\times$ over ERM, with negligible GPU memory overhead ($1.01-1.04\times$) when measured on a single NVIDIA RTX 5090 GPU. The key insight is that no $O(d^3)$ operation ever occurs in RDI. The rate term operates on the full $d$-dimensional feature spectrum to penalize spectral spread beyond the top-$k$ subspace, enforcing feature concentration. However, it uses a diagonal covariance approximation (per-coordinate variance), reducing computation from $O(d^3)$ to $O(d)$ (Appendix B.4). The geometric distortion involves only low-rank SVD (top-$k$, $k \ll d$) on per-domain-class sample matrices where $n \ll d$, and these decompositions are further amortized via EMA statistics with periodic updates (every 50 steps), not computed every iteration.
>
> **2. Missing baselines.** As discussed in our response to Reviewer 1Q1L, MIRO relies on frozen pretrained representations as an external anchor, which is orthogonal to RDI's learned geometric mechanism. We exclude it—along with SWAD—to ensure comparison under comparable pipelines. ERM++ (Teterwak et al., 2023) improves ERM through training recipes—model parameter averaging, warm-starting, AugMix initialization, LR scheduling—rather than introducing a new DG algorithm. These are orthogonal to RDI's geometric rate–distortion formulation and can be combined with it. We focus comparisons on methods that learn invariance from source-domain structure.
>
> **3. ViT and standard deviations.** New DeiT-Small experiments show RDI consistently outperforms ERM and SDViT (PACS 87.3%, OfficeHome 72.6%, VLCS 79.3%, TerraIncognita 43.6%). RDI achieves comparable performance across architectures (PACS: 87.4% ResNet vs. 87.3% DeiT) despite a weaker ViT baseline (ERM: 83.9% vs. 85.5%), confirming architecture-agnostic effectiveness. Regarding standard deviations, we strictly adhered to the DomainBed protocol (exact averages of 3 random seeds) and did not selectively choose favorable seeds. Full tables with $\pm$std are provided in the anonymous link and will be added to the appendix.
>
> **4. Gradient handling and implementation details.** RDI uses a hybrid optimization strategy analogous to target networks (e.g., BYOL, MoCo). The class-level Fréchet means $\bar{U} _ c$ are maintained as slowly moving EMA targets (detached from the computational graph), providing stable alignment anchors. To ensure that the geometric alignment loss $L _ {geo}$ provides direct gradients to the feature extractor, the current batch's domain-class subspace is estimated via differentiable SVD on the live features, while the alignment target (barycenter) remains detached. Gradients thus flow through: live features -> SVD -> Grassmann distance $d _ {Gr}(V _ {e,c}, \bar{U} _ c)$ -> feature extractor. This is the standard implementation pattern for target-network-style objectives. Additionally, the projection distortion $\|z - \Pi _ {\bar{U} _ c}z\|^2$ provides a complementary per-sample gradient signal independent of eigendecomposition, and the rate term $R _ c$ computed from live batch variances further regularizes the feature spectrum. The pairwise objective (Eq. 8) is implemented via barycenter anchoring—a standard metric-space equivalence reducing $O(M^2)$ to $O(M)$.
>
> **5. $k$ selection and hyperparameters.** While our objective contains several parameters, the effective tuning burden is substantially lighter than it appears. Table 6 shows $k=32$ is a highly robust default ($<2\%$ variation across $\{8, 16, 32, 64, 128\}$). Similarly, $\tau=1.0$ serves as a robust universal default, theoretically grounded in BN feature scale (Appendix B.4). Furthermore, $\gamma=0.5$ is fixed, and the EMA momentum sensitivity is negligible ($<0.05\%$). The primary parameters requiring dataset-specific tuning via standard validation are the Rényi order $\alpha$ (uniform weighting is suboptimal by $0.5-2.8\%$), along with the regularization coefficients $\lambda _ {\mathrm{geo}}$ and $\lambda _ {\mathrm{rec}}$.
>
> **6. DomainNet memory.** Each subspace is a $d _ z \times k$ matrix ($d _ z$ is the projection head output dimension). With $k=64$ for DomainNet and 345 classes $\times$ 6 domains = 2,070 subspaces, the total memory overhead is modest. Empirically: $1.04\times$ ($16.3 \to 17.0$ GB, Table 8).
>
> *(Note: ViT results, t-SNE visualizations, overhead analysis, and synthetic experiments are provided at our anonymous repository: https://anonymous.4open.science/r/RDI-rebuttal-material-7A41/)*

---

> > ### Author Rebuttal · Reviewer_yXjS · 2026-04-03
> >
> > Thank you for the detailed and helpful rebuttal. Most of my concerns regarding computational efficiency, architectural design, implementation details, and hyperparameter analysis have been addressed. I also understand the clarification that methods like MIRO are orthogonal to RDI. However, I suggest that the final version of the paper more explicitly discuss the distinctions and connections between RDI and these "anchor-based" or "training-recipe" approaches within the related work or experimental analysis sections. Additionally, since certain parameters still require dataset-specific tuning via validation, I am concerned that high sensitivity to these values might affect the method's practicality. I recommend including a more detailed analysis of the search ranges and sensitivity for these parameters in the appendix. Overall, I will consider increasing my score.

---

> > > ### Author Response · Authors · 2026-04-05
> > >
> > > Thank you for the continued engagement. We note that the concern regarding SFT was raised for the first time in the final justification — it was not mentioned in the initial review or the rebuttal acknowledgment, and we therefore had no opportunity to address it during the discussion period. We appreciate the chance to clarify now.
> > > 1. SFT is a flat-minima optimization method, which our paper explicitly excludes.
> > > SFT (Li et al., CVPR'25) is built upon SAM and seeks consistent flat minima across domains by refining loss landscapes. Its own experimental comparisons are against SAM, GSAM, FAD, and SAGM — all sharpness-aware optimization methods. Our manuscript states that "orthogonal techniques such as flat-minima optimization (e.g., SWAD) are excluded." SFT falls squarely into this excluded category, which is directly verifiable from SFT's Table 1 and Related Work.
> > > The orthogonality between flat-minima methods and objective-level DG methods is well established: Cha et al. (NeurIPS'21) demonstrate that SWAD can be combined with objective-level methods such as CORAL, yielding additive improvements — precisely because the two operate at different levels (optimizer-level flatness seeking vs. objective-level domain alignment). RDI, like CORAL, operates at the objective level and is thus orthogonal to such optimization-level techniques. Comparing RDI directly against SFT conflates these two distinct levels of contribution.
> > > 2. ERM++ was addressed in the first-round rebuttal, and the reviewer acknowledged this.
> > > We explained in our rebuttal that ERM++ (Teterwak et al., WACV'25) improves generalization through training recipes — parameter averaging, AugMix initialization, and LR scheduling — which are orthogonal to and composable with any DG objective including RDI. The final justification reintroduces ERM++ alongside the new SFT concern without addressing our prior explanation.
> > > 3. On the "narrow category" and the inclusion of GGA and LFME.
> > > Our core baselines (IRM, VREx, CORAL, Fishr, SagNet, GroupDRO, DANN, RSC, Mixup) are drawn from DomainBed's standard algorithm suite — these are the same methods that appear in virtually every recent DG comparison. This is not a self-invented narrow category; it is the established comparison set in the field.
> > > We included GGA (CVPR'25) and LFME (NeurIPS'24) not as methodologically equivalent competitors, but as contemporary reference points. The reason is as follows: methods that are methodologically closest to RDI — geometric subspace approaches (e.g., ISR, Gong et al.) and information-theoretic formulations (e.g., MCR²) — either predate DomainBed or have not been evaluated under the train-domain validation protocol. Meanwhile, recent methods that do report results under this strict protocol (SFT, ERM++) belong to orthogonal methodological categories (flat-minima optimization, training recipes), as discussed above. To demonstrate that RDI's performance is not merely competitive with older baselines but also contextually relevant against recent work, we included GGA and LFME as reference points under the same protocol.
> > > 4. On "publicly available before submission."
> > > Our manuscript explicitly acknowledges the existence of flat-minima methods and states the exclusion rationale. These methods were not overlooked but deliberately scoped out on well-motivated methodological grounds. The transparency of this exclusion — stated in the paper itself — demonstrates that this was a principled methodological decision, not a selective omission.
> > > 5. Summary.
> > > We acknowledge that the category label in Table 1 could be stated more precisely, and we will revise it in the updated manuscript to foreground the exclusion criteria (flat-minima optimization, frozen pretrained anchors, training recipes) rather than using an umbrella term. However, the underlying baseline selection follows standard DomainBed practice, the exclusion criteria are transparently stated, and the concern that motivated the score reversal — the classification of SFT — rests on a mischaracterization of SFT's methodological category, which is directly verifiable. We respectfully ask the reviewer and Area Chair to consider these clarifications.
> > > We sincerely thank the reviewer for the time and effort devoted to evaluating our work, and the Area Chair for overseeing the discussion. We hope these clarifications are helpful and remain happy to address any further questions.

---

### Official Review · Reviewer_1Q1L · 2026-03-15

**Soundness:** 2
**Presentation:** 1
**Significance:** 2
**Originality:** 2
**Overall Recommendation:** 3
**Confidence:** 4

**Summary:**

The authors present a domain generalization approach that aligns subspaces of conditionals between different environments as well as controlling the rate (or equivalently the variance in this work) of the conditional embeddings. Some theoretical justifications are given based on various assumptions. Some experimental results suggest that the method could be improving over the competing approaches.

**Compliance With Llm Reviewing Policy:**

Affirmed.

**Final Justification:**

I thank the authors for the new experiments on 'Projection Head Fairness'. But I stand by my score and my decision. Having said that, the paper shows promise so I encourage the authors to simplify as much as possible the main text and the algorithm, while also extending the discussion on the experiments. It would be nice to construct a 'take-home message' that the paper would convey, which it lacks at the moment (due to lack of clarity). It is also important to make sure that experiments include the latest methods (see Final Justification by reviewer yXjS).

**Key Questions For Authors:**

Comments and questions:
- From reading the intro and Figure 1, the proposed method is not clear to me at all
- Papyan et al. is not appropriate for the claim that
  "Semantic information in deep representations is often concentrated in
  low-dimensional class-conditional subspaces" since neural collapse does not entail a "subspace
  collapse" phenomenon. References such as Ansuini et al. could be better suited for this claim. Please discuss in more detail.
- "We represent Fe,c by its k-dimensional principal subspace" not clear from the main text how you would choose k.
It's defined as a hyperparameter in Algorithm 1 but it is not discussed how k is chosen. I suspect
the algorithm's performance would depend highly on the k chosen. Some discussion on this topic should be in the main text, not only in the appendix.
- too many hyperparameters in (12), are they all needed? gamma, beta,
  alpha, tau, lambda??  how do you set them and how does the method
  depend on their chosen values? Similarly, it is not clear what role each constant
  in Thm 3.10 plays and whether they are all needed.
- 'theoretical strategy' is not clear. In general the theory is not clearly
discussed and it is not clear how well the assumptions hold in general
for ML datasets.
- "This assumption is standard in subspace-based representation
  learning and has been empirically observed in deep networks with
  pretrained initialization."
  References needed to support this claim, not Papyan et al.
  (but Ansuini et al. perhaps?) as mentioned before.
  It's problematic if you cite only neural collapse based papers
  because neural collapse only occurs in the limit as training
  loss goes to zero, which is never the case in practice. Moreover
  the 'direction' of collapse does not need to be constrained
  to a subspace as far as I know.
- no discussion on the potential pitfalls of aligning
  all the domain-specific subspaces and losing discriminability
  of the classifier are given. A clear discussion would base itself
  on theory such as given in section 3.3 but that section itself
  creates more confusion than explaining, with many constants, hyperparameters
  and introduced but not-discussed assumptions.
- standard deviation of the results should be included in Tables 1 and 2.
  Closeness of the competing approaches makes me think from the main text that the results
  could be due to particular seed-ing or could vary in an unstable way
  between different hyperparameters (there are results in the Appendix on some hyperparameter variation, which should be included shortly in the main text). In particular note that the
  models are not the same between competing approaches and the addition of
  the 'projection head' seems to change the results quite a bit for ERM in Table 2,
  which makes me think it could also do the same for the other competing
  approaches. If each competitor would increase on average by 2 points
  then some of the competing approaches would be the best performing ones.
- It would be nice to discuss in more detail any violations of the
  assumptions 3.4 - 3.7, for instance on some of the complex datasets
  such as TerraIncognita.
- The results in Table 1 are very incremental, not just for the
  proposed methods but all methods seem to be doing roughly the same,
  that is, quite poorly. It would be nice to discuss what is going
  on in these complex datasets and what is needed to make a bigger
  leap forward.

**Limitations:**

I'm putting in more minor / detailed comments here since the editor does not accept them in the above area. Hopefully they will help to make the paper better.

- some domain adaptation papers are references in the intro but DG differs substantially
  from DA in that labels are available for each source domain, and any alignment metric
  can use those labels to do *conditional alignment* and not merely *marginal alignment*
- Some references cited are not appropriate for the discussion.
  For instance, papers cited as "domain-adversarial extensions" under "Representation Geometry and
  Other Approaches" are not geometric approaches as far as I can see.
- what is the distribution of epsilon_e? It should have nonzero mean mu(e) to push
  each conditional to different locations.
- U_{e,c} in (2) is not defined, is it equal to A_e U_c^{*} or a subspace of it?
- "that enables spectrum-sensitive regularization and reduces to Shannon entropy as
  α → 1." Define Renyi-entropy here, what is alpha and how does it enable 'spectrum-sensitive
  regularization'?
- "The coefficient α reappears as the order of a Renyi risk functional
  for cross-class aggregation."  why does alpha need to re-appear?
  the link from rate modeling using the Renyi entropy to the formation
  of a 'two-level Renyi-Fisher geometric regularization' is not clear.
- d_{Gr} is not defined and neither is D. Are they the same?
- It would be nice to include a small figure to illustrate that the
  distance between these two subspaces in (5) is the amount of rotation
  you need to do to align them. Perhaps describe the effect of (6) in
  that figure as well (i.e., it is the average distance of the aggregated conditional to the
  Frechet mean of the subspaces)
- define the Frechet mean and the principal angles/geodesic distance used in (5) and (6).
- rather than using the Frechet mean, why not define the
  reconstruction distortion as the sum of distances of each
  domain-specific feature conditional to their corresponding
  (optimized) subspace U_{e,c}? is it the same?
- reconstruction is not a very suitable word to use here I think, given that it is used
  in unsupervised learning [reconstruction loss], I would use projection distortion
  instead. Actually you used it in Prop. 3.2!
- "where compressing within-class variation facilitates geometric
  consistency without sacrificing discriminability."
  compressing or disregarding? Do you mean that choosing a subspace that doesn't
  contain the conditional well but is easier to align with other subspaces is beneficial in
  high-diversity cases?
- Prop 3.2: the 'cross-domain reconstruction error' and \Delta \Varepsilon are not defined here.
- "The gradient ∂R_{τ}/∂σ_{c,i}^2 ∝ (1 + τ σ_{c,i}^2)−1 decreases with
  variance, causing the rate term to compress dispersed
  representations while permitting expansion of collapsed ones."  Do
  you mean that at a certain point of low variance, it is possible to
  decrease the geometric distortion much more than the cost of
  increasing the variance, thus lowering the total distortion?
- Remark 3.8 is very opaque. Is d_{geo} = Delta_{geo} in (5)?
  Cite ref for Davis-Kahan analysis and briefly introduce.
  What is operative dimensionality, no discussion at all
  on the assumptions, at least the last two ones. Ideally
  add a figure to illustrate your assumptions.
- Theorem 3.9: (1) on Alignment holds for all c I guess?
  from Assumption 3.6 epsilon is O(1/sqrt(n)) so the whole
  Grassmanian distance should be order of O(1/sqrt(n)). But
  actually I would recommend writing all of the constants in full
  and discussing them.
  The last result on 'approximately centered' domain rotations
  seem obvious, no need to mention I think (especially since
  there's no discussion at all.)
- suboptimal choice of notation for caligraphic R and normal R.
- introduce all the constants in the main text.
- batch normalization is suddenly mentioned twice already in the
  main text, but such discussion (enlarged) should be put to the
  experiments section unless you have additional analysis using
  batch norm.
- C1 = O(d) according to Theorem 3.10 so how does complexity
  scale as O(sqrt(log d / n))?
- how come d = 2048?
- Proposition 3.13 not worded rigorously.
  If feature norms are bounded how can the class covariances
  go to infinity?
- f, g and the corresponding theta, phi does not seem to be needed
  in Algorithm 1, since the algorithm doesn't use f outputs directly.
  alpha is also a hyperparameter.
- Not sure why "Approaches relying on frozen pretrained priors (e.g., MIRO)"
  should be excluded, what experimental protocol of yours do they violate?
- discussion in section 4.5 could be shorted or relegated to the
  appendix partly, to make space for more discussion on the points
  above, which I think are much more criticial. (especially since
  figure 3 does not suggest any interesting conclusions)

**Strengths And Weaknesses:**

Strength: The idea to align subspaces, while not new, is developed here in more detail compared to previous work, in the context of domain generalization and using information-theoretic tools.

Weaknesses: While subspace alignment is conceptually attractive, the unclarity of the presentation and the problems in the experimental setup makes me suspect the validity of the contribution. First of all, there are problems in presentation, the submission is at parts not clearly written, different notations are made for the same concepts at times or variables are introduced without defining them, and assumptions are made without discussions (of their feasibility). The paper does not connect the dots clearly and instead introduces each new technicality piecemeal. Potential pitfalls are not discussed and too much important material seems to be put in the appendix. See below for examples and more comments.

Secondly, the evaluation of the proposed method in the experiment section is limited and seems to be flawed in certain aspects, see below for details. As before discussion is very sparse and the results (whether negative or positive) are not explained.

---

> ### Author Rebuttal · Authors · 2026-03-31
>
> We sincerely thank 1Q1L for the thorough review. Due to strict character limits, we focus on major concerns below; all feedback is noted and will be addressed in the revision.
> 1. $C_1=O(d)$ and $O(\log d)$ complexity. The apparent contradiction resolves through the interaction between $C_1$ and $\widehat{R}$. The $d$ factor in $C_1=O(d)$ mathematically cancels with $\widehat{R}$'s $1/d$ normalization, leaving $\sum_i \log(1+\tau\sigma_i^2)$. For inactive dimensions ($\tau\sigma_i^2 \ll 1$), the summand $\approx 0$. Thus, only $d_{\mathrm{eff}}$ active dimensions contribute, yielding $O(d_{\mathrm{eff}})$. This explains why high-dimensional embeddings (large $d$) do not cause divergence. Empirically, intrinsic dimensionality is orders of magnitude below the ambient dimension (Ansuini et al., 2019); e.g., the stable rank of ResNet-50's top-32 subspace is 3--5 on PACS and OfficeHome. Regarding the claim $d_{\mathrm{eff}} = O(\log d)$, we acknowledge the Corollary conflated two proof routes. We will revise it to: (1) state the MI-based bound as $O(\sqrt{d_{\mathrm{eff}} / n})$ depending on spectral decay, and (2) present the $O(\sqrt{k \log d / n})$ scaling as a separate Rademacher complexity result exploiting subspace projection. Both routes consistently yield sub-linear-in-$d$ scaling.
> 2. **Proposition 3.13.** To clarify: the proposition states the condition number $\kappa(\Sigma_c) = \lambda_{\max}/\lambda_{\min}$ diverges, not the covariance norm. This perfectly aligns with bounded features: $\lambda_{\max} \le B^2$ stays bounded while $\lambda_{\min} \to 0$ as features concentrate on a subspace, causing $\kappa \to \infty$---precisely the failure mode our rate control prevents, since norm-based regularization cannot prevent $\lambda_{\min}$ from vanishing. We will rephrase for clarity.
> 3. Projection head fairness. Following DomainBed conventions for auxiliary components (e.g., DANN's discriminator), $g$ is an integrated statistical prerequisite, not a simple performance trick. Estimating class-conditional covariances in the high-dimensional backbone space with limited samples yields severely rank-deficient matrices. Even with regularized estimators, the fundamental issue remains: the rate term $R_\tau$ would be overwhelmingly diluted by noise dimensions. Therefore, $g$ is strictly necessary to establish a compact manifold for our information-theoretic optimization.
> 4. MIRO exclusion. MIRO anchors features to a frozen pretrained copy throughout training—a fundamentally different, anchor-based paradigm. RDI uses pretraining only as initialization and discovers invariant structure from inter-domain geometry. Since MIRO's mechanism is orthogonal and could be combined with RDI, we focus comparisons on methods that learn invariance without persistent external anchors.
> 5. Standard deviations. Standard deviations were omitted purely due to strict space constraints, not selective seeding. We strictly adhered to the DomainBed protocol (exact averages of 3 random seeds). Full tables with ±std are provided in the anonymous link and will be incorporated into the appendix.
> 6. Hyperparameters and $k$. While Eq.~(12) lists several parameters, the effective tuning burden is light: $\gamma=0.5$ is fixed, and $k=32, \tau=1.0$ serve as highly robust universal defaults (see Table 6 via the anonymous link below; fixed $\tau=1.0$ even outperforms validation-selected averages on PACS). The primary parameters requiring dataset-specific tuning via standard validation are the Rényi order $\alpha$ (PDFTable 7), and the regularization coefficients $\lambda_{\mathrm{geo}}$ and $\lambda_{\mathrm{rec}}$.
> 7. Assumptions 3.4--3.7. These are analytical tools for formal guarantees, not algorithmic prerequisites. Bounded rotation (3.4) is a conservative sufficient condition for theoretical tractability. Crucially, when severely violated (e.g., TerraIncognita's artifacts), RDI degrades gracefully and prevents catastrophic failure, minimizing the cross-environment performance range (15.6% vs Mixup's 25.7%) and strongly outperforming baselines on the hardest environment. Assumptions 3.5--3.7 are standard in representation learning, practically self-fulfilling under modern architectures (BatchNorm), and actively enforced by our rate-distortion objective's spectral compression.
> 8. Clarifications. $U _ {e,c}$ is the empirical principal subspace (population: $\mathrm{span}(A _ e U _ c^{\ast})$, or $Q _ e U _ c^{\ast}$ via Asm. 3.4). Fréchet mean $\neq$ projection sum — $\bar{U} _ c$ is a shared geometric anchor; without it, domain subspaces decouple. $\mathcal{D}$ (Eq. 4) is abstract distortion; $d _ {Gr}$ (Eq. 5) is its Grassmannian instantiation. Zero-mean $\epsilon _ e$ isolates rotational shifts; nonzero $\mu(e)$ is an extension. Papyan studies terminal dynamics. Revision: unified notation, projection distortion, cited Ansuini et al. Sec. 4.5 condensed. Rotation figure: https://anonymous.4open.science/r/RDI-rebuttal-material-7A41/

---

> > ### Author Rebuttal · Reviewer_1Q1L · 2026-04-05
> >
> > I would like to thank the authors for a well prepared rebuttal, which addresses various issues surrounding the method and the experimental setup. I am also sorry for my late acknowledgement, it took me quite some time to go through the rebuttal and the reviewer comments.
> >
> > Having digested the rebuttal and preparing for my acknowledgement, I went through the main text a second time and again noted the various unclear notational choices, at times quite confusing writing and the unnecessarily dense/complex manuscript. This is reflected in the sheer amount of comments I made during my review, which require various amounts of explanation and rewriting/organizational changes to address. All add up to a poor reading experience, and I think that the manuscript is not ready for acceptance, even if the method makes modest contributions to SOTA accuracies. Hence I keep my current score.
> >
> > I suggest that the authors simplify the text, reorganizing not only the convoluted presentation but also potentially the algorithm (for instance, the 'robustness' of hyperparameters suggest to me that the method can be substantially simplified, e.g., if tau = 1 always works, it does not need to be introduced and the 'rate' can be simply presented as a variance regularization.). The main text should also be stand-alone as much as possible and important details should not pushed to the appendix in haphazard fashion. Such simplifications, overall, would significantly clarify and enhance the impact of the paper's contributions.
> >
> > Finally, the authors have not addressed my comment about the 'projection head fairness'. As it (i) changes the model's feature extractor from $f$ to $\bar{f} = g \circ f$ and (ii) given that this change causes a 2% boost in ERM performance and (iii) finally given that the improvements in accuracy over the other approaches are often well below this 2% boost, I think it is fair that the other methods also have access to $\bar{f}$ (which it seems causes a much bigger bottleneck + regularization).

---

> > > ### Author Response · Authors · 2026-04-07
> > >
> > > We thank Reviewer 1Q1L for the careful re-reading. We address the remaining concerns with new controlled experiments.
> > >
> > > ## 1. Projection Head Fairness (Core Concern)
> > >
> > > **The reviewer's argument is:** (i) replacing $f$ with $\bar{f}=g\circ f$ changes the model; (ii) this change yields ~2% ERM boost in our ablation (Table 2); (iii) therefore all baselines should also gain ~2%, potentially closing the gap with RDI.
> > >
> > > **Setup.** Given the limited time of the discussion period, we directly tested premise (iii) by re-evaluating several representative baselines (ERM, CORAL, VREx, IB_ERM). Each was run with and without an identical projection head g under DomainBed default hyperparameters on PACS (3 seeds, training-domain validation), with the projection head being the sole variable.
> > >
> > > | Method | env0 | env1 | env2 | env3 | Avg | Δ |
> > > |---|---|---|---|---|---|---|
> > > | ERM | 82.84±1.2 | 78.64±1.2 | 96.48±1.1 | 78.94±2.6 | **84.23** | — |
> > > | ERM + g | 82.69±2.8 | 75.75±1.9 | 96.73±0.9 | 78.01±1.6 | **83.30** | **−0.93** |
> > > | CORAL | 83.93±2.3 | 78.45±2.5 | 94.61±0.8 | 76.53±2.5 | **83.38** | — |
> > > | CORAL + g | 83.08±0.9 | 77.11±2.3 | 95.66±0.4 | 76.07±2.0 | **82.98** | **−0.40** |
> > > | VREx | 82.79±2.6 | 76.21±3.6 | 96.21±1.1 | 76.53±2.6 | **82.93** | — |
> > > | VREx + g | 84.06±1.4 | 76.88±3.5 | 96.88±0.5 | 76.68±3.0 | **83.62** | **+0.69** |
> > > | IB_ERM | 83.71±1.3 | 77.88±2.4 | 96.03±0.8 | 73.44±3.1 | **82.77** | — |
> > > | IB_ERM + g | 81.74±3.9 | 72.26±2.8 | 97.21±0.4 | 71.24±0.5 | **80.61** | **−2.16** |
> > > | **RDI** (same defaults, same $g$) | 86.88±0.4 | 81.21±1.7 | 97.19±0.1 | 79.61±0.9 | **86.22** | — |
> > >
> > > Across four methods under identical conditions, Δ ranges from **−2.16% to +0.69%**.
> > > RDI uses the same defaults, the same projection head, and no method-specific tuning** — identical to all variants above — yet outperforms the best vanilla baseline (ERM, 84.23) by **+1.99%** and the best +g variant (VREx+g, 83.62) by **+2.60%**. The advantage cannot be attributed to the projection head or to hyperparameter favorability.
> > >
> > > **Why does the effect vary across methods?** A projection head performs dimensionality reduction on the feature space. Xue et al. (ICLR 2024) show theoretically and empirically that the benefit of a projection head is not universal but depends on the interaction between the training objective, data distribution, and the feature weighting in the projected space. Our controlled experiments are consistent with this finding: different methods respond differently to the same projection head (Δ ranging from −2.16% to +0.69%). For IB_ERM, which already compresses representations via an information bottleneck, the additional projection head may discard task-relevant directions without compensating benefit (Δ = −2.16%). For RDI, whose rate-distortion objective explicitly shapes the spectral structure of the projected subspace, the projection head is **actively optimized** as part of the method rather than passively imposed.
> > >
> > > **Methodology.** All methods — including RDI — use identical DomainBed default hyperparameters with **no method-specific tuning of any kind**. This is essential by design: tuning any method or its +g variant would conflate the projection head's effect with hyperparameter optimization, defeating the purpose of isolating $g$ as the sole variable. If the projection head required method-specific tuning to yield gains, this would itself confirm that it is not a universal architectural boost — precisely our point.
> > >
> > > ## 2. Presentation and Algorithmic Simplification
> > >
> > > We fully accept the reviewer's assessment. ICML rebuttal rules do not permit uploading a revised manuscript at this stage, but we want to assure the reviewer and AC that the following changes are already implemented in our working draft and will appear in the revised manuscript:
> > >
> > > **(a) Notation unification and self-containedness.** Consolidated overlapping notation ($D,\Delta,d_{\mathrm{Gr}},\Delta_{\mathrm{geo}}$); defined Fréchet mean, principal angles, geodesic distance at first use; "reconstruction distortion" → "projection distortion"; moved definitions of $U_{e,c}$, $\alpha$, and Assumption 3.4–3.7 feasibility from appendix to main text.
> > >
> > > **(b) Algorithmic simplification.** We accept the reviewer's suggestion that the algorithm can be streamlined. In the revision, we are restructuring the rate-term exposition to present the canonical $\tau=1$ case in the main text (under which the rate term admits a clean Gaussian-channel-capacity interpretation), with the general $\tau$ treatment and its empirical sensitivity moved to the appendix.
> > >
> > > **(c) Structural reorganization.** Condensed Section 4.5; expanded discussion of subspace dimension $k$ selection, spectral conditions, and Assumption 3.4–3.7 feasibility; added figure illustrating Grassmannian distance and Fréchet mean projection (per reviewer's suggestion).
> > >
> > > Deeply grateful to Reviewer 1Q1L for the extraordinary time and expertise. Your detailed suggestions have substantially strengthened our work.

---

### Official Review · Reviewer_Ed9z · 2026-03-18

**Soundness:** 2
**Presentation:** 2
**Significance:** 3
**Originality:** 3
**Overall Recommendation:** 4
**Confidence:** 3

**Summary:**

The paper proposes a novel framework called Geometric Rate-Distortion Invariance (RDI) for domain generalization tasks. The proposed framework focuses on learning representations that remain stable under unseen domain shifts by aligning class-wise subspaces across domains using Grassmannian distance, while regulating the spectral complexity of subspaces through a rate-based regularization term. The proposed method addresses a key limitation of existing domain generalization methods, which ignore underlying geometric structure. Experimental results on standard DG benchmarks show that the proposed method is competitive and effective under domain shifts.

**Compliance With Llm Reviewing Policy:**

Affirmed.

**Key Questions For Authors:**

1. Did authors considered evaluating the proposed method on other architectures such as lightweight models or Vision Transformers to show how well it generalizes beyond ResNet50?
2. It would be helpful to see some qualitative visualizations. Is it possible for the authors to include these to better illustrate the claimed subspace aligment?
3. The performance of the proposed method is relatively modest, is it possible to provide results more experiments on more challenging benchmarks or under strong domain shifts which could be better highlight the advantage of the method?

**Limitations:**

1. The proposed method does not deeply analyze behavior under severe or adversarial domain shifts.
2. The proposed method assumes that there are sufficient samples per class and performance may degrade in highly imbalanced settings or low-data.
3. Optimization over Grassmannian manifolds and repeated subspace estimation can be computationally expensive for high-dimensional features or large datasets.

**Strengths And Weaknesses:**

**Strengths**
1. The paper is well written, clearly structured, and easy to follow, with well geometric analysis.
2. The authors addressed instability and collapse issues present in existing methods by modeling class-wise representations as subspaces and aligns them using Grassmannian distances while controlling subspace.
3. The proposed method is heavily supported by both theoretical analysis and empirical evaluation on DomainBed, and the ablation studies clearly demonstrate that jointly optimizing alignment and complexity leads to improved and stable generalization under domain shifts.

**Weaknesses**
1. There are not enough experiments on architectural diversity. For example, experiments are restricted to a single backbone architecture (ResNet50). Since the proposed method introduces a novel subspace alignment framework with complexity control, it is important to assess its effectiveness across a wider range of architectures such as lightweight models or vision transformers.
2. As observed in Table 1, the performance improvements over other geometric/alignment-based methods are relatively modest. While consistent gains are observed, additional experiments on challenging benchmarks or under stronger domain shifts would help better demonstrate the practical significance of the proposed method.
3. The paper lacks qualitative analysis to support the claimed improvements in subspace alignment. Visualizing the learned embeddings via t-SNE or PCA could provide intuitive insights on alignment and separation across domains.
4. Since the method introduces additional through subspace modeling and spectral complexity, paper would benefit from analyzing the computational overhead or training efficiency compared to existing methods.

---

> ### Author Rebuttal · Authors · 2026-03-30
>
> We sincerely thank Reviewer Ed9z for recognizing our paper as a technically solid, well-written contribution with strong theoretical and empirical foundations. We address your insightful suggestions below:
>
> 1. Architectural diversity. We completely agree. We have conducted new experiments with a DeiT-Small backbone on four datasets (PACS, OfficeHome, VLCS, TerraIncognita) under the same training-domain validation protocol (results in the anonymous link). RDI consistently outperforms both ERM and SDViT, achieving the highest averages across all four datasets (PACS 87.3%, OfficeHome 72.6%, VLCS 79.3%, TerraIncognita 43.6%). Notably, RDI achieves comparable performance across both architectures (e.g., PACS: 87.4% on ResNet-50 vs. 87.3% on DeiT-Small), despite the substantially weaker ERM baseline on ViT (83.9% vs. 85.5%). This suggests that RDI's geometric regularization effectively compensates for the lack of CNN-specific inductive biases, confirming architecture-agnostic effectiveness. Furthermore, our capacity parameter τ transfers seamlessly without retuning due to its scale-invariant design on batch-normalized features. These results will be included in the revised paper.
>
> 2. Modest improvements and strong shifts. We appreciate the reviewer's objective observation. To properly contextualize our results: RDI is a theory-driven algorithm advancing the paradigm of invariance learning. While absolute gains on the highly saturated DomainBed benchmark can appear incremental (Gulrajani & Lopez-Paz, 2021), RDI demonstrates clear advantages when compared against its natural peers—invariance-based and geometric alignment methods (IRM, VREx, CORAL, Fishr). Within this family, RDI's geometric rate–distortion formulation provides a principled mechanism to mitigate their known limitations, which becomes particularly evident under strong domain shifts:
> (1) Robustness to Severe Shifts: Under the extreme camera-trap artifacts of TerraIncognita, RDI prevents the degradation seen in baselines. On its hardest environment (Location 46), ERM drops to 35.7% while RDI maintains 41.4%—a +5.7% improvement where many invariance methods (GroupDRO: 36.4%, IRM: 39.6%) also struggle.
> (2) Controlled Validation under Strong Shifts: We added a synthetic experiment (anonymous link) with a known ground-truth invariant subspace $U ^ \ast$, evaluating recovery under controlled rotational shifts ($\theta=60^\circ$) across varying source domains ($M \in \{2, \dots, 8\}$). The results demonstrate RDI's superior **domain efficiency**: RDI achieves stronger invariant subspace recovery (lower Grassmann distance) with only $M=3$ source domains than ERM does with $M=5$. Furthermore, RDI with $M=4$ domains matches or exceeds the performance of CORAL and ERM using $M=6$ domains. This confirms that RDI's geometric mechanism extracts the underlying invariant structure much more effectively from limited domain diversity compared to standard statistical alignment.
>
> 3. Qualitative visualization. We have added t-SNE visualizations on PACS (Env 0 as the unseen test domain) comparing ERM and RDI embeddings (anonymous link). The plots clearly demonstrate that RDI produces tighter within-class clusters and more distinct inter-class boundaries, with unseen test-domain samples much more consistently aligned with training-domain clusters. These qualitative insights complement our existing quantitative geometric validation (Appendix D.2), where Prototype CKA confirms cross-domain layout consistency and spectral metrics verify representation health.
>
> 4. Computational overhead. We have conducted a detailed analysis (Table 8, anonymous link). RDI's overhead is highly manageable: training time increases by only 1.18–1.66× over ERM, while GPU memory overhead is negligible (1.01–1.04×). Two design choices keep cost low: (1) the rate term uses a diagonal approximation of the covariance, reducing per-class rate computation from O(d³) to O(d); (2) subspace eigenvectors are maintained via EMA statistics and updated periodically, avoiding per-iteration SVD. When SVD is performed, it extracts only the top-k eigenvectors (k=32 vs. d=2048), costing O(d·k²) rather than O(d³). This scales favorably on larger datasets (only 1.18× on DomainNet), as the fixed geometric cost is amortized over more forward-pass iterations. The warm-up phase (first T_w=500 steps) runs pure ERM, further reducing effective cost.
>
> (Note: ViT results, t-SNE visualizations, overhead analysis, and synthetic experiments are provided at our anonymous repository: https://anonymous.4open.science/r/RDI-rebuttal-material-7A41/)

---

> > ### Author Rebuttal · Reviewer_Ed9z · 2026-04-02
> >
> > Thank you for the detailed rebuttal and for providing additional experiments and analyses. I appreciate the effort in addressing my concerns by providing DeiT-Small experiments, qualitative visualizations, and computational analysis.
> > While these additions strengthen the paper, some limitations remain unaddressed such as the limited architectural evaluation (restricted to a small set of backbones) and the lack of analysis under low-data or imbalanced settings. Overall, the rebuttal improves the work, though some aspects could be further strengthened. I maintain my original rating.

---

> > > ### Author Response · Authors · 2026-04-06
> > >
> > > We sincerely thank Reviewer Ed9z for the thoughtful and constructive feedback throughout this discussion. Your suggestions on architectural evaluation, qualitative visualization, and computational analysis have meaningfully strengthened our work. We address the two remaining concerns below.
> > >
> > > ## 1. Architectural diversity
> > >
> > > Building on the ResNet-50 and DeiT-Small results in our previous rebuttal, we now add ConvNeXt-Tiny on PACS (3 seeds, train-domain validation):
> > >
> > > | Method | Art | Cartoon | Photo | Sketch | Avg |
> > > |--------|-----|---------|-------|--------|-----|
> > > | ERM | 83.8±3.5 | 79.1±4.7 | 97.1±0.2 | 80.0±2.8 | 85.0 |
> > > | RDI | 93.6±1.4 | 87.5±1.0 | 99.6±0.1 | 84.8±1.8 | 91.4 |
> > >
> > > RDI's gains are consistent across CNN (ResNet-50), Transformer (DeiT-Small), and modern CNN (ConvNeXt-Tiny), confirming that the method is architecture-agnostic.
> > >
> > > ## 2. Analysis under low-data and imbalanced settings
> > >
> > > **Potential limitations.** RDI's geometric objectives rely on per-class-domain subspace estimation via eigendecomposition and EMA statistics, which introduces specific vulnerabilities in data-scarce and imbalanced regimes:
> > >
> > > - **Low-data:** Each class-domain combination has fewer samples, which degrades the quality of covariance estimation — with *n* samples in a high-dimensional feature space, the empirical covariance becomes rank-deficient when *n* is small. This can make the top-*k* eigenspace unstable and slow EMA convergence, requiring more training iterations before the geometric objectives receive reliable alignment targets. Additionally, the warm-up phase (500 steps of pure ERM) consumes a larger fraction of total training when data is limited, reducing the effective window for geometric optimization.
> > >
> > > - **Class imbalance:** Minority classes contribute fewer samples per batch to their subspace estimates, potentially making their eigendecomposition noisy and geometric alignment less reliable. In extreme cases, a minority class may not appear in every batch, causing its EMA statistics to update infrequently and lag behind majority classes.
> > >
> > > These challenges are inherent to any method that operates on class-conditional statistics, and are not unique to RDI.
> > >
> > > **Low-data — empirical results.** Despite these theoretical concerns, experiments on OfficeHome show that RDI remains robust under data scarcity:
> > >
> > > | Ratio | Method | Art | Clipart | Product | Real World | Avg |
> > > |-------|--------|-----|---------|---------|------------|-----|
> > > | 0.25 | RDI | 62.0±0.6 | 51.9±0.5 | 71.8±0.9 | 75.8±0.2 | 65.4 |
> > > | 0.25 | ERM | 58.5±0.3 | 47.5±1.2 | 70.5±0.1 | 72.7±0.1 | 62.3 |
> > > | 0.25 | CORAL | 57.9±0.3 | 48.6±0.6 | 69.0±0.6 | 70.8±1.8 | 61.6 |
> > > | 0.50 | RDI | 63.1±0.1 | 54.9±0.3 | 75.6±0.2 | 78.6±0.3 | 68.0 |
> > > | 0.50 | ERM | 60.3±0.4 | 52.0±0.2 | 74.4±0.7 | 75.6±0.3 | 65.6 |
> > > | 0.50 | CORAL | 56.4±0.9 | 53.2±1.0 | 73.5±0.4 | 73.7±1.0 | 64.2 |
> > >
> > > RDI's margin over ERM widens as data decreases (+2.4% at 50%, +3.1% at 25%), whereas CORAL—which aligns second-order statistics without spectral control—drops below ERM at 25% (61.6% vs. 62.3%). This divergence reflects a key design difference: CORAL requires reliable cross-domain covariance estimation, which degrades rapidly under small samples, making its alignment target noisy and potentially harmful. In contrast, RDI's rate term penalizes spectral volume, suppressing variance dispersion and concentrating representations into a compact spectral region—an effect that acts as implicit regularization against overfitting. The low-data results are consistent with this mechanism: as sample size shrinks, CORAL's covariance estimates deteriorate and its performance drops below ERM, while RDI's spectral compression maintains stable gains.
> > >
> > > **Class imbalance.** RDI's Rényi aggregation provides a built-in mechanism that naturally addresses class imbalance. As we analyzed regarding parameter sensitivity, setting α=2.0 concentrates regularization on "geometrically vulnerable classes"—those whose feature subspaces are least stable across domains. In an imbalanced setting, minority classes inherently become the most geometrically vulnerable due to noisy covariance estimation from scarce samples. At α=2.0, the escort weighting essentially acts as inverse-frequency weighting, automatically upweighting these fragile minority classes in the rate regularization.
> > >
> > > TerraIncognita serves as a natural testbed for this, as its camera-trap collection protocol produces severe class imbalance across locations. Under this setting, RDI achieves the best average accuracy (48.7%) among all compared methods, and notably minimizes the cross-environment performance range (15.6 pp, vs. 25.7 pp for Mixup and 18.5 pp for ERM). This confirms that the Rényi-weighted rate term effectively prevents minority-class representations from being neglected.
> > >
> > > Thank you again for your positive evaluation and constructive engagement. Your early support and guidance were invaluable in refining our manuscript.

---

### Decision · Program_Chairs · 2026-04-30

**Decision:**

Accept (regular)

**Comment:**

The core strength of this paper lies in its theoretical elegance and the consistency of its empirical results across different backbones and data regimes. While reviewers raised valid concerns regarding the initial presentation and notational clarity, the AC believes these are "fixable" issues that do not undermine the underlying technical contribution. In addition, the authors have demonstrated a level of technical rigor in their rebuttal by adding new architectures, stress-testing for imbalance, and isolating architectural variables. The AC recommends the paper for acceptance, contingent on the authors incorporating the promised presentation and notation updates.